# Non-Chiral Vertex Operator Algebra Associated To Lorentzian Lattices And Narain CFTs

**Ranveer Kumar Singh,**[a] **Madhav Sinha**[b]

[a]*New High Energy Theory Center, Department of Physics and Astronomy, Rutgers University, 126 Frelinghuysen Rd., Piscataway NJ 08855, USA*

[b]*Department of Physics and Astronomy, Rutgers University, 126 Frelinghuysen Rd., Piscataway NJ 08855, USA*

*E-mail:* ranveersfl@gmail.com, ms3066@physics.rutgers.edu

ABSTRACT: Frenkel, Lepowsky, and Meurman constructed a vertex operator algebra (VOA) associated to any even, integral, Euclidean lattice. In the language of physics, these are examples of chiral conformal field theories (CFT). In this paper, we define non-chiral vertex operator algebra and some associated notions. We then give a construction of a non-chiral VOA associated to an even, integral, Lorentzian lattice and construct their irreducible modules. We obtain the moduli space of such modular invariant non-chiral VOAs based on self-dual Lorentzian lattices of signature $(m, n)$ assuming the validity of a technical result about automorphisms of the lattice. We finally show that Narain conformal field theories in physics are examples of non-chiral VOA. Our formalism helps us to identify the chiral algebra of Narain CFTs in terms of a particular sublattice and break its partition function into sum of characters.

# 1   Introduction

The study of vertex (operator) algebras started with the work of Borcherds [1] and Frenkel, Lepowsky, and Meurman [2] in relation to Monstrous Moonshine and two dimensional conformal field theory. The first non-trivial examples of the theory were constructed starting from an even, integral Euclidean lattice, see [3, 4] for a more physical construction. These are examples of what are called *chiral conformal field theory* in physics, see Subsection 1.1.1 for an introduction. There are ample examples of non-chiral conformal field theories which cannot be described mathematically in the language of vertex operator algebra. One large class of such theories is the Narain CFTs based on Lorentzian lattices.

In this paper, we define the notion of non-chiral vertex operator algebra and study various related notions. Our definition is based on the notion[1] of *full field algebras* introduced by Huang and Kong in [7] and *full vertex algebras* introduced by Moriwaki in [8]. We use formal calculus as well as complex analysis to formulate our axioms. We replace the Jacobi identity axiom of vertex (operator) algebras by a *locality* axiom which is general enough to imply the *duality* and hence operator product expansion of vertex operators. Various well-known examples of non-chiral conformal field theories in physics are examples of our definition. More concretely, we construct a class of examples of non-chiral VOAs based on Lorentzian lattices which cover the moduli space of Narain CFTs as examples of our definition. We then define modules and intertwiners of non-chiral VOAs on the lines of [9]. In the rest of this section, we describe a dictionary between non-chiral VOAs of this paper and the notion of (non-chiral) conformal field theory in physics.

## 1.1   The dictionary from non-chiral VOA to non-chiral CFTs

In this section, we give a dictionary between conformal field theories as generally understood in physics and the non-chiral vertex operator algebra definition in this paper. This dictionary, for the case of chiral conformal field theory, is well-known to experts. We extend it to the case of non-chiral VOA. We start by describing the defining data of a conformal field theory in physics.

### 1.1.1   Physical definition of a CFT

We begin with the Belavin-Polyakov-Zamalodchikov (BPZ) definition of a conformal field theory [10]. We follow [11–16] for this exposition.

A bosonic CFT is an inner product space $\mathscr{H}$ which[2] is decomposable as a direct

---

[1]See also [5, 6] for some earlier related but different discussion on non-chiral VOAs.
[2]Physically, one always has a Hilbert space.

sum of tensor product

$$\mathcal{H} = \bigoplus_{\substack{h,\bar{h} \\ h-\bar{h}\in\mathbb{Z}}} V(h,c) \otimes \overline{V}(\bar{h},\bar{c}) \,, \tag{1.1}$$

of irreducible highest weight modules of $\text{Vir}_c \times \overline{\text{Vir}}_{\bar{c}}$, where where $\text{Vir}_c$ and $\overline{\text{Vir}}_{\bar{c}}$ are two copies of the Virasoro algebra with central charge $c, \bar{c}$ :

$$[L_m, L_n] = (m-n)L_{m+n} + \frac{c}{12}m\left(m^2-1\right)\delta_{m+n,0}\,,$$
$$[\bar{L}_m, \bar{L}_n] = (m-n)\bar{L}_{m+n} + \frac{\bar{c}}{12}m\left(m^2-1\right)\delta_{m+n,0}\,, \tag{1.2}$$
$$[L_m, \bar{L}_n] = 0\,,$$

such that the following are satisfied :

1. *Identity Property* : There is a unique vector $|0\rangle \in V(0,c) \otimes \overline{V}(0,\bar{c})$ which is invariant under the $\mathfrak{sl}(2) \times \overline{\mathfrak{sl}(2)}$-subalgebra of $\text{Vir}_c \times \overline{\text{Vir}}_{\bar{c}}$ generated by $L_0, \bar{L}_0, L_{\pm 1}$, and $\bar{L}_{\pm 1}$.

2. For each vector $\alpha \in \mathcal{H}$ there is an operator $\phi_\alpha(z,\bar{z})$ acting on $\mathcal{H}$, parameterized by $z \in \mathbb{C}$. Also, for every operator $\phi_\alpha$ there exists a conjugate operator $\phi_{\alpha^\vee}$, partially characterized by the requirement that the operator product expansion (OPE) of $\phi_\alpha$ and $\phi_{\alpha^\vee}$ contains a descendant of the identity operator.

3. $L_n$ *Property* : For $\alpha \in V(h,c) \otimes \overline{V}(\bar{h},\bar{c})$ a highest weight state of the ($\text{Vir}_c \times \overline{\text{Vir}}_{\bar{c}}$)-action, we have

$$[L_n, \phi_\alpha(z,\bar{z})] = \left(z^{n+1}\frac{d}{dz} + h(n+1)z^n\right)\phi_\alpha(z,\bar{z})\,,$$
$$[\bar{L}_n, \phi_\alpha(z,\bar{z})] = \left(\bar{z}^{n+1}\frac{d}{d\bar{z}} + \bar{h}(n+1)\bar{z}^n\right)\phi_\alpha(z,\bar{z})\,, \tag{1.3}$$

for real numbers $h$ and $\bar{h}$.

4. *Duality Property* : The inner products $\langle 0|\phi_{\alpha_1}(z_1,\bar{z}_1)\dots\phi_{\alpha_n}(z_n,\bar{z}_n)|0\rangle$ exist for $|z_1| > \dots > |z_n| > 0$ and admit an unambiguous real-analytic continuation, independent of ordering[3], to $\mathbb{C}^n \setminus \{z_1,\dots,z_n = 0,\infty; z_i = z_j\}$.

5. *Modular invariance property* : The torus partition function and correlation functions, given in terms of traces exist and are modular invariant.

These axioms do not characterise a CFT but are necessary for a well defined CFT. A particular subset of operators which only depend only on $z$ or $\bar{z}$ are of interest. Such operators are called *chiral* ($z$ dependent) and *anti-chiral* ($\bar{z}$ dependent) operators.

---

[3]For Fermionic fields, one has to keep track of signs while commuting them past each other.

The set of chiral and anti-chiral operators form an algebra which we denote by $\mathscr{A}$ and $\overline{\mathscr{A}}$ respectively. For any CFT, $\mathbb{1} \in \mathscr{A} \otimes \overline{\mathscr{A}}$, $T(z) \in \mathscr{A}$ and $\overline{T}(\bar{z}) \in \overline{\mathscr{A}}$ where $T$ and $\overline{T}$ are the holomorphic and anti-holomorphic stress tensor with modes $L_n$ and $\bar{L}_n$ respectively. Let $\{\mathcal{O}^i(z)\}$ be a basis of $\mathscr{A}$. The OPE of chiral operators takes the form

$$\mathcal{O}^i(z)\mathcal{O}^j(w) = \sum_k \frac{c_{ijk}}{(z-w)^{h_{ijk}}}\mathcal{O}^k(w)\,, \tag{1.4}$$

for some coefficients $c_{ijk}$ where $h_{ijk} = h_i + h_j - h_k$. By the usual contour integral manipulations we can write the above OPE as the algebra of modes:

$$\left[\mathcal{O}^i_n, \mathcal{O}^j_m\right] = \sum_k c_{ijk}(n,m)\mathcal{O}^k_{n+m}\,, \tag{1.5}$$

where

$$\mathcal{O}^j(z) = \sum_m \mathcal{O}^j_m z^{-m-h_j}\,. \tag{1.6}$$

Similar OPE holds for anti-chiral operators. This is called the *chiral algebra* (*anti-chiral algebra*) of the CFT. In the following, we will only speak of the chiral algebra but all statements hold for anti-chiral algebra equally well. The chiral algebra of any CFT contains the (universal enveloping algebra of) Virasoro algebra since the stress tensor is always a chiral operator. Other examples of chiral algebra include the affine-Kac Moody algebra [17] and the $W_3$ algebra [18]. Only the zero mode the of chiral operator $\mathcal{O}(z)$ commute with the Hamiltonian $(L_0 + \bar{L}_0)$ and are hence called the *symmetry-generating algebra*. In the inner product space (1.1), one can talk about subspaces $\mathscr{H}_i$ which form irreducible representations of the chiral algebra $\mathscr{A}$. For this reason the full chiral algebra is sometimes also called the *spectrum-generating algebra*. We can thus decompose the physical Hilbert space $\mathscr{H}$ as:

$$\mathscr{H} = \bigoplus_{i,\bar{i}} N_{i,\bar{i}} \mathscr{H}_i \otimes \overline{\mathscr{H}}_{\bar{i}}\,, \tag{1.7}$$

where $\mathscr{H}_i, \overline{\mathscr{H}}_{\bar{i}}$ are irreducible representations of $\mathscr{A}, \overline{\mathscr{A}}$ respectively and $N_{i,\bar{i}} \in \mathbb{N}_0 = \mathbb{N} \cup \{0\}$, the set of non-negative integers, is the number of times $\mathscr{H}_i \otimes \overline{\mathscr{H}}_{\bar{i}}$ appears in $\mathscr{H}$. For the index value $i, \bar{i} = 0$, we take $\mathscr{H}_0$ and $\overline{\mathscr{H}}_0$ to be the subspace of $\mathscr{H}$ which contains states corresponding to $\mathscr{A}$ and $\overline{\mathscr{A}}$ respectively modulo the null states. Hence, $N_{i,0} = \delta_{i,0}, N_{0,\bar{i}} = \delta_{0,\bar{i}}$.

Let us now describe the relation between the two decompositions (1.1) and (1.7). A general state $|h, \bar{h}\rangle \in \mathscr{H}$ is called a *Virasoro primary* of conformal dimension $(h, \bar{h})$ if

$$\begin{aligned} L_0|h,\bar{h}\rangle &= h|h,\bar{h}\rangle \quad \bar{L}_0|h,\bar{h}\rangle = \bar{h}|h,\bar{h}\rangle \\ L_n|h,\bar{h}\rangle &= \bar{L}_n|h,\bar{h}\rangle = 0, \quad n > 0. \end{aligned} \tag{1.8}$$

This follows from the OPE

$$T(z)\phi(w,\bar{w}) = \frac{h}{(z-w)^2}\phi(w,\bar{w}) + \frac{\partial_w\phi(w,\bar{w})}{z-w} + \text{ hol. },$$

$$\overline{T}(\bar{z})\phi(w,\bar{w}) = \frac{\bar{h}}{(\bar{z}-\bar{w})^2}\phi(w,\bar{w}) + \frac{\partial_{\bar{w}}\phi(w,\bar{w})}{\bar{z}-\bar{w}} + \text{ anti-hol. },$$

$$(1.9)$$

where $\phi(w,\bar{w})$ is the operator corresponding to the $|h,\bar{h}\rangle$. One can identify the state $|h,\bar{h}\rangle$ with

$$|h,\bar{h}\rangle \equiv \lim_{z,\bar{z}\to 0}\phi(z,\bar{z})|0\rangle . \tag{1.10}$$

We will consider $|h,\bar{h}\rangle$ as the tensor product of states $|h\rangle, |\bar{h}\rangle$: $|h,\bar{h}\rangle \equiv |h\rangle \otimes |\bar{h}\rangle$. The *Verma modules* $V(h,c)$ and $\overline{V}(\bar{h},\bar{c})$ is given by

$$V(h,c) := \text{Span}_{\mathbb{C}}\{L_{-n_1}\ldots L_{-n_k}|h\rangle : n_1,\ldots,n_k > 0, k \in \mathbb{N}\},$$

$$\overline{V}(\bar{h},\bar{c}) := \text{Span}_{\mathbb{C}}\{\bar{L}_{-n_1}\ldots \bar{L}_{-n_k}|\bar{h}\rangle : n_1,\ldots,n_k > 0, k \in \mathbb{N}\}.$$

$$(1.11)$$

The commutators (1.2) and the Virasoro primary condition makes the Verma module $V(h,c) \otimes \overline{V}(\bar{h},\bar{c})$ into a $(\text{Vir}_c \times \overline{\text{Vir}}_{\bar{c}})$-representation. In general, this representation is *reducible* and one has to quotient out *singular* or *null states* [4] from these Verma modules to make them irreducible. We will assume that this has already been done and that $V(h,c) \otimes \overline{V}(\bar{h},\bar{c})$ is an irreducible $(\text{Vir}_c \times \overline{\text{Vir}}_{\bar{c}})$-module. Note that from the OPE (1.9) we must have $\bar{h} = 0$ ($h = 0$) for a chiral (anti-chiral field). Now we can identify $\mathscr{H}_0 \otimes \overline{\mathscr{H}}_0$ as

$$\mathscr{H}_0 \otimes \overline{\mathscr{H}}_0 \cong \left(\bigoplus_{h\in\mathcal{S}} V(h,c)\right) \otimes \left(\bigoplus_{\bar{h}\in\overline{\mathcal{S}}} \overline{V}(\bar{h},\bar{c})\right) \subset \mathscr{H}, \tag{1.12}$$

where

$$\mathcal{S} := \{h \in \mathbb{Z} : |h\rangle \otimes |0\rangle \equiv |h,0\rangle \in \mathscr{H}\},$$

$$\overline{\mathcal{S}} := \{\bar{h} \in \mathbb{Z} : |0\rangle \otimes |\bar{h}\rangle \equiv |0,\bar{h}\rangle \in \mathscr{H}\}.$$

$$(1.13)$$

A *chiral primary* is a Virasoro primary $|h,\bar{h}\rangle$ satisfying

$$\mathcal{O}_m|h,\bar{h}\rangle = \overline{\mathcal{O}}_m|h,\bar{h}\rangle = 0, \quad m > 0, \tag{1.14}$$

and is an eigenstate of $\mathcal{O}_0$ and $\overline{\mathcal{O}}_0$ for every $\mathcal{O} \in \mathscr{A}$ and $\overline{\mathcal{O}} \in \overline{\mathcal{A}}$, the operator corresponding to it will be called the *chiral primary field*. Then each irreducible factor in (1.7) is a subspace of a Verma module constructed over a chiral primary by the modes of every chiral and anti-chiral operator.

---

[4]A singular or null state is a vector which is not a highest weight vector but is annihilated by $L_n, \bar{L}_n$, $n > 0$, see [19, Section 7.1.3] for a detailed discussion.

One can then talk about characters of the CFT defined for each irreducible factor $\mathscr{H}_i \otimes \overline{\mathscr{H}_{\bar{i}}}$:

$$\chi_{i,\bar{i}}(\tau, \bar{\tau}) = \mathrm{Tr}_{\mathscr{H}_i \otimes \overline{\mathscr{H}_{\bar{i}}}} q^{L_0 - \frac{c}{24}} \bar{q}^{\bar{L}_0 - \frac{\bar{c}}{24}}, \quad q = e^{2\pi i \tau}, \quad \bar{q} = e^{-2\pi i \bar{\tau}}, \tag{1.15}$$

where $\tau \in \mathbb{H} := \{x + iy \in \mathbb{C} : y > 0\}$ is in the upper half plane. From the fact that $\mathscr{H}_i \otimes \overline{\mathscr{H}_{\bar{i}}}$ is built over some highest vector $|h_i, \bar{h}_i\rangle$, we see that the character has the form

$$\chi_{i,\bar{i}}(\tau, \bar{\tau}) = q^{h_i - \frac{c}{24}} \bar{q}^{\bar{h}_i - \frac{\bar{c}}{24}} \sum_{n,m=0}^{\infty} a(n) \bar{a}(m) q^n \bar{q}^m, \tag{1.16}$$

for some integers $a(n), \bar{a}(m)$. Thus we can separate the character into *chiral* and *anti-chiral characters*:

$$\chi_{i,\bar{i}}(\tau, \bar{\tau}) = \chi_i(\tau) \overline{\chi_{\bar{i}}}(\bar{\tau}) \tag{1.17}$$

where

$$\chi_i(\tau) = q^{h_i - \frac{c}{24}} \sum_{n=0}^{\infty} a(n) q^n, \quad \overline{\chi_{\bar{i}}}(\bar{\tau}) = \bar{q}^{\bar{h}_i - \frac{\bar{c}}{24}} \sum_{m=0}^{\infty} \bar{a}(m) \bar{q}^m. \tag{1.18}$$

The partition function of the CFT is then given by sum over chiral characters

$$Z(\tau, \bar{\tau}) := \mathrm{Tr}_{\mathscr{H}} q^{L_0 - \frac{c}{24}} \bar{q}^{\bar{L}_0 - \frac{\bar{c}}{24}} = \sum_{i,\bar{i}} N_{i,\bar{i}} \chi_i(\tau) \overline{\chi_{\bar{i}}}(\bar{\tau}). \tag{1.19}$$

Modular invariance of the partition function implies that the chiral characters form a weight-zero weakly holomorphic vector valued modular form [5]. Using this property of chiral characters, one can classify CFTs with finitely many primary fields and given central charge [16, 21–27]. A CFT is called a chiral CFT if the partition function and its correlation functions decompose into a product of an analytic function of $\tau$ and an analytic function of $\bar{\tau}$. In such a case, we say that the CFT admits a *holomorphic factorisation*. A non-chiral conformal field theory does not admit a holomorphic factorisation.

### 1.1.2 Non-chiral vertex operator algebras

We now briefly describe the notion of non-chiral vertex operator algebras studied in this paper. We refer to Section 2 for precise and detailed discussions. Just as vertex operator algebras, we start with a vector space $V$ which is $(\mathbb{R} \times \mathbb{R})$-graded. We have the vertex operator map $Y_V : V \longrightarrow \mathrm{End}(V)\{x, \bar{x}\}$ where $x, \bar{x}$ are two formal variables. We require the existence of a vacuum vector $\mathbf{1}$ and conformal vectors $\omega, \bar{\omega}$ such that $Y_V(\mathbf{1}, x, \bar{x}) = \mathbb{1}$ and the coefficients of the formal series for $Y_V(\omega, x, \bar{x}), Y_V(\bar{\omega}, x, \bar{x})$ satisfies two copies of the Virasoro algebra. The vertex operators are required to satisfy certain translation and $L(0)$ axioms similar to VOAs.

---

[5]See for example [20] for the definition of vector valued modular forms.

It turns out that that the Jacobi identity for VOAs is difficult to formulate for non-chiral VOAs [8]. The appropriate locality axiom is motivated from full field algebras of [7]. Roughly speaking, locality of vertex operators says that the matrix elements of the product $Y_V(v, z_1, \bar{z}_1) Y_V(w, z_2, \bar{z}_2)$ and $Y_V(v, z_1, \bar{z}_1) Y_V(w, z_2, \bar{z}_2)$, defined when $|z_1| > |z_2|$ and $|z_2| > |z_1|$ respectively, are equal to the same function, which is multi-valued and analytic in $z_1, \bar{z}_1, z_2, \bar{z}_2$ and single valued when $\bar{z}_1, \bar{z}_2$ are complex-conjugates of $z_1, z_2$, defined on $\mathbb{C}^4$ minus a diagonal subset. This version of locality turns into a statement of analytic continuation of matrix elements for chiral vertex operators and allows us to use contour integration for manipulating the modes of the vertex operators. Moreover, locality allows us to prove *duality* of vertex operators which in turn gives us the operator product expansion of the product of two vertex operators. Modules and intertwiners of a non-chiral VOA are then defined analogous to modules of a VOA.

### 1.1.3 The Dictionary

Let us now describe the dictionary between non-chiral VOA and its modules and a non-chiral CFT. Given a bosonic CFT, its chiral and anti-chiral algebra is a non-chiral VOA according to our definition. Note that our notion of non-chiral VOA allows for more general structure in the sense that the chiral algebra of a CFT always has the structure of a tensor product $\mathcal{H}_0 \otimes \overline{\mathcal{H}}_0$ as described above but non-chiral VOAs are allowed to have more general vector spaces. The chiral and anti-chiral operators are identified with the vertex operators of the VOA. For general non-chiral VOA, chiral and anti-chiral vertex operators form only a subsector of the set of vertex operators, see Definition 2.2 and Theorem 2.1 below. Next, the irreducibles $\mathcal{H}_i \otimes \overline{\mathcal{H}}_{\bar{i}}$ must be identified with modules of the VOA. Again in our generic construction we allow for the modules to have more general structure rather than a tensor product. The chiral primary field corresponding to an irreducible $\mathcal{H}_i \otimes \overline{\mathcal{H}}_{\bar{i}}$ is identified with the intertwiner $\mathcal{Y}_{(i,\bar{i})(0,0)}^{(i,\bar{i})}(w, x, \bar{x})$ of type $\binom{(i,\bar{i})}{(i,\bar{i})(0,0)}$ where $(0,0)$ indicates the VOA as a module for itself. The state-operator correspondence for the space (1.7) corresponds to the vertex operator map for the VOA and its modules along with the intertwining operators of type $\binom{(i,\bar{i})}{(i,\bar{i})(0,0)}$. The full dictionary is summarised in Table 1. We hope to expand the dictionary to include fusion rules and rationality on the CFT side to the notion of tensor product of modules and (strong) regularity of non-chiral VOAs in a future publication.

| Non-chiral CFT | Non-chiral VOA |
|---|---|
| Chiral space of states $\mathscr{H}_0 \otimes \overline{\mathscr{H}}_0$ (1.12) | $(\mathbb{R} \times \mathbb{R})$-graded vector space $V$ (2.12) |
| State-operator map $\alpha \mapsto \phi_\alpha$ (2) on Page 3 | Vertex operator map $v \mapsto Y_V(v, x, \bar{x})$ (2.15) |
| Stress tensor $T(z), \overline{T}(\bar{z})$ | Conformal vertex operators $Y_V(\omega, x), Y_V(\bar{\omega}, \bar{x})$ (2.26) |
| Duality of operators (4) on Page 3 | Locality of vertex operators Def. 2.1 (9) |
| Chiral and anti-chiral algebra (1.5) | Chiral and anti-chiral algebra (2.73) |
| Irreducible representation $\mathscr{H}_i \otimes \overline{\mathscr{H}}_{\bar{i}}$ of $\mathscr{A} \otimes \overline{\mathscr{A}}$ | Irreducible module $(W, Y_W)$ of non-chiral VOA $(V, Y_V)$ Def. 4.1 |
| Characters $\chi_{i,\bar{i}}(\tau, \bar{\tau})$ (1.15) | Graded dimension $\chi_W(\tau, \bar{\tau})$ (4.21) |
| Chiral primary operator $\phi_\alpha$ for $\alpha \in \mathscr{H}_i \otimes \overline{\mathscr{H}}_i$ | Intertwiner $\mathcal{Y}^{(i,i)}_{(i,\bar{i})(0,0)}(w, x, \bar{x})$ of type $\binom{(i,\bar{i})}{(i,\bar{i})(0,0)}$ Def. 4.2 |

**Table 1**: Dictionary between a CFT and non-chiral VOA

## 1.2 Lorentzian Lattice Vertex Operator Algebra (LLVOA)

One of the main constructions of this paper is a concrete example of a non-chiral VOA based on an even integral Lorentzian lattice. The construction is similar in spirit to Euclidean lattice VOAs but differs in that it is inherently non-chiral and does not admit holomorphic factorisation thus providing the first non-trivial example of a non-chiral VOA. We call it the Lorentzian lattice vertex operator algebra (LLVOA). The modules for a non-chiral VOA introduced in this paper can be defined analogous to modules for usual VOAs. Infact, we are able to construct modules for the LLVOA in one to one correspondence with certain cosets of the lattice. Collecting the VOA and its modules, we define the partition function of the non-chiral CFT (see Definition 4.3) thus obtained. We find that the modules of the LLVOA constructed using the cosets of the lattice give rise to a modular invariant partition function. We further attempt to classify all possible modular invariant non-chiral CFT based on even, integral self-dual Lorentzian lattices of a given signature. This leads us to a conjecture about automorphisms of Lorentzian lattices which we prove for signature $(m, m)$ but are unable to prove for general signature $(m, n)$ with $m \neq n$. Following the physics terminology, we call the equivalence classes of non-chiral CFTs based on Lorentzian lattices as the moduli space of LLVOAs. As expected from physical arguments, the moduli space in signature $(m, n)$ is given by (see Theorem 5.5)

$$\mathcal{M}_{m,n} \cong \frac{\mathrm{O}(m, n, \mathbb{R})}{\mathrm{O}(m, \mathbb{R}) \times \mathrm{O}(n, \mathbb{R}) \times \mathrm{O}(m, n, \mathbb{Z})}. \tag{1.20}$$

The paper is organised as follows: in Section 2, we introduce the notion of non-chiral VOA and prove some elementary consequences of the definition. Then in Section 3, we construct the LLVOA and prove that it is an example of a non-chiral VOA. In Section 4 we define the notion of modules and intertwining operators and prove some important consequences and results required later. In Section 5 we construct the modules of LLVOAs. We formulate a precise conjecture about automorphism of Lorentzian lattices and under that assumption, prove that the moduli space of LLVOAs in signature $(m, n)$ is given by (1.20). Finally in Section 6 we review the physical construction of Narain CFTs and comment on their relation to LLVOAs. Appendix A deals with the independence of central extensions of lattices on the chosen basis of the lattice. Appendix C contains some technical results about modules of Heisenberg algebras and Appendix D contains the proof of Conjecture 1 for the special case $m = n$.

# 2 Non-Chiral Vertex Operator Algebra

## 2.1 Formal calculus

We begin by collecting some notatations about formal calculus. The reader is referred to [2, Chapter 3,8] and [28, Chapter 2] for more details.

Let $x$ be a formal variable. For a vector space $V$, we define the following:

$$V[x] = \left\{ \sum_{n \in \mathbb{N}_0} v_n x^n : v_n \in V, \text{ where only finitely many } v_n \neq 0 \right\}, \qquad (2.1)$$

$$V[[x]] = \left\{ \sum_{n \in \mathbb{N}_0} v_n x^n : v_n \in V \right\}, \qquad (2.2)$$

$$V\{x\} = \left\{ \sum_{n \in \mathbb{F}} v_n x^n : v_n \in V \right\}, \qquad (2.3)$$

where $\mathbb{N}_0 = \mathbb{N} \cup \{0\}$ and $\mathbb{F}$ is an arbitrary field of characteristic not 2. We will mostly be interested in the case $\mathbb{F} = \mathbb{R}$ or $\mathbb{C}$. For a complex number $s \in \mathbb{C}$ and formal variables $x_1, x_2$, we will define

$$(x_1 + x_2)^s := \sum_{n=0}^{\infty} \binom{s}{n} x_1^{s-n} x_2^n, \qquad (2.4)$$

where the binomial coefficient is defined as

$$\binom{s}{n} = \frac{s(s-1)\cdots(s-n+1)}{n!}. \qquad (2.5)$$

Note, in a series like this, we will always have non-negative integral powers of the second variable. With this formula, one can check that

$$\left( 1 - \frac{x_1}{x_2} \right)^s x_2^s = (x_2 - x_1)^s. \qquad (2.6)$$

If we replace $x_1, x_2$ by complex variables $z_1, z_2$ then by definition [29]

$$\begin{aligned}
(z_1 - z_2)^s &:= \exp(s \log(z_1 - z_2)), \\
(z_2 - z_1)^s &:= \exp(s \log(z_2 - z_1)).
\end{aligned} \qquad (2.7)$$

Using the fact that

$$\log(1 - z) = -\sum_{n=0}^{\infty} \frac{z^n}{n}, \quad |z| < 1, \qquad (2.8)$$

and the identity [6]

$$(-1)^k \binom{s}{k} = \sum_{\ell=1}^{k} \frac{(-s)^\ell}{\ell!} \sum_{\substack{n_1+\cdots+n_\ell=k \\ n_1,\dots,n_\ell \geq 1}} \frac{1}{n_1 \cdots n_\ell}, \tag{2.9}$$

it is easy to see that [7]

$$\begin{aligned}
(z_1 - z_2)^s &= \exp(s \log(z_1 - z_2)) = \sum_{n \geq 0} \binom{s}{n} (-1)^n z_1^{s-n} z_2^n, \quad |z_1| > |z_2|, \\
(z_2 - z_1)^s &= \exp(s \log(z_2 - z_1)) = \sum_{n \geq 0} \binom{s}{n} (-1)^n z_2^{s-n} z_1^n, \quad |z_2| > |z_1|,
\end{aligned} \tag{2.10}$$

which is consistent with the definition (2.4). Let $f(x) = \sum v_n x^n \in V[[x, x^{-1}]]$, then we have the formal version of Taylor's theorem:

$$e^{x_0 \frac{d}{dx}} f(x) = f(x + x_0), \tag{2.11}$$

One can prove this by expanding both sides and comparing terms of equal power of $x, x_0$. As before, we need to expand the RHS in non-negative integral powers of $x_0$.

## 2.2 Definition of non-chiral VOA

Let

$$V = \coprod_{(h,\bar{h}) \in \mathbb{R} \times \mathbb{R}} V_{(h,\bar{h})}, \tag{2.12}$$

be an $\mathbb{R} \times \mathbb{R}$-graded complex vector space vector space. Let

$$\overline{V} = \prod_{(h,\bar{h}) \in \mathbb{R} \times \mathbb{R}} V_{(h,\bar{h})}, \tag{2.13}$$

denote the algebraic completion of $V$. Let

$$V' = \coprod_{(h,\bar{h}) \in \mathbb{R} \times \mathbb{R}} V'_{(h,\bar{h})}, \tag{2.14}$$

be the contragradient of $V$ where $V'_{(h,\bar{h})}$ is the dual of $V_{(h,\bar{h})}$. A series $\sum f_n$ in $\overline{V}$ is said to be absolutely convergent if for every $f' \in V'$ the series $\sum |\langle f', f_n \rangle|$ is convergent. Here, $\langle f', f_n \rangle = f'(f_n) \in \mathbb{C}$ is just the action of the linear functional on $f'$ on $f_n$.

---

[6]This identity can be proven by using the relation $(1-x)^s = \exp(s \log(1-x))$ for any real $x$ with $|x| < 1$ and $s \in \mathbb{C}$.

[7]Another way of proving this identity is to use Taylor's theorem.

**Definition 2.1.** A *non-chiral vertex operator algebra* of *central charge* $(c, \bar{c})$ is a quintuple $(V, Y_V, \omega, \bar{\omega}, \mathbf{1})$ where $V$ is an $\mathbb{R} \times \mathbb{R}$-graded complex vector space and $Y_V$ is a linear map, called the *vertex operator map*,

$$
\begin{aligned}
Y_V : V \otimes V &\longrightarrow V\{x^{\pm 1}, \bar{x}^{\pm 1}\}\,, \\
u \otimes v &\longmapsto Y_V(u, x, \bar{x})v\,,
\end{aligned}
\tag{2.15}
$$

or equivalently a map

$$
\begin{aligned}
Y_V : \mathbb{C}^\times \times \mathbb{C}^\times &\longrightarrow \mathrm{Hom}(V \otimes V, \overline{V}) \\
(z, \bar{z}) &\longmapsto Y_V(\cdot, z, \bar{z}) : u \otimes v \longmapsto Y_V(u, z, \bar{z})v,
\end{aligned}
\tag{2.16}
$$

which is multi-valued and analytic if $z, \bar{z}$ are independent complex variables and single valued when $\bar{z}$ is the complex conjugate of $z$. The vertex operator $Y_V(u, x, \bar{x})$ is expanded as a formal power series

$$
Y_V(u, x, \bar{x}) = \sum_{m,n \in \mathbb{R}} u_{m,n} x^{-m-1} \bar{x}^{-n-1} \in \mathrm{End}(V)\{x^{\pm 1}, \bar{x}^{\pm 1}\}\,,
\tag{2.17}
$$

and when $u \in V_{(h, \bar{h})}$, it can also be expanded as

$$
Y_V(u, x, \bar{x}) = \sum_{m,n \in \mathbb{R}} x_{m,n}(u) x^{-m-h} \bar{x}^{-n-\bar{h}} \in \mathrm{End}(V)\{x^{\pm 1}, \bar{x}^{\pm 1}\}\,,
\tag{2.18}
$$

so that

$$
x_{m,n}(u) = u_{m+h-1, n+\bar{h}-1}\,, \quad m, n \in \mathbb{Z}\,.
\tag{2.19}
$$

We call $x_{m,n}(u)$ the *modes* of the vertex operators $Y_V(u, x, \bar{x})$. The degree $(h, \bar{h})$ is called the *conformal weight* of $u \in V_{(h, \bar{h})}$ and we write

$$
\mathrm{wt}(u) = h\,, \quad \overline{\mathrm{wt}}(u) = \bar{h}\,.
\tag{2.20}
$$

The vector $\mathbf{1} \in V_{(0,0)}$ is called the *vacuum vector* and $\omega \in V_{(2,0)}, \bar{\omega} \in V_{(0,2)}$ are *chiral* and *anti-chiral conformal vectors* respectively. This data is required to satisfy the following properties:

1. *Identity property:* The vertex operator corresponding to the vacuum vector acts as identity, i.e.
$$
Y_V(\mathbf{1}, x, \bar{x})u = u\,, \quad \forall\; u \in V.
\tag{2.21}
$$

2. *Grading-restriction property:* For every $(h, \bar{h}) \in \mathbb{R} \times \mathbb{R}$,
$$
\dim(V_{(h, \bar{h})}) < \infty\,,
\tag{2.22}
$$
and there exists $M \in \mathbb{R}$, such that
$$
V_{(h, \bar{h})} = 0\,, \quad \text{for } h < M \text{ or } \bar{h} < M.
\tag{2.23}
$$

3. *Single-valuedness property:* For every homogenous subspace $V_{(h,\bar{h})}$

$$h - \bar{h} \in \mathbb{Z}. \tag{2.24}$$

4. *Creation property:* For any $v \in V$

$$\lim_{x,\bar{x} \to 0} Y_V(v, x, \bar{x})\mathbf{1} = v , \tag{2.25}$$

that is $Y_V(v, x, \bar{x})\mathbf{1}$ involves only non-negative powers of $x, \bar{x}$ and the constant term is $v$.

5. *Virasoro property:* The vertex operators $Y_V(\omega, x, \bar{x})$ and $Y_V(\bar{\omega}, x, \bar{x})$, called *conformal vertex operators*, have Laurent series in $x, \bar{x}$ given by

$$
\begin{aligned}
T(x) &:= Y_V(\omega, x, \bar{x}) = \sum_{n \in \mathbb{Z}} L(n) x^{-n-2}, \\
\bar{T}(\bar{x}) &:= Y_V(\bar{\omega}, x, \bar{x}) = \sum_{n \in \mathbb{Z}} \bar{L}(n) \bar{x}^{-n-2},
\end{aligned}
\tag{2.26}
$$

where $L(n), \bar{L}(n)$ are operators which satisfy the *Virasoro algebra* with central charge $c, \bar{c}$ respectively:

$$
\begin{aligned}
[L(m), L(n)] &= (m - n)L(m + n) + \frac{c}{12}m\left(m^2 - 1\right)\delta_{m+n,0}, \\
\left[\bar{L}(m), \bar{L}(n)\right] &= (m - n)\bar{L}(m + n) + \frac{\bar{c}}{12}m\left(m^2 - 1\right)\delta_{m+n,0},
\end{aligned}
\tag{2.27}
$$

and

$$\left[L(m), \bar{L}(n)\right] = 0. \tag{2.28}$$

6. *Grading property:* The operator $(L(0), \bar{L}(0))$ is the grading operator on $V$, that is for $v \in V_{(h,\bar{h})}$

$$L(0)v = hv, \quad \bar{L}(0)v = \bar{h}v. \tag{2.29}$$

7. *$L(0)$-property :*

$$
\begin{aligned}
[L(0), Y_V(u, x, \bar{x})] &= x\frac{\partial}{\partial x}Y_V(u, x, \bar{x}) + Y_V(L(0)u, x, \bar{x}), \\
[\bar{L}(0), Y_V(u, x, \bar{x})] &= \bar{x}\frac{\partial}{\partial \bar{x}}Y_V(u, x, \bar{x}) + Y_V(\bar{L}(0)u, x, \bar{x}).
\end{aligned}
\tag{2.30}
$$

8. *Translation property:* For any $u \in V$

$$
\begin{aligned}
[L(-1), Y_V(u, x, \bar{x})] &= Y_V\left(L(-1)u, x, \bar{x}\right) = \frac{\partial}{\partial x}Y_V(u, x, \bar{x}), \\
\left[\bar{L}(-1), Y_V(u, x, \bar{x})\right] &= Y_V\left(\bar{L}(-1)u, x, \bar{x}\right) = \frac{\partial}{\partial \bar{x}}Y_V(u, x, \bar{x}).
\end{aligned}
\tag{2.31}
$$

9. *Locality property:* For $u_1, \ldots, u_n \in V$, there is an operator-valued function [8]

$$m_n(u_1, ..., u_n; z_1, \bar{z}_1, ..., z_n, \bar{z}_n) \, ,$$

defined on [9]

$$\{(z_1, \ldots, z_n, \bar{z}_1, \ldots, \bar{z}_n) \in \mathbb{C}^{2n} \,|\, z_i, \bar{z}_i \neq 0, z_i \neq z_j, \bar{z}_i \neq \bar{z}_j\}, \qquad (2.32)$$

which is multi-valued and analytic when $\bar{z}_1, ..., \bar{z}_n$ are viewed as independent variables and is single-valued when $\bar{z}_1, ..., \bar{z}_n$ are equal to the complex conjugates of $z_1, ..., z_n$. Moreover, for any permutation $\sigma \in S_n$, the product of vertex operators

$$Y_V\left(u_{\sigma(1)}, z_{\sigma(1)}, \bar{z}_{\sigma(1)}\right) \cdots Y_V\left(u_{\sigma(n)}, z_{\sigma(n)}, \bar{z}_{\sigma(n)}\right) \, , \qquad (2.33)$$

is the expansion of $m_n(u_1, \ldots, u_n; z_1, \bar{z}_1, \ldots, z_n, \bar{z}_n)$ in the domain $\left|z_{\sigma(1)}\right| > \left|z_{\sigma(2)}\right| > \cdots > \left|z_{\sigma(n)}\right| > 0$. Here, $\bar{z}_{\sigma(1)}, ..., \bar{z}_{\sigma(n)}$ are complex conjugates of $z_{\sigma(1)}, ..., z_{\sigma(n)}$ respectively. If a function $m_n$ satisfying above properties exists, we say that the vertex operators $Y_V\left(u_1, z_1, \bar{z}_1\right), \ldots, Y_V\left(u_n, z_n, \bar{z}_n\right)$ are mutually *local* with respect to each other.

We will often denote the non-chiral VOA by $(V, Y_V)$ or simply by $V$.

**Remark 2.1.** For a homogeneous vector $u \in V_{(h,\bar{h})}$, the sum in (2.17) and (2.18) runs only over the set $\{(m, n) \in \mathbb{R}^2 \,|\, m - n \in \mathbb{Z}\}$. To see this, first note that the $L(0)$-property 7 implies the commutator

$$
\begin{aligned}
[L(0), x_{m,n}(u)] &= -m x_{m,n}(u), \\
[\bar{L}(0), x_{m,n}(u)] &= -n x_{m,n}(u).
\end{aligned}
\qquad (2.34)
$$

Equivalently,

$$
\begin{aligned}
[L(0), u_{m,n}] &= (h - m - 1) u_{m,n}, \\
[\bar{L}(0), u_{m,n}] &= (\bar{h} - n - 1) u_{m,n}.
\end{aligned}
\qquad (2.35)
$$

This implies that

$$
\begin{aligned}
\text{wt } x_{m,n}(u) &= -m, \quad \overline{\text{wt}} \, x_{m,n}(u) = -n, \\
\text{wt } u_{m,n} &= h - m - 1, \quad \overline{\text{wt}} \, u_{m,n} = \bar{h} - n - 1.
\end{aligned}
\qquad (2.36)
$$

The single-valuedness property 3 implies that $m - n \in \mathbb{Z}$ in both the sums. We will thus write the expansions of the vertex operators as

$$
\begin{aligned}
Y_V(u, x, \bar{x}) &= \sum_{\substack{m,n\in\mathbb{R} \\ (m-n)\in\mathbb{Z}}} u_{m,n} x^{-m-1} \bar{x}^{-n-1} \in \text{End}(V)\{x^{\pm 1}, \bar{x}^{\pm 1}\} \\
&= \sum_{\substack{m,n\in\mathbb{R} \\ (m-n)\in\mathbb{Z}}} x_{m,n}(u) x^{-m-h} \bar{x}^{-n-\bar{h}}.
\end{aligned}
\qquad (2.37)
$$

---

[8]We thank Yi-Zhi Huang for discussions on this point.
[9]The matrix elements of this operator are called *correlation functions* in Physics.

**Remark 2.2.** The single-valuedness property 3 implies that the vertex operators (2.15) is single-valued. To prove this, we must show that

$$Y_V(u, z, \bar{z}) = Y_V(u, e^{2\pi i} z, e^{-2\pi i} \bar{z}). \tag{2.38}$$

From Remark 2.1, we have

$$Y_V(u, e^{2\pi i} z, e^{-2\pi i} \bar{z}) = \sum_{\substack{m,n \in \mathbb{R} \\ (m-n) \in \mathbb{Z}}} u_{m,n} z^{-m-1} \bar{z}^{-n-1} e^{2\pi i(-m+n)} \tag{2.39}$$

$$= Y_V(u, z, \bar{z}).$$

**Remark 2.3.** For $v \in V_{(h,\bar{h})}$, if in the expansion (2.18) the index runs over $m \in \mathbb{Z} - h$, $n \in \mathbb{Z} - \bar{h}$, then assuming that $z, \bar{z}$ are independent complex variables, we can use Cauchy's residue theorem to write

$$x_{r,s}(v) = \frac{1}{(2\pi i)^2} \oint \oint dz d\bar{z} \, Y_V(v, z, \bar{z}) z^{r+h-1} \bar{z}^{s+\bar{h}-1} \,, \tag{2.40}$$

where the contour of the integration is a circle around $z = 0$ and $\bar{z} = 0$ respectively.

**Remark 2.4.** The creation property implies also the *injectivity* condition, i.e.

$$Y_V(v, x, \bar{x}) = 0 \text{ implies } v = 0, \text{ for } v \in V. \tag{2.41}$$

## 2.3   Some consequences of the definition

We now prove some consequences of the definition.

**Lemma 2.1.** *For any $v \in V$, we have*

$$Y_V(v, x, \bar{x})\mathbf{1} = e^{\bar{x}\,\bar{L}(-1)} e^{x\,L(-1)} v. \tag{2.42}$$

*Proof.* We use the translation property 8 and Taylor's theorem (2.11). For another formal variable $x_0, \bar{x}_0$, Taylor's theorem gives

$$Y_V\left(e^{xL(-1)} e^{\bar{x}\bar{L}(-1)} v, x_0, \bar{x}_0\right) = e^{x\frac{d}{dx}} e^{\bar{x}\frac{d}{d\bar{x}}} Y_V(v, x_0, \bar{x}_0) = Y_V(v, x + x_0, \bar{x} + \bar{x}_0). \tag{2.43}$$

Now applying this operator on $\mathbf{1}$, taking limit $x_0, \bar{x}_0 \to 0$ we get

$$\lim_{x_0, \bar{x}_0 \to 0} Y_V\left(e^{xL(-1)} e^{\bar{x}\bar{L}(-1)} v, x_0, \bar{x}_0\right) \mathbf{1} = \lim_{x_0, \bar{x}_0 \to 0} Y_V(v, x + x_0, \bar{x} + \bar{x}_0)\mathbf{1}, \tag{2.44}$$

and then using (2.25) we obtain (2.42). $\square$

**Lemma 2.2.** *For any $v \in V$ we have*

$$e^{x_2 L(-1)} e^{\bar{x}_2 \bar{L}(-1)} Y_V(v, x_1, \bar{x}_1) e^{-x_2 L(-1)} e^{-\bar{x}_2 \bar{L}(-1)} = Y_V(v, x_1 + x_2, \bar{x}_1 + \bar{x}_2). \tag{2.45}$$

*Proof.* Using the BCH formula (3.107) and translation property 8, we have

$$e^{x_2 L(-1)} Y_V(v, x_1, \bar{x}_1) e^{-x_2 L(-1)} = \sum_{n=0}^{\infty} \frac{[(x_2 L(-1))^n, Y_V(v, x_1, \bar{x}_1)]}{n!},$$

$$= \sum_{n=0}^{\infty} \frac{1}{n!} x_2^n \frac{\partial^n}{\partial x_1^n} Y_V(v, x_1, \bar{x}_1), \qquad (2.46)$$

$$= e^{x_2 \frac{\partial}{\partial x_1}} Y_V(v, x_1, \bar{x}_1),$$

$$= Y_V(v, x_1 + x_2, \bar{x}_1),$$

where in the last step we used Taylor's theorem (2.11). Similarly we have

$$e^{\bar{x}_2 \bar{L}(-1)} Y_V(v, x_1, \bar{x}_1) e^{-\bar{x}_2 \bar{L}(-1)} = Y_V(v, x_1, \bar{x}_1 + \bar{x}_2). \qquad (2.47)$$

Since $L(-1)$ and $\bar{L}(-1)$ commute, the result follows. $\qquad \square$

We now prove *skew-symmetry* which will be useful in proving the *duality* of vertex operators.

**Lemma 2.3.** *For any $u, v \in V$, we have*

$$Y_V(u, z, \bar{z})v = e^{z L(-1)} e^{\bar{z} \bar{L}(-1)} Y_V(v, -z, -\bar{z})u. \qquad (2.48)$$

*Proof.* Using Lemma locality property 9, 2.1 and Lemma 2.2, we have

$$Y_V(u, z, \bar{z}) Y_V(v, z', \bar{z}')\mathbf{1} \sim Y_V(v, z', \bar{z}') Y_V(u, z, \bar{z})\mathbf{1}$$

$$= Y_V(v, z', \bar{z}') e^{\bar{z} \bar{L}(-1)} e^{z L(-1)} u \qquad (2.49)$$

$$= e^{\bar{z} \bar{L}(-1)} e^{z L(-1)} Y_V(v, z' - z, \bar{z}' - \bar{z})u .$$

Now taking $z', \bar{z}' \to 0$ and using the creation property, we obtain the required result.

$\qquad \square$

The following proposition shows the uniqueness of vertex operators. The proof is on the lines of [3].

**Proposition 2.1.** *Let $U : V \longrightarrow V\{x, \bar{x}\}$ be a linear operator which is local with respect to every other vertex operator, in the sense of Property 9, and satisfies*

$$U(x, \bar{x})\mathbf{1} = e^{\bar{x} \bar{L}(-1)} e^{x L(-1)} v, \qquad (2.50)$$

*for some $v \in V$, then*

$$U(z, \bar{z}) = Y_V(v, z, \bar{z}), \qquad (2.51)$$

*for a non-zero complex number $z$.*

*Proof.* For any $w \in V$, from Lemma 2.1 and locality property 9 we have

$$
\begin{aligned}
U(z_1, \bar{z}_1)e^{\bar{z}_2 \bar{L}(-1)}e^{z_2 L(-1)}w &= U(z_1, \bar{z}_1)Y_V(w, z_2, \bar{z}_2)\mathbf{1}\,, \\
&\sim Y_V(w, z_2, \bar{z}_2)U(z_1, \bar{z}_1)\mathbf{1}\,, \\
&= Y_V(w, z_2, \bar{z}_2)e^{\bar{z}_1 \bar{L}(-1)}e^{z_1 L(-1)}v\,, \qquad (2.52) \\
&= Y_V(w, z_2, \bar{z}_2)Y_V(v, z_1, \bar{z}_1)\mathbf{1}\,, \\
&\sim Y_V(v, z_1, \bar{z}_1)Y_V(w, z_2, \bar{z}_2)\mathbf{1}\,,
\end{aligned}
$$

where $\sim$ indicates equality up to analytic extension in the sense of Property 9. Now taking $z_2, \bar{z}_2 \to 0$ we obtain,

$$
U(z_1, \bar{z}_1)w = Y_V(v, z_1, \bar{z}_1)w. \qquad (2.53)
$$

As the two operators in (2.53) are equal for all $w \in V$, they are equal as operators. $\quad\square$

We now prove the *duality* of vertex operators.

**Proposition 2.2.** *For any $v, w \in V$ we have*

$$
Y_V(v, z_1, \bar{z}_1)Y_V(w, z_2, \bar{z}_2) = Y_V(Y_V(v, z_1 - z_2, \bar{z}_1 - \bar{z}_2)w, z_2, \bar{z}_2)\,, \qquad (2.54)
$$

*in the domain $|z_1| > |z_2| > |z_1 - z_2| > 0$, where the RHS is defined by*

$$
Y_V(Y_V(v, z_1-z_2, \bar{z}_1-\bar{z}_2)w, z_2, \bar{z}_2) = \sum_{\substack{m,n\in\mathbb{R} \\ (m-n)\in\mathbb{Z}}} Y_V(v_{m,n}\cdot w, z_2, \bar{z}_2)(z_1-z_2)^{-m-1}(\bar{z}_1-\bar{z}_2)^{-n-1}\,.
$$

$$(2.55)$$

*Proof.* The proof is on the lines of [30, Page 23]. For any $u \in V$, we have

$$
\begin{aligned}
Y_V(v, z_1, \bar{z}_1)&Y_V(w, z_2, \bar{z}_2)e^{\bar{z}_3 \bar{L}(-1)}e^{z_3 L(-1)}u \\
&= Y_V(v, z_1, \bar{z}_1)Y_V(w, z_2, \bar{z}_2)Y_V(u, z_3, \bar{z}_3)\mathbf{1}\,, \\
&\sim Y_V(u, z_3, \bar{z}_3)Y_V(v, z_1, \bar{z}_1)Y_V(w, z_2, \bar{z}_2)\mathbf{1}\,, \\
&= Y_V(u, z_3, \bar{z}_3)Y_V(v, z_1, \bar{z}_1)e^{\bar{z}_2 \bar{L}(-1)}e^{z_2 L(-1)}w\,, \qquad (2.56) \\
&= Y_V(u, z_3, \bar{z}_3)e^{\bar{z}_2 \bar{L}(-1)}e^{z_2 L(-1)}Y_V(v, z_1 - z_2, \bar{z}_1 - \bar{z}_2)w\,, \\
&= Y_V(u, z_3, \bar{z}_3)Y_V\left(Y_V(v, z_1 - z_2, \bar{z}_1 - \bar{z}_2)w, z_2, \bar{z}_2\right)\mathbf{1}\,, \\
&\sim Y_V\left(Y_V(v, z_1 - z_2, \bar{z}_1 - \bar{z}_2)w, z_2, \bar{z}_2\right)Y_V(u, z_3, \bar{z}_3)\mathbf{1}\,,
\end{aligned}
$$

where we used Lemma 2.1, Lemma 2.2, and Locality property 9. Now taking $z_3, \bar{z}_3 \to 0$ and using Proposition 2.1, we obtain the duality relation. Note that the sum on the RHS of (2.55) converges. Indeed for any $u \in V$, using skew-symmetry [10]

---

[10] We thank Yi-Zhi Huang for clarification on this point.

(Lemma 2.3) we have

$$
\begin{aligned}
Y_V(Y_V(v,&z_1 - z_2, \bar{z}_1 - \bar{z}_2)w, z_2, \bar{z}_2)u \\
&= \sum_{\substack{m,n\in\mathbb{R} \\ (m-n)\in\mathbb{Z}}} Y_V(v_{m,n} \cdot w, z_2, \bar{z}_2)(z_1 - z_2)^{-m-1}(\bar{z}_1 - \bar{z}_2)^{-n-1}u \\
&= \sum_{\substack{m,n\in\mathbb{R} \\ (m-n)\in\mathbb{Z}}} \mathrm{e}^{\bar{z}_2 \bar{L}(-1)}\mathrm{e}^{z_2 L(-1)}Y_V(u, -z_2, -\bar{z}_2)v_{m,n} \cdot w(z_1 - z_2)^{-m-1}(\bar{z}_1 - \bar{z}_2)^{-n-1} \\
&= \mathrm{e}^{\bar{z}_2 \bar{L}(-1)}\mathrm{e}^{z_2 L(-1)}Y_V(u, -z_2, -\bar{z}_2) \sum_{\substack{m,n\in\mathbb{R} \\ (m-n)\in\mathbb{Z}}} v_{m,n} \cdot w(z_1 - z_2)^{-m-1}(\bar{z}_1 - \bar{z}_2)^{-n-1} \\
&= \mathrm{e}^{\bar{z}_2 \bar{L}(-1)}\mathrm{e}^{z_2 L(-1)}Y_V(v, -z_2, -\bar{z}_2)Y_V(u, z_1 - z_2, \bar{z}_1 - \bar{z}_2)w \ .
\end{aligned}
\tag{2.57}
$$

Since the RHS of the last line is well defined in $|z_2| > |z_1 - z_2|$, the operator $Y_V(Y_V(v, z_1 - z_2, \bar{z}_1 - \bar{z}_2)w, z_2, \bar{z}_2)$ is well defined in $|z_2| > |z_1 - z_2|$. $\qquad\square$

Proposition 2.2 shows that a product of two vertex operators can be written as a sum of single vertex operator:

$$
Y_V(v, z_1, \bar{z}_1)Y_V(w, z_2, \bar{z}_2) = \sum_{\substack{m,n\in\mathbb{R} \\ (m-n)\in\mathbb{Z}}} Y_V(v_{m,n} \cdot w, z_2, \bar{z}_2)(z_1 - z_2)^{-m-1}(\bar{z}_1 - \bar{z}_2)^{-n-1}.
\tag{2.58}
$$

In physics, we usually ignore the non-singular terms in the expansion above and call it the *operator product expansion*.

**Remark 2.5.** The sum in the operator product expansion has finitely many terms with negative powers of $(z_1 - z_2)$ and $(\bar{z}_1 - \bar{z}_2)$. To see this, we first expand the vertex operator $Y_V(v, x, \bar{x})$ for $v \in V_{(h,\bar{h})}$ as

$$
Y_V(v, x, \bar{x}) = \sum_{\substack{m,n\in\mathbb{R} \\ (m-n)\in\mathbb{Z}}} x_{m,n}(v)x^{-m-h}\bar{x}^{-n-\bar{h}},
\tag{2.59}
$$

Since

$$
\mathrm{wt}\, x_{m,n}(v) = -m, \quad \overline{\mathrm{wt}}\, x_{m,n}(v) = -n,
\tag{2.60}
$$

for $w \in V_{(h',\bar{h}')}$ we have

$$
x_{m,n}(v) \cdot w \in V_{(h'-m,\bar{h}'-n)}.
\tag{2.61}
$$

Due to the grading-restriction property 2, there exists $M \in \mathbb{Z}$ such that

$$
x_{m,n}(v) \cdot w = 0, \quad m, n > M.
\tag{2.62}
$$

Thus the operator product expansion is upper truncated.

**Proposition 2.3.** *The operator product expansion of the conformal vertex operator $T(x)$ with itself is given by*

$$T(x_1)T(x_2) = \frac{c}{2}\frac{1}{(x_1-x_2)^4} + \frac{2\,T(x_2)}{(x_1-x_2)^2} + \frac{1}{(x_1-x_2)}\frac{\partial}{\partial x_2}T(x_2) + G_1(x_1,x_2)$$

$$\bar{T}(\bar{x}_1)\bar{T}(\bar{x}_2) = \frac{\bar{c}}{2}\frac{1}{(\bar{x}_1-\bar{x}_2)^4} + \frac{2\,\bar{T}(\bar{x}_2)}{(\bar{x}_1-\bar{x}_2)^2} + \frac{1}{(\bar{x}_1-\bar{x}_2)}\frac{\partial}{\partial \bar{x}_2}\bar{T}(\bar{x}_2) + G_2(\bar{x}_1,\bar{x}_2) \tag{2.63}$$

$$T(x_1)\bar{T}(\bar{x}_2) = G_3(x_1,\bar{x}_2),$$

*where* $G_1(x_1,x_2), G_3(x_1,x_2) \in \mathrm{End}(V)[[x_2^{\pm 1}, (x_1-x_2)]], G_2(\bar{x}_1,\bar{x}_2) \in \mathrm{End}(V)[[\bar{x}_2^{\pm 1}, (\bar{x}_1-\bar{x}_2)]]$.

*Proof.* The proof is straightforward using the Virasoro algebra (2.27), see [31, Chapter 3] for more details. □

Of particular interest are the *chiral* and *anti-chiral* vertex operators.

**Definition 2.2.** A vector $u \in V$ is called a *chiral* (*anti-chiral*) vector if the corresponding vertex operator $Y_V(u,x,\bar{x})$ belongs in $\mathrm{End}(V)\{x^{\pm 1}\}$ ($\mathrm{End}(V)\{\bar{x}^{\pm 1}\}$) or equivalently only depends on $z$ ($\bar{z}$). Such vertex operators will be called chiral (anti-chiral) vertex operators.

**Remark 2.6.** From the translation property 8 we see that the vertex operator corresponding to $v$ is chiral if and only if $\bar{L}(-1)v = 0$ and anti-chiral if and only if $L(-1)v = 0$. The algebra of the modes of chiral (anti-chiral) vertex operators is called the *chiral (anti-chiral) algebra* in physics, see Corollary 2.10.

**Remark 2.7.** In the locality property 9 involving a chiral (resp. anti-chiral) vertex operator $Y_V(u_1,z_1)$(resp. $Y_V(u_1,\bar{z}_1)$) and another vertex operator $Y_V(u_2,z_2,\bar{z}_2)$, we will often denote the function $m$ by

$$R(Y_V(u_1,z_1)Y_V(u_2,z_2,\bar{z}_2)) \quad (\text{resp. } R(Y_V(u_1,\bar{z}_1)Y_V(u_2,z_2,\bar{z}_2))) \tag{2.64}$$

so that

$$R(Y_V(u_1,z_1)Y_V(u_2,z_2,\bar{z}_2)) = \begin{cases} Y_V(u_1,z_1)\,Y_V(u_2,z_2,\bar{z}_2) & \text{for } |z_1| > |z_2|, \\ Y_V(u_2,z_2,\bar{z}_2)\,Y_V(u_1,z_1) & \text{for } |z_2| > |z_1| \end{cases} \tag{2.65}$$

and

$$R(Y_V(u_1,\bar{z}_1)Y_V(u_2,z_2,\bar{z}_2)) = \begin{cases} Y_V(u_1,\bar{z}_1)\,Y_V(u_2,z_2,\bar{z}_2) & \text{for } |z_1| > |z_2|, \\ Y_V(u_2,z_2,\bar{z}_2)\,Y_V(u_1,\bar{z}_1) & \text{for } |z_2| > |z_1| \end{cases} \tag{2.66}$$

respectively. In physics, this is called *radial ordering*. Here $z_2, \bar{z}_2$ are complex conjugates of each other.

**Lemma 2.4.** *Let $u, v$ be homogeneous chiral and anti-chiral vector. Then the associated chiral and anti-chiral vertex operator has an expansion of the form*

$$Y_V(u, x) = \sum_{n \in \mathbb{Z}} x_n(u) x^{-n-(\text{wt}\, u - \overline{\text{wt}}\, u)} \in \text{End}(V)[[x^{\pm 1}]]\,,$$

$$Y_V(v, \bar{x}) = \sum_{n \in \mathbb{Z}} \bar{x}_n(v) \bar{x}^{-n-(\overline{\text{wt}}\, v - \text{wt}\, v)} \in \text{End}(V)[[\bar{x}^{\pm 1}]]\,, \tag{2.67}$$

*where*

$$x_n(u) := x_{n-\overline{\text{wt}}\, u, -\overline{\text{wt}}\, u}(u), \quad \bar{x}_n(v) := x_{-\text{wt}\, v, n-\text{wt}\, v}(v). \tag{2.68}$$

*Proof.* From the expansion (2.37), we see that $Y_V(u, x, \bar{x})$ will be independent of $\bar{x}$ if and only if

$$x_{m,n}(u) = 0 \text{ unless } n = -\overline{\text{wt}}\, u\,. \tag{2.69}$$

But as $m - n \in \mathbb{Z}$ we then have

$$x_{m,n}(u) = 0 \text{ unless } n = -\overline{\text{wt}}\, u,\ m \in \mathbb{Z} - \overline{\text{wt}}\, u. \tag{2.70}$$

This gives us the required expansion. The proof for anti-chiral vector $v$ is similar. The fact that $Y_V(u, x) \in \text{End}(V)[[x^{\pm 1}]], Y_V(v, \bar{x}) \in \text{End}(V)[[\bar{x}^{\pm 1}]]$ follows from the single-valuedness property 3. $\qquad \square$

**Remark 2.8.** By the above lemma, for chiral and anti-chiral vertex operators, the requirements in Remark 2.3 is satisfied and hence we can write

$$x_n(u) = \frac{1}{2\pi i} \oint dz\ Y_V(u, z) z^{n+(\text{wt}\, u - \overline{\text{wt}}\, u)-1}\,,$$

$$\bar{x}_n(v) = \frac{1}{2\pi i} \oint d\bar{z}\ Y_V(v, \bar{z}) \bar{z}^{n+(\overline{\text{wt}}\, v - \text{wt}\, v)-1}\,, \tag{2.71}$$

where $u, v$ are chiral and anti-chiral vectors respectively and the contour of integration is a circle around $z = 0, \bar{z} = 0$ respectively.

We now derive the commutator of the modes of two vertex operators and the *Borcherd's identity* [1].

**Theorem 2.1.** *Let $u_i \in V_{(h_i, \bar{h}_i)}$ and $u_j \in V_{(h_j, h_j')}$ ($v_i \in V_{(h_i', \bar{h}_i')}$ and $v_j \in V_{(h_j', \bar{h}_j')}$) be homogeneous chiral (resp. anti-chiral) vectors with corresponding vertex operators*

$$Y_V(u_i, x) = \sum_{n \in \mathbb{Z}} x_n(u_i) x^{-n-(h_i - \bar{h}_i)}, \qquad Y_V(u_j, x) = \sum_{n \in \mathbb{Z}} x_n(u_j) x^{-n-(h_j - \bar{h}_j)},$$

$$Y_V(v_i, \bar{x}) = \sum_{n \in \mathbb{Z}} \bar{x}_n(v_i) \bar{x}^{-n-(\bar{h}_i' - h_i')}, \qquad Y_V(v_j, \bar{x}) = \sum_{n \in \mathbb{Z}} \bar{x}_n(v_j) \bar{x}^{-n-(\bar{h}_j' - h_j')}. \tag{2.72}$$

Then the vectors $x_p(u_i) \cdot u_j$ and $\bar{x}_p(v_k) \cdot v_\ell$ are chiral and anti-chiral vectors respectively.
Further, we have

$$[x_n(u_i), x_k(u_j)] = \sum_{p \geq -(h_i - \bar{h}_i) + 1} \binom{n + (h_i - \bar{h}_i) - 1}{p + (h_i - \bar{h}_i) - 1} x_{k+n}(x_p(u_i) \cdot u_j),$$

$$[\bar{x}_n(v_i), \bar{x}_k(v_j)] = \sum_{p \geq -(\bar{h}'_i - h'_i) + 1} \binom{n + (\bar{h}'_i - h'_i) - 1}{p + (\bar{h}'_i - h'_i) - 1} \bar{x}_{k+n}(\bar{x}_p(v_i) \cdot v_j), \tag{2.73}$$

$$[x_n(u_i), \bar{x}_k(v_j)] = 0.$$

In particular,

$$[L(n), x_k(u_i)] = \sum_{p \geq -1} \binom{n+1}{p+1} x_{k+n}(L(p) \cdot u_i),$$

$$[L(n), \bar{x}_k(v_i)] = 0,$$

$$[\bar{L}(n), \bar{x}_k(v_i)] = \sum_{p \geq -1} \binom{n+1}{p+1} \bar{x}_{k+n}(\bar{L}(p) \cdot v_i), \tag{2.74}$$

$$[\bar{L}(n), x_k(u_i)] = 0.$$

More generally, for $m \in \mathbb{Z}$ and $m_+ \in \mathbb{Z}_{\geq 0}$ we have the Borcherd's identity:

$$\sum_{r \geq 0} \binom{m}{r} \left( (-1)^r x_{n+m-r}(u_i) x_{k+r}(u_j) - (-1)^{m+r} x_{k+m-r}(u_j) x_{n+r}(u_i) \right)$$

$$= \sum_{p \geq 1 - (h_i - \bar{h}_i)} \binom{n + (h_i - \bar{h}_i) - 1}{p + (h_i - \bar{h}_i) - 1} x_{k+n+m+\bar{h}_i - \bar{h}_j}(x_{p+m}(u_i) \cdot u_j), \tag{2.75}$$

$$\sum_{r \geq 0} \binom{m}{r} \left( (-1)^r \bar{x}_{n+m-r}(v_i) \bar{x}_{k+r}(v_j) - (-1)^{m+r} \bar{x}_{k+m-r}(v_j) \bar{x}_{n+r}(v_i) \right)$$

$$= \sum_{p \geq 1 - (\bar{h}'_i - h'_i)} \binom{n + (\bar{h}'_i - h'_i) - 1}{p + (\bar{h}'_i - h'_i) - 1} \bar{x}_{k+n+h'_i - h'_j}(\bar{x}_{p+m}(v_i) \cdot v_j), \tag{2.76}$$

$$\sum_{r \geq 0} \binom{m_+}{r} \left( (-1)^r x_{n+m_+-r}(u_i) \bar{x}_{k+r}(v_j) - (-1)^{m_++r} \bar{x}_{k+m_+-r}(v_j) x_{n+r}(u_i) \right) = 0. \tag{2.77}$$

*Proof.* We first show that $x_p(u_i) \cdot u_j$ and $\bar{x}_p(v_k) \cdot v_\ell$ are chiral and anti-chiral vectors respectively. Indeed by the translation property 8

$$[\bar{L}(-1), x_p(u_i)] = 0, \tag{2.78}$$

which implies that

$$\bar{L}(-1) \cdot (x_p(u_i) \cdot u_j) = x_p(u_i) \cdot \bar{L}(-1) u_j = 0. \tag{2.79}$$

Similarly

$$L(-1) \cdot (\bar{x}_p(v_k) \cdot v_\ell) = 0 \,. \tag{2.80}$$

Now, we will follow the usual contour integration procedure, see for example [31, Section 3.3.10]. First note that

$$Y_V(u_i, z_1)Y_V(u_j, z_2), Y_V(u_j, z_2)Y_V(u_i, z_1), R(Y_V(u_i, z_1)Y_V(u_j, z_2)) \,, \tag{2.81}$$

are single-valued and analytic in $z_1, z_2$ since their partial derivative with respect to $\bar{z}_1, \bar{z}_2$ is zero. So we can use Cauchy's residue theorem to integrate over $z_1, z_2$ on any contour. Now let $r_1 > r_2 > r_3 > 0$ be real numbers. Let $C_i^a(z)$ denote a contour in the variable $z_i$, in counterclockwise direction, of radius $a$ and centered around $z$. Further, $C_i^r := C_i^r(0)$. Let $f(z_1, z_2)$ be a rational function analytic in $z_1, z_2$ with poles only at $z_1 = 0, z_2 = 0, z_1 = z_2$. The integrals

$$\oint_{C_2^{r_2}} dz_2 \oint_{C_1^{r_1}} dz_1 \, Y_V(u_i, z_1)Y_V(u_j, z_2)f(z_1, z_2) \quad \text{and}$$

$$\oint_{C_2^{r_2}} dz_2 \oint_{C_1^{r_3}} dz_1 \, Y_V(u_j, z_2)Y_V(u_i, z_1)f(z_1, z_2) \,, \tag{2.82}$$

are well-defined. By the locality property 9 and the OPE (2.55), we see that

$$\oint_{C_2^{r_2}} dz_2 \oint_{C_1^{r_1}} dz_1 \, Y_V(u_i, z_1)Y_V(u_j, z_2)f(z_1, z_2)$$

$$- \oint_{C_2^{r_2}} dz_2 \oint_{C_1^{r_3}} dz_1 \, Y_V(u_j, z_2)Y_V(u_i, z_1)f(z_1, z_2)$$

$$= \oint_{C_2^{r_2}} dz_2 \oint_{C_1^{r_1}-C_1^{r_3}} dz_1 \, R(Y_V(u_i, z_1)Y_V(u_j, z_2))f(z_1, z_2) \tag{2.83}$$

$$= \oint_{C_2^{r_2}} dz_2 \oint_{C_1^{\delta}(z_2)} dz_1 \, Y_V(Y_V(u_i, z_1 - z_2)u_j, z_2)f(z_1, z_2)$$

$$= \oint_{C_2^{r_2}} dz_2 \oint_{C_1^{\delta}(z_2)} dz_1 \, \sum_{p \in \mathbb{Z}} Y_V(x_p(u_i) \cdot u_j, z_2)(z_1 - z_2)^{-p-(h_i - \bar{h}_i)}f(z_1, z_2) \,,$$

where $\delta$ is some small real number, see [31, Section 3.3.10] for details of the change in contour. If we now choose

$$f = z_1^{n+(h_i-\bar{h}_i)-1} z_2^{k+(h_j-\bar{h}_j)-1} \,, \tag{2.84}$$

then using (2.71) the LHS is[11] $[x_n(u_i), x_k(u_j)]$ while Cauchy's residue theorem gives the RHS to be

$$\oint_{C_2^{r_2}} dz_2 \sum_{p \geq -(h_i-\bar{h}_i)+1} \binom{n+(h_i-\bar{h}_i)-1}{p+(h_i-\bar{h}_i)-1} Y_V(x_p(u_i) \cdot u_j, z_2) z_2^{k+(h_j-\bar{h}_j)+n-p-1} \,, \tag{2.85}$$

---

[11]There is a factor of $(2\pi i)^2$ which cancels on both sides, so we ignore it.

where we used the identity

$$\oint_{C_1^\delta(z_2)} dz_1 \, \frac{z_1^{n+(h_i-\bar{h}_i)-1}}{(z_1-z_2)^{p+(h_i-\bar{h}_i)}} = \binom{n+(h_i-\bar{h}_i)-1}{p+(h_i-\bar{h}_i)-1} z_2^{n-p}. \tag{2.86}$$

Note, that it is necessary that $(h_i - \bar{h}_i) \in \mathbb{Z}$, which is true by the single valuedness property 3, for (2.86) to hold. Finally, using (2.71) and the fact that

$$x_p(u_i).u_j \in V_{h_j-p+\bar{h}_i, \bar{h}_i+\bar{h}_j}, \tag{2.87}$$

the RHS becomes

$$\sum_{p \geq -(h_i-\bar{h}_i)+1} \binom{n+(h_i-\bar{h}_i)-1}{p+(h_i-\bar{h}_i)-1} x_{k+n}(x_p(u_i) \cdot u_j). \tag{2.88}$$

The second commutator is similar. To prove the third commutator, note that since

$$\partial_{\bar{z}_1} R(Y_V(u_i, z_1) Y_V(v_j, \bar{z}_2)) = \partial_{z_2} R(Y_V(u_i, z_1) Y_V(v_j, \bar{z}_2)) = 0, \tag{2.89}$$

$R(Y_V(u_i, z_1) Y_V(v_j, \bar{z}_2))$ cannot have any dependence on $(z_1 - z_2)$ or $(\bar{z}_1 - \bar{z}_2)$. Moreover, from the proof of the OPE in Proposition 2.2, we see that it cannot also have $(z_1 - \bar{z}_2)$ dependence as well. This implies that the contour integral on the RHS of (2.83) vanishes and we get

$$[x_m(u_i), \bar{x}_n(v_j)] = 0. \tag{2.90}$$

The three Borcherd's identity follow by using

$$\begin{aligned}
f_1 &= z_1^{n+(h_i-\bar{h}_i)-1} z_2^{k+(h_j-\bar{h}_j)-1} (z_1-z_2)^m, \\
f_2 &= \bar{z}_1^{n+(\bar{h}_i'-h_i')-1} \bar{z}_2^{k+(\bar{h}_i'-h_i')-1} (\bar{z}_1-\bar{z}_2)^m, \\
f_3 &= z_1^{n+(h_i-\bar{h}_i)-1} \bar{z}_2^{k+(\bar{h}_j-h_j')-1} (z_1-\bar{z}_2)^m,
\end{aligned} \tag{2.91}$$

where for the second Borcherd's identity, we need to integrate against $d\bar{z}_1, d\bar{z}_2$ on the curves $C_{\bar{z}_1}^{r_1}, C_{\bar{z}_2}^{r_2}$ respectively and for the third Borcherd's identity, we need to integrate against $dz_1, d\bar{z}_2$ on the curves $C_{z_1}^{r_1}, C_{\bar{z}_2}^{r_2}$ respectively. □

**Remark 2.9.** For $n = 0, -1$ in (2.74) we obtain the $L(0)$-property 7 and the translation property 8 of chiral vertex operators. Note that we already used these properties in proving the OPE.

**Remark 2.10.** The commutator of the modes of chiral and anti-chiral vertex operators is closed. The algebra in (2.73) thus obtained is called the *chiral* and *anti-chiral algebra* respectively of the non-chiral VOA $(V, Y_V)$.

**Definition 2.3.** Let $(V, Y_V)$ be a non-chiral VOA with central charge $(c, \bar{c})$. The *graded dimension* or *character* of $V$ is defined by

$$\chi_V(\tau, \bar{\tau}) := \text{Tr}_V \, q^{L(0) - \frac{c}{24}} \bar{q}^{\bar{L}(0) - \frac{\bar{c}}{24}} = \sum_{(h, \bar{h}) \in \mathbb{R} \times \mathbb{R}} \left( \dim V_{(h, \bar{h})} \right) q^{h - \frac{c}{24}} \bar{q}^{\bar{h} - \frac{\bar{c}}{24}}, \qquad (2.92)$$

where $q = e^{2\pi i \tau}$, $\bar{q} = e^{-2\pi i \bar{\tau}}$ and $\tau \in \mathbb{H} := \{\tau = x + iy : y > 0\}$.

Note that the single-valuedness property implies that $\chi_V(\tau + 1, \bar{\tau} + 1) = \chi_V(\tau, \bar{\tau})$ if

$$c - \bar{c} = 24k, \qquad (2.93)$$

for some integer $k$.

Let $(V_1, Y_{V_1}, \omega_{V_1}, \bar{\omega}_{V_1}, \mathbf{1}_{V_1}), (V_2, Y_{V_2}, \omega_{V_2}, \bar{\omega}_{V_2}, \mathbf{1}_{V_2})$ be two non-chiral VOAs with the same central charge. Then a map $f : V_1 \to V_2$ is called a non-chiral VOA homomorphism if it is a grading-preserving linear map such that

$$f(Y_{V_1}(u, x, \bar{x})v) = Y_{V_2}(f(u), x, \bar{x}) f(v) \text{ for } u, v \in V_1, \qquad (2.94)$$

or equivalently,

$$f(u_{n,m} \cdot v) = f(u)_{n,m} f(v) \text{ for } u, v \in V_1, \quad n, m \in \mathbb{R}, \qquad (2.95)$$

and such that

$$f(\mathbf{1}_{V_1}) = \mathbf{1}_{V_2}, \quad f(\omega_{V_1}) = \omega_{V_2}, \quad f(\bar{\omega}_{V_1}) = \bar{\omega}_{V_2}. \qquad (2.96)$$

An isomorphism of non-chiral VOAs is a bijective homomorphism. An endomorphism of a non-chiral VOA $V$ is a homomorphism from $V$ to itself, and an automorphism of $V$ is a bijective endomorphism. In particular, an automorphism can be defined as a linear isomorphism $f : V \to V$ such that

$$f \circ Y_V(v, x, \bar{x}) \circ f^{-1} = Y_V(f(v), x, \bar{x}) \text{ for } v \in V,$$
$$f(\omega) = \omega, \quad f(\bar{\omega}) = \bar{\omega}. \qquad (2.97)$$

It follows that $f$ is grading-preserving and $f(\mathbf{1}_V) = \mathbf{1}_V$.

It is easy to see that the graded dimension of isomorphic non-chiral VOAs are identical.

## 3 Lorentzian Lattice Vertex Operator Algebra (LLVOA)

In this section, we will construct a non-chiral vertex operator algebra corresponding to an even, integral Lorentzian lattice $\Lambda \subset \mathbb{R}^{m,n}$. In the first subsection, we recall some basic facts about Lorentzian lattices and set up the notations for the rest of

the paper. We also record some results we will need later. In the next subsection, we gather the ingredients needed to construct a non-chiral vertex operator algebra, i.e. we will construct a vector space $V_\Lambda$ associated to the lattice, a vertex operator map $Y_{V_\Lambda}$ for this vector space, a vacuum $\mathbf{1}$, and conformal vectors $\omega_L, \omega_R$. In the last subsection, we will prove that $(V_\Lambda, Y_{V_\Lambda}, \omega_L, \omega_R, \mathbf{1})$ is a non-chiral VOA, which we will call the Lorentzian lattice vertex operator algebra (LLVOA).

## 3.1 Lorentzian lattices

We begin with some basic definitions. Let $\mathbb{R}^m$ be the Euclidean space equipped with a symmetric bilinear form $\langle \cdot, \cdot \rangle_m$. Let $\mathbb{R}^{m,n}$ denote the $(m+n)$-dimensional vector space $\mathbb{R}^{m+n}$ equipped with the symmetric bilinear form

$$\boldsymbol{x} \circ \boldsymbol{x}' := \langle \vec{x}, \vec{x}' \rangle_m - \langle \vec{y}, \vec{y}' \rangle_n \,, \tag{3.1}$$

where

$$\boldsymbol{x} = (x^1, \ldots, x^m, y^1, \ldots, y^n) \equiv (\vec{x}, \vec{y}) \,, \tag{3.2}$$

and similarly $\boldsymbol{x}'$. We will omit the subscript on $\langle \cdot, \cdot \rangle_m$ to make the notation lighter.

**Definition 3.1.** 1. A $d = (m+n)$-dimensional *Lorentzian lattice* of signature $(m,n)$ is a subset $\Lambda \subset \mathbb{R}^{m,n}$ which is also a free $\mathbb{Z}$-module spanned by $m+n$ vectors $\lambda_j \in \mathbb{R}^{m,n}, 1 \leq j \leq m+n$, linearly independent in $\mathbb{R}^{m,n}$. More explicitly

$$\Lambda = \left\{ \sum_{j=1}^{m+n} n_j \lambda_j : n_j \in \mathbb{Z} \right\} \,. \tag{3.3}$$

$\{\lambda_j\}_{j=1}^{m+n}$ is called an *integral basis* of $\Lambda$. When $n = 0$ we call $\Lambda$ a *Euclidean lattice*. We will simply refer them as lattices when we do not need to specify their signature.

2. The *dual lattice* of a lattice $\Lambda$, denoted by $\Lambda^\star$, is defined as

$$\Lambda^\star = \{ \boldsymbol{x}' \in \mathbb{R}^{m,n} : \boldsymbol{x} \circ \boldsymbol{x}' \in \mathbb{Z} \ \forall \ \boldsymbol{x} \in \Lambda \} \,. \tag{3.4}$$

The lattice $\Lambda$ is said to be *integral* if $\Lambda \subseteq \Lambda^\star$, i.e. $\boldsymbol{x} \circ \boldsymbol{y} \in \mathbb{Z}$ for all $\boldsymbol{x}, \boldsymbol{x}' \in \Lambda$ and *self-dual* if $\Lambda = \Lambda^\star$. The lattice $\Lambda$ is said to be *even* if

$$\boldsymbol{x} \circ \boldsymbol{x} = ||\vec{x}||^2 - ||\vec{y}||^2 \in 2\mathbb{Z} \,, \tag{3.5}$$

for all $\boldsymbol{x} = (\vec{x}, \vec{y}) \in \Lambda$, where $||\vec{x}||^2 := \langle \vec{x}, \vec{x} \rangle$.

3. A generator matrix for $\Lambda$ is an $(m+n) \times (m+n)$ matrix such that the $\mathbb{Z}$-span of its rows is $\Lambda$.

4. A *lattice homomorphism* of two lattices $f : \Lambda \longrightarrow \tilde{\Lambda}$ of the same signature is simply a $\mathbb{Z}$-module morphism which also preserves the bilinear form:

$$f(\boldsymbol{x}) \circ f(\boldsymbol{x}') = \boldsymbol{x} \circ \boldsymbol{x}', \quad \forall \ \boldsymbol{x}, \boldsymbol{x}' \in \Lambda. \tag{3.6}$$

A bijective lattice homomorphism is called a lattice isomorphism. Two lattices are said to be isomorphic if there exists a lattice isomorphism between them.

5. An *automorphism of the lattice* $\Lambda$ is a lattice isomorphism from the $\Lambda$ to itself. The group of all automorphisms (the group operation being composition) is called the automorphism group of $\Lambda$ and denoted by $\mathrm{Aut}(\Lambda)$.

A generator matrix for the lattice $\Lambda$ in (3.3) is given by

$$\mathcal{G}_\Lambda = \begin{pmatrix} \lambda_1^1 & \lambda_1^2 & \cdots & \lambda_1^{m+n} \\ \vdots & \vdots & \cdots & \vdots \\ \lambda_{m+n}^1 & \lambda_{m+n}^1 & \cdots & \lambda_{m+n}^{m+n} \end{pmatrix}, \tag{3.7}$$

where $\lambda_i = (\lambda_i^1, \dots, \lambda_i^{m+n})$ is a basis vector of $\Lambda$. It is not hard to show that two generator matrices $\mathcal{G}_\Lambda, \mathcal{G}'_\Lambda$ generate the same lattice if and only if they are related by an $(m+n) \times (m+n)$ *unimodular matrix*[12] $U \in \mathrm{GL}(m+n, \mathbb{Z})$:

$$\mathcal{G}_\Lambda = U \mathcal{G}'_\Lambda. \tag{3.8}$$

Indeed $U$ is the change of basis matrix between the primed and unprimed generator matrices since it is invertible and since it is also integral, it preserves the lattice. If we take the symmetric bilinear form $\langle \cdot, \cdot \rangle$ on $\mathbb{R}^m, \mathbb{R}^n$ to be the standard inner product, that is,

$$\langle \vec{x}, \vec{x}' \rangle = \sum_{i=1}^{m} x^i x'^i, \tag{3.9}$$

where $\vec{x} = (x^1, \dots, x^m) \in \mathbb{R}^m$ and similarly $\vec{x}'$ and analogous inner product on $\mathbb{R}^n$, then a lattice isomorphism between lattices of signature $(m, n)$ can be identified with an element of $\mathrm{O}(m, n, \mathbb{R})$ where $\mathrm{O}(m, n, \mathbb{R})$ is the group of matrices, $A$, satisfying

$$A^T g_{m,n} A = g_{m,n}, \quad g_{m,n} = \begin{pmatrix} \mathbb{1}_m & 0 \\ 0 & -\mathbb{1}_n \end{pmatrix}. \tag{3.10}$$

We have the following theorem:

**Theorem 3.1.** [32, Chapter V] *An even, self-dual lattice of signature $(m, n)$ exists if and only if $(m - n) \equiv 0 \bmod 8$. Moreover, there is a unique such lattice when $n \geq 1$ up to an $\mathrm{O}(m, n, \mathbb{R})$ transformation.*

---

[12]A matrix $U$ is called unimodular if $\det(U) = \pm 1$.

The canonical choice of an even, self-dual lattice of signature $\mathbb{R}^{m,n}$, denoted by $\text{II}_{m,n}$, is

$$\text{II}_{m,n} = \left\{ (a_1, \ldots, a_{m+n}) \in \mathbb{R}^{m,n} : a_i \in \mathbb{Z} \text{ or } a_i \in \mathbb{Z} + \frac{1}{2}, \ \sum_{i=1}^{m+n} a_i \in 2\mathbb{Z} \right\}. \quad (3.11)$$

A generator matrix for this lattice is

$$\mathcal{G}_{\text{II}_{m,n}} = \begin{pmatrix} 1 & 0 & \cdots & 0 & 0 & 0 & \cdots & 0 & 0 & -1 \\ 0 & 1 & \cdots & 0 & 0 & 0 & \cdots & 0 & 0 & -1 \\ \vdots & \vdots & \cdots & \vdots & \vdots & \vdots & \cdots & \vdots & \vdots & \vdots \\ 0 & 0 & \cdots & 1 & 0 & 0 & \cdots & 0 & 0 & -1 \\ \hline 0 & 0 & \cdots & 0 & 1 & 0 & \cdots & 0 & 0 & -1 \\ 0 & 0 & \cdots & 0 & 0 & 1 & \cdots & 0 & 0 & -1 \\ \vdots & \vdots & \cdots & \vdots & \vdots & \vdots & \cdots & \vdots & \vdots & \vdots \\ 0 & 0 & \cdots & 0 & 0 & 0 & \cdots & 1 & 0 & -1 \\ 0 & 0 & \cdots & 0 & 0 & 0 & \cdots & 0 & 0 & 2 \\ \frac{1}{2} & \frac{1}{2} & \cdots & \frac{1}{2} & \frac{1}{2} & \frac{1}{2} & \cdots & \frac{1}{2} & \frac{1}{2} & \frac{1}{2} \end{pmatrix}. \quad (3.12)$$

We will use this lattice to elucidate many of the notations which we now introduce. Consider a $d$-dimensional even, integral, Lorentzian lattice $\Lambda \subset \mathbb{R}^{m,n}$ with Lorentzian inner product, denoted as before by $\circ$, where $m+n = d$. We will often write a vector $\lambda \in \Lambda$ as $\lambda = (\alpha^\lambda, \beta^\lambda)$, where $\alpha^\lambda \in \mathbb{R}^m$ and $\beta^\lambda \in \mathbb{R}^n$. Then we can write

$$\lambda_1 \circ \lambda_2 = \langle \alpha^{\lambda_1}, \alpha^{\lambda_2} \rangle - \langle \beta^{\lambda_1}, \beta^{\lambda_2} \rangle \in \mathbb{Z}. \quad (3.13)$$

Note that in general $\langle \alpha^{\lambda_1}, \alpha^{\lambda_2} \rangle, \langle \beta^{\lambda_1}, \beta^{\lambda_2} \rangle \notin \mathbb{Z}$. We define the $\mathbb{Z}$-modules

$$\begin{aligned} \Lambda_1 &= \{ \alpha^\lambda \,|\, \lambda = (\alpha^\lambda, \beta^\lambda) \in \Lambda \text{ for some } \beta^\lambda \in \mathbb{R}^n \} \subset \mathbb{R}^m, \\ \Lambda_2 &= \{ \beta^\lambda \,|\, \lambda = (\alpha^\lambda, \beta^\lambda) \in \Lambda \text{ for some } \alpha^\lambda \in \mathbb{R}^m \} \subset \mathbb{R}^n. \end{aligned} \quad (3.14)$$

Let $\{ \lambda_i \equiv (\alpha^{\lambda_i}, \beta^{\lambda_i}) \}_{i=1}^d$ be a basis of $\Lambda$. Then it is easy to see that

$$\Lambda_1 = \text{Span}_{\mathbb{Z}} \{ \alpha^{\lambda_i} \}_{i=1}^d, \quad \Lambda_2 = \text{Span}_{\mathbb{Z}} \{ \beta^{\lambda_i} \}_{i=1}^d. \quad (3.15)$$

Note that in general $\Lambda_1$ and $\Lambda_2$ are not lattices, they are just finitely generated $\mathbb{Z}$ modules possibly with non-trivial torsion. For the lattice $\text{II}_{m,n}$ in (3.12), it is easy to see that

$$\begin{aligned} (\text{II}_{m,n})_1 &= \mathbb{Z}^m \bigcup \left( \mathbb{Z} + \frac{1}{2} \right)^m, \\ (\text{II}_{m,n})_2 &= \mathbb{Z}^n \bigcup \left( \mathbb{Z} + \frac{1}{2} \right)^n. \end{aligned} \quad (3.16)$$

We further identify even, integral, Euclidean sublattices of $\Lambda$ as follows:

$$
\begin{aligned}
\Lambda_1^0 &:= \{(\alpha, 0) \in \Lambda \mid \alpha \in \mathbb{R}^m\}, \\
\Lambda_2^0 &:= \{(0, \beta) \in \Lambda \mid \beta \in \mathbb{R}^n\}.
\end{aligned}
\tag{3.17}
$$

These can be identified naturally with submodules of $\Lambda_1$ and $\Lambda_2$ respectively. Clearly, $\Lambda_1^0, \Lambda_2^0$ are sublattices of $\Lambda$ since any submodule of a finitely generated free module is free. We also introduce the notation

$$
\Lambda_0 := \Lambda_1^0 \oplus \Lambda_2^0.
\tag{3.18}
$$

Note that the direct sum of $\Lambda_1^0$ and $\Lambda_2^0$ is meaningful as they are two $\mathbb{Z}$-modules. For the lattice $\mathrm{II}_{m,n}$ in (3.12), $(\mathrm{II}_{m,n})_0 = (\mathrm{II}_{m,n})_1^0 \oplus (\mathrm{II}_{m,n})_2^0$ is easily seen to be generated by

$$
\mathcal{G}_{(\mathrm{II}_{m,n})_0} := \begin{pmatrix} \mathcal{G}_m & 0_{m\times n} \\ 0_{n\times m} & \mathcal{G}_n, \end{pmatrix}
\tag{3.19}
$$

where

$$
\mathcal{G}_m := \begin{pmatrix} 1 & 0 & 0 & \cdots & 0 & -1 \\ 0 & 1 & 0 & \cdots & 0 & -1 \\ \vdots & \vdots & \vdots & \cdots & \vdots & \vdots \\ 0 & 0 & 0 & \cdots & 1 & -1 \\ 0 & 0 & 0 & \cdots & 0 & 2 \end{pmatrix}_{m\times m},
\tag{3.20}
$$

and $\mathcal{G}_n$ is defined similarly. It is useful to characterize the automorphisms of $\Lambda$. We will take the symmetric bilinear form on $\mathbb{R}^m, \mathbb{R}^n$ to be the standard inner product for brevity. We have the following important result.

**Theorem 3.2.** *Let $\Lambda \in \mathbb{R}^{m,n}$ be an integral Lorentzian lattice. Then $\mathrm{Aut}(\Lambda) \cong \mathrm{O}_\Lambda(m, n, \mathbb{Z})$ where*

$$
\mathrm{O}_\Lambda(m, n, \mathbb{Z}) := \{A \in \mathrm{GL}(m + n, \mathbb{Z}) : \mathcal{G}_\Lambda^{-1} A \mathcal{G}_\Lambda \in \mathrm{O}(m, n, \mathbb{R})\}.
\tag{3.21}
$$

*Proof.* Choose an integral basis $\{\lambda_i\}$ of $\Lambda$. Then the group of $\mathbb{Z}$-module automorphisms of $\Lambda$ can be identified with $\mathrm{GL}(m + n, \mathbb{Z})$. Now given any $\lambda, \lambda' \in \Lambda$ there exists coulumn vectors $\vec{n}, \vec{n}' \in \mathbb{Z}^{m+n}$ such that $\lambda = \vec{n}^T \mathcal{G}_\Lambda$ and $\lambda' = \vec{n}'^T \mathcal{G}_\Lambda$. Any module automorphism $A \in \mathrm{GL}(m + n, \mathbb{Z})$ acts by

$$
A(\lambda) = \vec{n}^T A \mathcal{G}_\Lambda.
\tag{3.22}
$$

For $A$ to preserve inner product, we must have

$$
\begin{aligned}
A(\lambda) \circ A(\lambda') &= \vec{n}^T A \mathcal{G}_\Lambda g_{m,n} \mathcal{G}_\Lambda^T A^T \vec{n}', \\
&= \vec{n}^T \mathcal{G}_\Lambda g_{m,n} \mathcal{G}_\Lambda^T \vec{n}',
\end{aligned}
\tag{3.23}
$$

which implies

$$
A \mathcal{G}_\Lambda g_{m,n} \mathcal{G}_\Lambda^T A^T = \mathcal{G}_\Lambda g_{m,n} \mathcal{G}_\Lambda^T.
\tag{3.24}
$$

Since $\{\lambda_i\}$ is a basis for $\mathbb{R}^{m,n}$ and $A$ must preserve the inner products of $\lambda_i$'s, we must have that $A = \mathcal{G}_\Lambda O \mathcal{G}_\Lambda^{-1}$ for some $O \in \mathrm{O}(m, n, \mathbb{R})$. $\qquad\square$

## 3.2 Construction of the LLVOA

Let $\Lambda \subset \mathbb{R}^{m,n}$ be a $d = (m+n)$-dimensional Lorentzian lattice. We denote by $\mathbb{C}[\Lambda]$ the group algebra of the lattice $\Lambda$ and denote the element $\lambda \in \Lambda$ embedded in $\mathbb{C}[\Lambda]$ by $e^\lambda$. The multiplication in $\mathbb{C}[\Lambda]$ is defined[13] by

$$e^{\lambda_1} \cdot e^{\lambda_2} = e^{\lambda_1 + \lambda_2} \,. \tag{3.25}$$

Define the vector space

$$\mathfrak{h}_i := \Lambda_i \otimes_{\mathbb{Z}} \mathbb{C}, \quad i = 1, 2 \,, \tag{3.26}$$

and extend the bilinear form on $\Lambda_i$ to $\mathfrak{h}_i$ $\mathbb{C}$-linearly. Here $\Lambda_i$ is defined as in (3.14). We define the Lie algebra

$$\hat{\mathfrak{h}} := \left( \bigoplus_{r,s \in \mathbb{Z}} (\mathfrak{h}_1 \otimes t^r) \oplus (\mathfrak{h}_2 \otimes \bar{t}^s) \right) \oplus (\mathbb{C}\mathbf{k} \oplus \mathbb{C}\bar{\mathbf{k}}). \tag{3.27}$$

Introduce the notation

$$\alpha(r) := \alpha \otimes t^r, \quad \beta(s) := \beta \otimes \bar{t}^s, \quad \alpha \in \mathfrak{h}_1, \beta \in \mathfrak{h}_2. \tag{3.28}$$

The non-zero Lie bracket on $\hat{\mathfrak{h}}$ is

$$\begin{aligned}
[\alpha(r_1), \alpha'(r_2)] &= r_1 \langle \alpha, \alpha' \rangle \, \delta_{r_1+r_2,0} \, \mathbf{k}, \\
[\beta(s_1), \beta'(s_2)] &= s_1 \langle \beta, \beta' \rangle \, \delta_{s_1+s_2,0} \, \bar{\mathbf{k}} \,.
\end{aligned} \tag{3.29}$$

Note that

$$\hat{\mathfrak{h}} = \hat{\mathfrak{h}}_1^\star \oplus \hat{\mathfrak{h}}_2^\star \oplus \hat{\mathfrak{h}}_1^0 \oplus \hat{\mathfrak{h}}_2^0 \,, \tag{3.30}$$

where $\hat{\mathfrak{h}}_1^\star, \hat{\mathfrak{h}}_2^\star$ are the standard Heisenberg algebras associated to the abelian Lie algebras $\mathfrak{h}_1, \mathfrak{h}_2$ respectively [2, Chapter 1] and

$$\hat{\mathfrak{h}}_1^0 := \mathfrak{h}_1 \otimes t^0 \cong \mathfrak{h}_1, \quad \hat{\mathfrak{h}}_2^0 := \mathfrak{h}_2 \otimes \bar{t}^0 \cong \mathfrak{h}_2. \tag{3.31}$$

Define

$$\hat{\mathfrak{h}}^- := \left( \bigoplus_{r,s < 0} (\mathfrak{h}_1 \otimes t^r) \oplus (\mathfrak{h}_2 \otimes \bar{t}^s) \right), \quad \hat{\mathfrak{h}}^0 := (\mathfrak{h}_1 \otimes t^0) \oplus (\mathfrak{h}_2 \otimes \bar{t}^0) \oplus \mathbb{C}\mathbf{k} \oplus \mathbb{C}\bar{\mathbf{k}} \,,$$

$$\hat{\mathfrak{h}}^+ := \left( \bigoplus_{r,s > 0} (\mathfrak{h}_1 \otimes t^r) \oplus (\mathfrak{h}_2 \otimes \bar{t}^s) \right). \tag{3.32}$$

Note that

$$\hat{\mathfrak{h}} = \hat{\mathfrak{h}}^- \oplus \hat{\mathfrak{h}}^0 \oplus \hat{\mathfrak{h}}^+ \,. \tag{3.33}$$

---

[13]Technically speaking, $\mathbb{C}[\Lambda]$ is the group algebra of the formal group $e^\Lambda := \{e^\lambda : \lambda \in \Lambda\}$ with group multiplication given by $e^{\lambda_1} \cdot e^{\lambda_2} = e^{\lambda_1+\lambda_2}$, i.e. $\mathbb{C}[\Lambda] = \mathbb{C}[e^\Lambda]$

We now define the space

$$V_\Lambda := S(\hat{\mathfrak{h}}^-) \otimes \mathbb{C}[\Lambda_1^0 \oplus \Lambda_2^0]\,, \tag{3.34}$$

where $\Lambda_1^0$ and $\Lambda_2^0$ is defined as in (3.17) and for any Lie algebra $\mathfrak{g}$, $S(\mathfrak{g})$ is the symmetric algebra for $\mathfrak{g}$, and $\mathbb{C}[\Lambda_1^0 \oplus \Lambda_2^0] \equiv \mathbb{C}[\Lambda_0]$ is considered as a subspace of $\mathbb{C}[\Lambda]$. The space $V_\Lambda$ is generated by elements of the form

$$(\alpha_1(-m_1) \cdot \alpha_2(-m_2) \cdots \alpha_k(-m_k) \cdot \beta_1(-\bar{m}_1) \cdot \beta_2(-\bar{m}_2) \cdots \beta_{\bar{k}}(-\bar{m}_{\bar{k}})) \otimes \mathrm{e}^{(\alpha,\beta)} \tag{3.35}$$

for $m_i$, $\bar{m}_i > 0$, $k, \bar{k} \geq 0$, $(\alpha,\beta) \in \Lambda_0$, $\alpha_i \in \mathfrak{h}_1$, and $\beta_i \in \mathfrak{h}_2$. The space $V_\Lambda$ is a natural module of $\hat{\mathfrak{h}}^-$. We define the action of $\hat{\mathfrak{h}}^0$ on $\mathbb{C}[\Lambda]$, and hence on $\mathbb{C}[\Lambda_0]$, by

$$\begin{aligned}
\alpha'(0)\, e^\lambda &= \langle \alpha', \alpha^\lambda \rangle\, e^\lambda\,, \\
\beta'(0)\, e^\lambda &= \langle \beta', \beta^\lambda \rangle\, e^\lambda\,,
\end{aligned} \tag{3.36}$$

where $\alpha'(0) \in \hat{\mathfrak{h}}_1^0$, $\beta'(0) \in \hat{\mathfrak{h}}_2^0$. The central elements $\mathbf{k}$ and $\bar{\mathbf{k}}$ act on $\mathbb{C}[\Lambda]$ as identity. Let $\hat{\mathfrak{h}}^+$ act on $\mathbb{C}[\Lambda]$ by 0. We can extend the action of these subspaces of $\hat{\mathfrak{h}}$ to $V_\Lambda$ by using the Lie bracket given in (3.29). This makes $V_\Lambda$ into an $\hat{\mathfrak{h}}$-module.

We define a $\mathbb{Z}$-bilinear map $\epsilon : \Lambda \times \Lambda \to \mathbb{Z}$, which acts on the basis as [33]

$$\epsilon\,(\lambda_i, \lambda_j) = \begin{cases} \lambda_i \circ \lambda_j & i > j\,, \\ 0 & i \leq j\,, \end{cases} \tag{3.37}$$

where $\{\lambda_i\}_{i=1}^{m+n}$ is an integral basis of $\Lambda$. The action of $\epsilon$ on general vectors is defined by the $\mathbb{Z}$-bilinearity of $\epsilon$. Consider $\hat{\Lambda} = \mathbb{Z}_2 \times \Lambda$, with the multiplication on it given by

$$(\theta, \lambda) \cdot (\tau, \lambda') = \left(\theta\tau\,(-1)^{\epsilon(\lambda,\lambda')}, \lambda + \lambda'\right)\,. \tag{3.38}$$

We now consider the $\mathbb{Z}_2$ central extension of the lattice $\Lambda$:

$$0 \longrightarrow \mathbb{Z}_2 \longrightarrow \hat{\Lambda} \longrightarrow \Lambda \longrightarrow 0\,. \tag{3.39}$$

We denote elements $(1, \lambda), (\theta, 0) \in \hat{\Lambda}$ by $\mathrm{e}_\lambda = (1, \lambda)$ and $\theta = (\theta, 0)$ respectively. Then it is easy to check that

$$(\theta, \lambda) = \theta\, \mathrm{e}_\lambda = \mathrm{e}_\lambda \theta\,, \tag{3.40}$$

and

$$\mathrm{e}_\lambda \mathrm{e}_\mu = (-1)^{\epsilon(\lambda,\mu)}\, \mathrm{e}_{\lambda+\mu}\,. \tag{3.41}$$

Using the above relation, it can be shown that (see Lemma A.1 for proof)

$$\mathrm{e}_\lambda \mathrm{e}_\mu = (-1)^{\lambda \circ \mu}\, \mathrm{e}_\mu \mathrm{e}_\lambda\,. \tag{3.42}$$

This property requires that the lattice be even and integral. Note that we chose an integral basis of $\Lambda$ to define the central extension. In Appendix A we show that a cocycle $\tilde{\epsilon}$ defined analogous to (3.37) for a different choice of basis is cohomologous to $\epsilon$, and hence gives rise to an isomorphic central extension. $\hat{\Lambda}$ acts on $\mathbb{C}[\Lambda]$ as follows

$$(\theta, \lambda')\, e^\lambda = \theta(-1)^{\epsilon(\lambda', \lambda)}\, e^{\lambda+\lambda'}\,. \tag{3.43}$$

In particular for $(\theta, \lambda') = (1, \lambda') = e_{\lambda'}$, we have

$$e_{\lambda'}\, e^\lambda = (-1)^{\epsilon(\lambda', \lambda)}\, e^{\lambda+\lambda'}\,. \tag{3.44}$$

Note that the same cocycle $\epsilon$ restricted to $\Lambda_0$

$$\epsilon : \Lambda_0 \times \Lambda_0 \longrightarrow \mathbb{Z}\,, \tag{3.45}$$

defines a central extension $\hat{\Lambda}_0 := \mathbb{Z}_2 \times \Lambda_0 \subset \hat{\Lambda}$:

$$0 \longrightarrow \mathbb{Z}_2 \longrightarrow \hat{\Lambda}_0 \longrightarrow \Lambda_0 \longrightarrow 0\,. \tag{3.46}$$

Moreover, the action (3.43) restricted to $\hat{\Lambda}_0$ makes $\mathbb{C}[\Lambda_0]$ into a $\hat{\Lambda}_0$-module. This makes $V_\Lambda$ into a $\hat{\Lambda}_0$-module where $\hat{\Lambda}_0$ acts only on $\mathbb{C}[\Lambda_0]$. Let $x, \bar{x}$ be formal variables. For any vector $\lambda = (\alpha^\lambda, \beta^\lambda)$, define the operators $x^{\alpha^\lambda}, \bar{x}^{\beta^\lambda}$ by the following actions

$$\begin{aligned}
x^{\alpha^\lambda}(u \otimes e^{\lambda'}) &= x^{\langle \alpha^\lambda, \alpha^{\lambda'}\rangle}(u \otimes e^{\lambda'})\,,\\
\bar{x}^{\beta^\lambda}(u \otimes e^{\lambda'}) &= \bar{x}^{\langle \beta^\lambda, \beta^{\lambda'}\rangle}(u \otimes e^{\lambda'})\,,
\end{aligned} \tag{3.47}$$

where $u \in S(\hat{\mathfrak{h}}^-)$, $\lambda' \in \Lambda_0$. Note that $x^{\alpha^\lambda}, \bar{x}^{\beta^\lambda}$ acts as $x^{\alpha^\lambda(0)}, \bar{x}^{\beta^\lambda(0)}$. For $\lambda = (\alpha^\lambda, \beta^\lambda) \in \Lambda_0$, define the vertex operators

$$\begin{aligned}
Y_{V_\Lambda}(e^\lambda, x, \bar{x}) := &\left[ \exp\left( -\sum_{r<0} \frac{\alpha^\lambda(r)}{r} x^{-r} \right) \exp\left( -\sum_{r>0} \frac{\alpha^\lambda(r)}{r} x^{-r} \right)\right.\\
&\left. \exp\left( -\sum_{r<0} \frac{\beta^\lambda(r)}{r} \bar{x}^{-r} \right) \exp\left( -\sum_{r>0} \frac{\beta^\lambda(r)}{r} \bar{x}^{-r} \right) \right] e_\lambda x^{\alpha^\lambda} \bar{x}^{\beta^\lambda}\,.
\end{aligned} \tag{3.48}$$

From the Lie bracket in (3.29), it is easy to show

$$[\alpha^\lambda(r), \beta^\lambda(s)] = 0 \tag{3.49}$$

for all $r, s \in \mathbb{Z}$, so that the order of exponentials with $\alpha^\lambda(r)$ and $\beta^\lambda(r)$ does not matter. For a formal variable $x$, we introduce the notation

$$\alpha^\lambda(x) = \underbrace{\sum_{r>0} \alpha^\lambda(r) x^{-r-1}}_{\alpha^\lambda(x)^+} + \underbrace{\sum_{r<0} \alpha^\lambda(r) x^{-r-1}}_{\alpha^\lambda(x)^-} + \alpha^\lambda(0) x^{-1}\,, \tag{3.50}$$

Similarly, we can also define $\beta^\lambda(\bar{x})$. We define the formal integration as the map by

$$\int dx\ x^r = \frac{x^{r+1}}{r+1}, \quad n \neq -1\,. \tag{3.51}$$

We can then write the vertex operator as

$$Y_{V_\Lambda}(\mathrm{e}^\lambda, x, \bar{x}) = \exp\left(\int dx\ \alpha^\lambda(x)^-\right)\exp\left(\int dx\ \alpha^\lambda(x)^+\right)$$
$$\times \exp\left(\int d\bar{x}\ \beta^\lambda(\bar{x})^-\right)\exp\left(\int d\bar{x}\ \beta^\lambda(\bar{x})^+\right)\mathrm{e}_\lambda\, x^{\alpha^\lambda}\bar{x}^{\beta^\lambda}\,. \tag{3.52}$$

For a general vector $v$ of the form (3.35), the vertex operator is defined as

$$Y_{V_\Lambda}(v, x, \bar{x}) = {}_{\circ}^{\circ}\prod_{r=1}^k \prod_{s=1}^{\bar{k}}\left(\frac{1}{(m_r-1)!}\frac{d^{m_r-1}\alpha_r(x)}{dx^{m_r-1}}\right)\left(\frac{1}{(\bar{m}_s-1)!}\frac{d^{\bar{m}_s-1}\beta_s(\bar{x})}{d\bar{x}^{\bar{m}_s-1}}\right)Y_{V_\Lambda}(\mathrm{e}^\lambda, x, \bar{x}){}_{\circ}^{\circ}\,, \tag{3.53}$$

where the normal ordering ${}_{\circ}^{\circ}$ is defined as

$$\begin{aligned}
{}_{\circ}^{\circ}\,\alpha^\lambda(p)\,\alpha^{\lambda'}(q){}_{\circ}^{\circ} &= {}_{\circ}^{\circ}\alpha^{\lambda'}(q)\,\alpha^\lambda(p){}_{\circ}^{\circ} = \begin{cases}\alpha^\lambda(p)\,\alpha^{\lambda'}(q) & p \leq q\,, \\ \alpha^{\lambda'}(q)\,\alpha^\lambda(p) & p \geq q\,,\end{cases} \\
{}_{\circ}^{\circ}\,\alpha^\lambda(p)\mathrm{e}_{\lambda'}{}_{\circ}^{\circ} &= {}_{\circ}^{\circ}\mathrm{e}_{\lambda'}\,\alpha^\lambda(p){}_{\circ}^{\circ} = \mathrm{e}_{\lambda'}\,\alpha^\lambda(p)\,, \\
{}_{\circ}^{\circ}\,x^{\alpha^\lambda}\,\mathrm{e}_{\lambda'}{}_{\circ}^{\circ} &= {}_{\circ}^{\circ}\mathrm{e}_{\lambda'}\,x^{\alpha^\lambda}{}_{\circ}^{\circ} = \mathrm{e}_{\lambda'}\,x^{\alpha^\lambda}\,,
\end{aligned} \tag{3.54}$$

and similarly for $\beta^\lambda$ and $\bar{x}^{\beta^\lambda}$. The vertex operator for general vectors in $V_\Lambda$ is defined by linear extension to all of $V_\Lambda$.

**Remark 3.1.** Using the central extension (3.39), (3.53) can be used to define vertex operators even if $\mathrm{e}^\lambda \in \mathbb{C}[\Lambda]$. These vertex operators will act on vectors of the form (3.35) with $\mathrm{e}^\lambda \in \mathbb{C}[\Lambda]$ rather than $\mathbb{C}[\Lambda_0]$. This will be crucial when we construct module vertex operators and intertwining operators on the modules of $V_\Lambda$.

The vacuum vector is given by $\mathbf{1} = \mathrm{e}^0$. The conformal vector is constructed below, see (3.70).

### 3.3  Proof of axioms

We now prove that $(V_\Lambda, Y_{V_\Lambda}, \omega_L, \omega_R, \mathbf{1})$ is a non-chiral VOA.

*Proof of identity property 1:* From the definition (3.48), it is clear that $Y_{V_\Lambda}(\mathbf{1}, x, \bar{x}) = \mathrm{e}_0 = \mathbb{1}$.

*Proof of grading-restriction property 2:* The grading on $V_\Lambda$ is given by defining the conformal weight of vector $v$ of the form (3.35) by

$$h_v = \frac{\langle \alpha, \alpha \rangle}{2} + \sum_{i=1}^k m_i, \quad \bar{h}_v = \frac{\langle \beta, \beta \rangle}{2} + \sum_{j=1}^{\bar{k}} \bar{m}_j \,, \tag{3.55}$$

where $m_i$ and $\bar{m}_j$ are positive integers appearing in (3.35). Note that for $e^\lambda$ with $\lambda = (\alpha, \beta) \in \Lambda_0$, we have $(\alpha, 0) \in \Lambda$, $(\alpha, 0) \circ (\alpha, 0) = \langle \alpha, \alpha \rangle \in 2\mathbb{Z}_+$. Then we have that $h_v, \bar{h}_v \geq 0$ so that $V_{(h, \bar{h})} = 0$ for $h$ or $\bar{h} < 0$, i.e. $M = 0$ in (2.23). Similarly, $\langle \beta, \beta \rangle \in 2\mathbb{Z}_+$, which implies that both $h_v$ and $\bar{h}_v$ are positive integers [14].

We will now show that $\dim(V_{(h, \bar{h})}) < \infty$. Note that $\Lambda_1^0$ and $\Lambda_2^0$ are lattices. It suffices to show that there exist only finitely many vectors of the form (3.35), satisfying the conditions in (3.55). We first show that for any $h, \bar{h} \in \mathbb{R}$ the number of distinct $\lambda = (\alpha, \beta) \in \Lambda_0$ satisfying

$$\langle \alpha, \alpha \rangle \leq 2h \text{ and } \langle \beta, \beta \rangle \leq 2\bar{h}\,, \text{ where } \alpha \in \Lambda_1^0, \beta \in \Lambda_2^0\,, \tag{3.56}$$

can be only finitely many. Consider the sets

$$X_1 = \{\alpha \in \Lambda_1^0 \mid \langle \alpha, \alpha \rangle \leq 2h\}, \tag{3.57}$$

$$X_2 = \{\beta \in \Lambda_2^0 \mid \langle \beta, \beta \rangle \leq 2\bar{h}\}, \tag{3.58}$$

which have finite cardinality, say $N_1$ and $N_2$, due to the fact that $\Lambda_1^0$ and $\Lambda_2^0$ are discrete. Then the set

$$X = \{\lambda = (\alpha, \beta) \in \Lambda_0 \mid \langle \alpha, \alpha \rangle \leq 2h, \ \langle \beta, \beta \rangle \leq 2\bar{h}\} \tag{3.59}$$

is finite because the map

$$
\begin{aligned}
X &\longrightarrow X_1 \times X_2 \\
\lambda = (\alpha, \beta) &\longmapsto (\alpha, \beta)
\end{aligned}
\tag{3.60}
$$

is injective. More precisely $\#X \leq N_1 N_2$. Now, as there are only finitely many combinations of positive integers $\{m_i\}_{i=1}^k$ and $\{\bar{m}_i\}_{i=1}^{\bar{k}}$ such that

$$h_v - \frac{\langle \alpha, \alpha \rangle}{2} = \sum_{i=1}^k m_i, \quad \bar{h}_v - \frac{\langle \beta, \beta \rangle}{2} = \sum_{j=1}^{\bar{k}} \bar{m}_j\,, \tag{3.61}$$

hence there are only finitely many generating vectors possible, which implies that $\dim(V_{h, \bar{h}}) < \infty$.

---

[14] Note that this argument also works for a general $e^\lambda \in \mathbb{C}[\Lambda]$ with $\lambda \in \Lambda$ since $\langle \alpha, \alpha \rangle, \langle \beta, \beta \rangle \geq 0$ even in this case.

*Proof of single-valuedness property 3:* For the general vector $v$ of the form (3.35) we have

$$h_v - \bar{h}_v = \frac{\langle \alpha, \alpha \rangle - \langle \beta, \beta \rangle}{2} + \sum_{i=1}^{k} m_i - \sum_{j=1}^{\bar{k}} \bar{m}_j$$

$$= \frac{\lambda \circ \lambda}{2} + \sum_{i=1}^{k} m_i - \sum_{j=1}^{\bar{k}} \bar{m}_j \in \mathbb{Z},$$

(3.62)

where we used the fact that $\Lambda$ is an even Lorentzian lattice.

*Proof of creation property 4:* We want to show that for any state $v \in V_\Lambda$,

$$\lim_{x, \bar{x} \to 0} Y_{V_\Lambda}(v, x, \bar{x}) \mathbf{1} = v.$$

(3.63)

Let us first consider the case when $v = \mathrm{e}^\lambda$, then the $Y_{V_\Lambda}$ operator is given in (3.48). One then has to expand the exponentials, we ignore the terms when $\alpha^\lambda(n)$ and $\beta^\lambda(n)$ have $n > 0$, as they annihilate $\mathbb{C}[\Lambda_0]$. The two exponentials that remain will only have positive powers of $x$ and $\bar{x}$, which vanish when we take the limit. Hence

$$\lim_{x, \bar{x} \to 0} Y_{V_\Lambda}(\mathrm{e}^\lambda, x, \bar{x}) \mathbf{1} = \mathrm{e}_\lambda \cdot \mathbf{1}.$$

(3.64)

Here, we used the fact that $\mathbf{1} = \mathrm{e}^0$ so that the action of $x^{\alpha^\lambda}(\bar{x}^{\beta^\lambda})$ on this is by identity, since $\langle \alpha^\lambda, 0 \rangle = 0$ $\left( \langle \beta^\lambda, 0 \rangle = 0 \right)$. Hence

$$\lim_{x, \bar{x} \to 0} Y_{V_\Lambda}(\mathrm{e}^\lambda, x, \bar{x}) \mathbf{1} = \mathrm{e}_\lambda \, \mathrm{e}^0 = (-1)^{\epsilon(0, \lambda)} \mathrm{e}^\lambda = \mathrm{e}^\lambda,$$

(3.65)

where we have used (3.37), (3.43) and that $\mathrm{e}_\lambda = (1, \lambda)$. We now prove (3.63) for a general vector $v$ of the form (3.35). The normal ordering in the definition (3.53) and the fact that $\hat{\mathfrak{h}}^+$ annihilates $\mathbb{C}[\Lambda_0]$ forces the product to take the form

$$\frac{d^{m_r - 1} \alpha_r(x)}{dx^{m_r - 1}} \to \sum_{p_r \leq -m_r} (m_r - 1)! \alpha_r(p_r) x^{-p_r - m_r},$$

(3.66)

and

$$\frac{d^{\bar{m}_s - 1} \beta_s(\bar{x})}{d\bar{x}^{\bar{m}_s - 1}} \to \sum_{q_s \leq -\bar{m}_s} (\bar{m}_s - 1)! \beta_s(q_s) \bar{x}^{-q_s - \bar{m}_s}.$$

(3.67)

Thus we have

$$
\substack{\circ\\\circ} \prod_{r=1}^{k}\prod_{s=1}^{\bar{k}} \left( \frac{1}{(m_r-1)!}\frac{d^{m_r-1}\alpha_r(x)}{dx^{m_r-1}} \right)\left( \frac{1}{(\bar{m}_s-1)!}\frac{d^{\bar{m}_s-1}\beta_s(\bar{x})}{d\bar{x}^{\bar{m}_s-1}} \right) Y_{V_\Lambda}(\mathrm{e}^\lambda,x,\bar{x})\,\substack{\circ\\\circ}\,\mathbf{1}
$$

$$
\to \sum_{\substack{p_1\leq -m_1\\ \cdots\\ p_k\leq -m_k}} \substack{\circ\\\circ}\alpha_1(p_1)\alpha_2(p_2)\dots\alpha_k(p_k)x^{-(p_1+\dots p_k)-(m_1+\dots+m_k)}
$$

$$
\times \sum_{\substack{q_1\leq -\bar{m}_1\\ \cdots\\ q_{\bar{k}}\leq -\bar{m}_{\bar{k}}}} \beta_1(q_1)\beta_2(q_2)\dots\beta_{\bar{k}}(q_{\bar{k}})\bar{x}^{-(q_1+\dots q_{\bar{k}})-(\bar{m}_1+\dots+\bar{m}_{\bar{k}})}Y_{V_\Lambda}(\mathrm{e}^\lambda,x,\bar{x})\,\substack{\circ\\\circ}\,\mathbf{1}. \tag{3.68}
$$

When we take $x,\bar{x}\to 0$ only the $p_r=-m_r, q_s=-\bar{m}_s$ terms in the sum survives. Combining this fact with the proof of (3.63) for $v=\mathrm{e}^\lambda$, we get

$$
\lim_{x,\bar{x}\to 0} Y_{V_\Lambda}(v,x,\bar{x})\mathbf{1}
$$
$$
= (\alpha_1(-m_1)\cdot\alpha_2(-m_2)\cdots\alpha_k(-m_k)\,\beta_1(-\bar{m}_1)\cdot\beta_2(-\bar{m}_2)\cdots\beta_\ell(-\bar{m}_{\bar{k}}))\otimes\mathrm{e}^\lambda = v, \tag{3.69}
$$

where we also use the fact that $Y_{V_\Lambda}(\mathrm{e}^\lambda,x,\bar{x})$ can only contribute terms with $x^n$ and $\bar{x}^m$, where $n$ and $m$ are greater than 0.

*Proof of Virasoro property 5:* The conformal vector is given by

$$
\omega_\Lambda := \frac{1}{2}\sum_{i=1}^{\dim(\mathfrak{h}_1)}\left(u_i(-1)^2\right)\otimes\mathbf{1} + \frac{1}{2}\sum_{i=1}^{\dim(\mathfrak{h}_2)}\left(v_i(-1)^2\right)\otimes\mathbf{1}\equiv\omega_L+\omega_R, \tag{3.70}
$$

where $u_i\in\Lambda_1\otimes_{\mathbb{Z}}\mathbb{C}, v_i\in\Lambda_2\otimes_{\mathbb{Z}}\mathbb{C}$ are orthonormal basis of $\mathfrak{h}_1$ and $\mathfrak{h}_2$ respectively[15]:

$$
\langle u_i,u_j\rangle = \delta_{i,j},\quad \langle v_i,v_j\rangle=\delta_{i,j}. \tag{3.71}
$$

Since an integral basis of $\Lambda$ is also a basis of $\mathbb{R}^{m,n}$, it is clear that $\dim(\mathfrak{h}_1)=m$ and $\dim(\mathfrak{h}_2)=n$. One can check that the conformal vertex operator is given by

$$
Y_{V_\Lambda}(\omega,x,\bar{x}) = Y_{V_\Lambda}(\omega_L,x,\bar{x}) + Y_{V_\Lambda}(\omega_R,x,\bar{x})
$$
$$
= \sum_{p\in\mathbb{Z}} L_\Lambda(p)x^{-p-2} + \sum_{p\in\mathbb{Z}} \bar{L}_\Lambda(p)\bar{x}^{-p-2}, \tag{3.72}
$$

where the Virasoro generators are given by [2, Section 8.7]

$$
L_\Lambda(p) = \frac{1}{2}\sum_{i=1}^{m}\sum_{k\in\mathbb{Z}} \substack{\circ\\\circ}u_i(k)u_i(p-k)\substack{\circ\\\circ}
$$
$$
\bar{L}_\Lambda(p) = \frac{1}{2}\sum_{i=1}^{n}\sum_{k\in\mathbb{Z}} \substack{\circ\\\circ}v_i(k)v_i(p-k)\substack{\circ\\\circ}\,. \tag{3.73}
$$

---

[15]Note that a different choice of orthonormal basis will give isomorphic LLVOAs. Indeed if $\{u_i'\}$ and $\{v_j'\}$ are orthonormal bases of $\mathfrak{h}_1$ and $\mathfrak{h}_2$ respectively, different from $\{u_i\}$ of and $\{v_j\}$. Then the map $f:V_\Lambda\longrightarrow V_\Lambda$ which acts trivially on $\mathbb{C}[\Lambda_0]$ and maps $u_i\mapsto u_i'$, $v_i\mapsto v_i'$ is a non-chiral VOA isomorphism.

Using the Lie brackets

$$
[\, u_i(p), u_j(q) \,] = p\,\delta_{i,j}\,\delta_{p+q,0}\,\mathbf{k},
$$
$$
[\, v_i(p), v_j(q) \,] = p\,\delta_{i,j}\,\delta_{p+q,0}\,\bar{\mathbf{k}},
$$
\hfill (3.74)

one can show that the Virasoro generators indeed satisfy the Virasoro algebra (2.27) with central charge $m = \dim(\mathfrak{h}_1), n = \dim(\mathfrak{h}_2)$ respectively, see [2, Chapter 2].

*Proof of grading property 6:* From (3.73) and normal ordering (3.54), we have

$$
L_\Lambda(0) = \frac{1}{2}\sum_{i=1}^{m}\sum_{r\in\mathbb{Z}}\,{}^{\circ}_{\circ}u_i(r)u_i(-r){}^{\circ}_{\circ} = \sum_{i=1}^{m}\sum_{r>0}u_i(-r)u_i(r) + \frac{1}{2}u_i(0)^2,
$$
$$
\bar{L}_\Lambda(0) = \frac{1}{2}\sum_{i=1}^{n}\sum_{r\in\mathbb{Z}}\,{}^{\circ}_{\circ}v_i(r)v_i(-r){}^{\circ}_{\circ} = \sum_{i=1}^{n}\sum_{r>0}v_i(-r)v_i(r) + \frac{1}{2}v_i(0)^2.
$$
\hfill (3.75)

Then for $v \in V_\Lambda$ of the form (3.35), using the Lie bracket (3.74) and the action (3.36), we have

$$
L_\Lambda(0)v = \sum_{j=1}^{k}\left[\cdots\left(m_j\sum_{i=1}^{m}\langle u_i,\alpha_j\rangle u_i(-m_j)\right)\cdots\right]\otimes e^\lambda + \frac{1}{2}\sum_{i=1}^{m}\langle u_i,\alpha\rangle^2 v\,,
$$
$$
= \sum_{j=1}^{k}m_j\left[\cdots\alpha_j(-m_j)\cdots\right]\otimes e^\lambda + \frac{1}{2}\sum_{i=1}^{m}\langle\langle u_i,\alpha\rangle u_i,\alpha\rangle v\,,
$$
$$
= \left[\sum_{j=1}^{k}m_j + \frac{\langle\alpha,\alpha\rangle}{2}\right]v\,,
$$
\hfill (3.76)

where we used the fact that $\{u_i\}$ is an orthonormal basis of $\mathfrak{h}_1$. Similarly

$$
\bar{L}_\Lambda(0)v = \left[\sum_{\bar{j}=1}^{\bar{k}}\bar{m}_{\bar{j}} + \frac{\langle\beta,\beta\rangle}{2}\right]v\,.
$$
\hfill (3.77)

*Proof of L(0)-property 7:* Let $(\alpha, \beta) \in \Lambda_0$. Then

$$
\begin{aligned}
\left[L_\Lambda(0), \alpha(x)^\pm\right] &= \frac{1}{2} \sum_{r \in \mathbb{Z}_\pm} \sum_{s \in \mathbb{Z}} \sum_{i=1}^{m} \left[\substack{\bullet\\\bullet} u_i(s) u_i(-s) \substack{\bullet\\\bullet}, \alpha(r)\right] x^{-r-1} \\
&= \frac{1}{2} \sum_{r \in \mathbb{Z}_\pm} \sum_{i=1}^{m} \left( \sum_{s \geq 1} [u_i(-s) u_i(s), \alpha(r)] + [u_i(0)^2, \alpha(r)] \right. \\
&\hspace{4cm} \left. + \sum_{s \leq -1} [u_i(s) u_i(-s), \alpha(r)] \right) x^{-r-1} \\
&= \frac{1}{2} \sum_{r \in \mathbb{Z}_\pm} \sum_{s \neq 0} (s\alpha(-s) \delta_{s+r,0} - n\alpha(s)) \, \delta_{r-s,0} x^{-r-1} \\
&= \sum_{r \in \mathbb{Z}_\pm} \sum_{s \neq 0} n\alpha(-s) \delta_{s+r,0} x^{-r-1} \\
&= -\sum_{r \in \mathbb{Z}_\pm} r\alpha(r) x^{-r-1},
\end{aligned}
$$

$$(3.78)$$

where we used

$$
\begin{aligned}
\sum_{i=1}^{m} [u_i(-s) u_i(s), \alpha(r)] &= \sum_{i=1}^{m} u_i(-s) [u_i(s), \alpha(r)] + [u_i(-s), \alpha(r)] u_i(s) \\
&= \sum_{i=1}^{m} (s \langle u_i, \alpha \rangle \delta_{s+r,0} u_i(-s) - s \langle u_i, \alpha \rangle \delta_{r-s,0} u_i(s)) \\
&= s\alpha(-s) \delta_{s+r,0} - s\alpha(s) \delta_{r-s,0}.
\end{aligned}
$$

$$(3.79)$$

Rearranging terms, we get

$$
\left[L_\Lambda(0), \alpha(x)^\pm\right] = x \frac{d\alpha(x)^\pm}{dx} + \alpha(x)^\pm. \tag{3.80}
$$

Similarly

$$
\left[\bar{L}_\Lambda(0), \beta(\bar{x})^\pm\right] = \bar{x} \frac{d}{d\bar{x}} \beta(\bar{x})^\pm + \beta(\bar{x})^\pm. \tag{3.81}
$$

Note that the same proof also shows that

$$
\begin{aligned}
[L_\Lambda(0), \alpha(x)] &= x \frac{d}{dx} \alpha(x) + \alpha(x) \\
[\bar{L}_\Lambda(0), \beta(\bar{x})] &= \bar{x} \frac{d}{d\bar{x}} \beta(\bar{x}) + \beta(\bar{x}).
\end{aligned}
\tag{3.82}
$$

Next using (3.80) we have

$$
\begin{aligned}
\left[L_\Lambda(0), \int dx\, \alpha(x)^\pm\right] &= \int dx\, \left[L_\Lambda(0), \alpha(x)^\pm\right] \\
&= \int dx\, \left(x \frac{d}{dx} \alpha(x)^\pm + \alpha(x)^\pm\right) \\
&= x\alpha(x)^\pm,
\end{aligned}
\tag{3.83}
$$

where we used integration by parts for the formal integration. Similarly

$$\left[\bar{L}_\Lambda(0), \int d\bar{x}\ \beta(\bar{x})^\pm\right] = \bar{x}\beta(\bar{x})^\pm. \tag{3.84}$$

By BCH formula (3.107) we have

$$L_\Lambda(0)\exp\left(\int dx\ \alpha(x)^\pm\right) = \exp\left(-\int dx\ \alpha(x)^\pm\right)\sum_{n=0}^\infty \frac{1}{n!}\left[\left(\int dx\ \alpha(x)^\pm\right)^n, L_\Lambda(0)\right]$$

$$= \exp\left(\int dx\ \alpha(x)^\pm\right)L_\Lambda(0) + \sum_{n=1}^\infty \frac{1}{n!}\left[\left(-\int dx\ \alpha(x)^\pm\right)^n, L_\Lambda(0)\right]. \tag{3.85}$$

By (3.83) and the fact that $[\alpha(r), \alpha(s)] = 0$ for $r, s \le 0$ or $r, s \ge 0$, we get

$$\left[L_\Lambda(0), \exp\left(\int dx\ \alpha(x)^\pm\right)\right] = x\exp\left(\int dx\ \alpha(x)^\pm\right)\alpha(x)^\pm. \tag{3.86}$$

Similarly

$$\left[\bar{L}_\Lambda(0), \exp\left(\int d\bar{x}\ \beta(x)^\pm\right)\right] = \bar{x}\exp\left(\int d\bar{x}\ \beta(\bar{x})^\pm\right)\beta(\bar{x})^\pm. \tag{3.87}$$

Finally, it is clear that for $\lambda' = (\alpha', \beta')$ and $u \in S(\mathfrak{h}^-)$ we have

$$[L_\Lambda(0), \mathrm{e}_\lambda x^\alpha]\left(u \otimes \mathrm{e}^{\lambda'}\right) = (-1)^{\epsilon(\lambda,\lambda')}\left(\frac{\langle\alpha+\alpha', \alpha+\alpha'\rangle}{2} - \frac{\langle\alpha', \alpha'\rangle}{2}\right)x^{\langle\alpha,\alpha'\rangle}\left(u \otimes \mathrm{e}^{\lambda+\lambda'}\right)$$

$$= \left(\frac{\langle\alpha, \alpha\rangle}{2} + \langle\alpha, \alpha'\rangle\right)(-1)^{\epsilon(\lambda,\lambda')}x^{\langle\alpha,\alpha'\rangle}\left(u \otimes \mathrm{e}^{\lambda+\lambda'}\right)$$

$$= \left(\frac{\langle\alpha, \alpha\rangle}{2}\mathrm{e}_\lambda x^\alpha + \mathrm{e}_\lambda x\frac{d}{dx}x^\alpha\right)\left(u \otimes \mathrm{e}^{\lambda'}\right) \tag{3.88}$$

Putting all this together, we obtain

$$\left[L_\Lambda(0), Y_{V_\Lambda}\left(\mathrm{e}^\lambda, x, \bar{x}\right)\right] = x\frac{d}{dx}\left[\exp\left(\int dx\ \alpha(x)^-\right)\exp\left(\int dx\ \alpha(x)^+\right)\right]$$

$$\times \exp\left(\int d\bar{x}\ \beta(\bar{x})^-\right)\exp\left(\int d\bar{x}\ \beta(\bar{x})^+\right)\mathrm{e}_\lambda x^\alpha \bar{x}^\beta$$

$$+ \exp\left(\int dx\ \alpha(x)^-\right)\exp\left(\int dx\ \alpha(x)^+\right)$$

$$\times \exp\left(\int d\bar{x}\ \beta(\bar{x})^-\right)\exp\left(\int d\bar{x}\ \beta(\bar{x})^+\right)$$

$$\times \left(\frac{\langle\alpha, \alpha\rangle}{2}\mathrm{e}_\lambda x^\alpha + \mathrm{e}_\lambda x\frac{d}{dx}x^\alpha\right)$$

$$= x\frac{d}{dx}Y_{V_\Lambda}\left(\mathrm{e}^\lambda, x, \bar{x}\right) + \frac{\langle\alpha, \alpha\rangle}{2}Y_{V_\Lambda}\left(\mathrm{e}^\lambda, x, \bar{x}\right). \tag{3.89}$$

Similarly

$$\left[\bar{L}_\Lambda(0), Y_{V_\Lambda}\left(\mathrm{e}^\lambda, x, \bar{x}\right)\right] = \bar{x}\frac{d}{d\bar{x}}Y_{V_\Lambda}\left(\mathrm{e}^\lambda, x, \bar{x}\right) + \frac{\langle\beta,\beta\rangle}{2}Y_{V_\Lambda}\left(\mathrm{e}^\lambda, x, \bar{x}\right). \tag{3.90}$$

For general vertex operators, we observe that

$$\begin{aligned}
\left[L_\Lambda(0), \frac{d^r}{dx^r}\alpha(x)\right] &= \frac{d^r}{dx^r}\left(x\frac{d}{dx}\alpha(x) + \alpha(x)\right) \\
&= \frac{d^{r-1}}{dx^{r-1}}\left(\frac{d}{dx}\alpha(x) + x\frac{d^2}{dx^2}\alpha(x)\right) + \frac{d^r}{dx^r}\alpha(x) \\
&= x\frac{d^r}{dx^r}\alpha(x) + (r+1)\frac{d^r}{dx^r}\alpha(x).
\end{aligned} \tag{3.91}$$

This implies that for a general vector of the form (3.35) we have

$$\begin{aligned}
[L_\Lambda(0), Y_{V_\Lambda}(v, x, \bar{x})] &= x\frac{d}{dx}Y_{V_\Lambda}(v, x, \bar{x}) + \left(\sum_{i=1}^k m_i + \frac{\langle\alpha,\alpha\rangle}{2}\right)Y_{V_\Lambda}(v, x, \bar{x}) \\
&= x\frac{d}{dx}Y_{V_\Lambda}(v, x, \bar{x}) + Y_{V_\Lambda}\left(L_\Lambda(0)v, x, \bar{x}\right).
\end{aligned} \tag{3.92}$$

Similarly

$$\left[\bar{L}_\Lambda(0), Y_{V_\Lambda}(v, x, \bar{x})\right] = \bar{x}\frac{d}{d\bar{x}}Y_{V_\Lambda}(v, x, \bar{x}) + Y_{V_\Lambda}\left(\bar{L}_\Lambda(0)v, x, \bar{x}\right). \tag{3.93}$$

*Proof of translation property 8:* Observe that

$$\begin{aligned}
[L_\Lambda(-1), \alpha(-r)] &= \frac{1}{2}\sum_{i=1}^m \sum_{s\in\mathbb{Z}}[{}_\circ^\circ u_i(s)u_i(-1-s){}_\circ^\circ, \alpha(-r)] \\
&= \frac{1}{2}\sum_{i=1}^m \sum_{s\in\mathbb{Z}}[u_i(-1-s)\, u_i(s), \alpha(-r)] \\
&= \frac{1}{2}\sum_{i=1}^m \sum_{s\in\mathbb{Z}}\Big(u_i(-1-s)[u_i(s), \alpha(-r)] + [u_i(-1-s), \alpha(-r)]\, u_i(s)\Big) \\
&= \frac{1}{2}\sum_{i=1}^m \sum_{s\in\mathbb{Z}}\Big(s\delta_{s-r,0}\langle u_i, \alpha\rangle\, u_i(-1-s) - (1+s)\delta_{r+s+1,0}\langle u_i, \alpha\rangle\, u_i(s)\Big) \\
&= \frac{1}{2}\Big(r\,\alpha(-1-r) - (-r)\,\alpha(-1-r)\Big) = r\,\alpha(-r-1).
\end{aligned} \tag{3.94}$$

Using the above commutator, it is easy to see that

$$L_\Lambda(-1)\left(\alpha_1(-m_1)\cdots\alpha_k(-m_k)\,\beta_1(-\bar{m}_1)\cdot\beta_2(-\bar{m}_2)\cdots\beta_{\bar{k}}(-\bar{m}_{\bar{k}})\right)\otimes \mathrm{e}^{(\alpha,\beta)}$$

$$= \sum_{i=1}^{m}\left(\alpha_1(-m_1)\cdots\alpha_k(-m_k)\,\beta_1(-\bar{m}_1)\cdots\beta_{\bar{k}}(-\bar{m}_{\bar{k}})\right)\otimes L_\Lambda(-1)\mathrm{e}^{(\alpha,\beta)}$$

$$+ \sum_{i=1}^{k}m_i\left(\alpha_1(-m_1)\cdots\alpha_i(-1-m_i)\cdots\alpha_k(-m_k)\,\beta_1(-\bar{m}_1)\cdots\beta_{\bar{k}}(-\bar{m}_{\bar{k}})\right)\otimes \mathrm{e}^{(\alpha,\beta)}.$$
(3.95)

Now since

$$\begin{aligned}
L_\Lambda(-1)\mathrm{e}^{(\alpha,\beta)} &= \frac{1}{2}\sum_{i=1}^{m}\sum_{s\in\mathbb{Z}} {}_\bullet^\bullet u_i(s)u_i(-1-s){}_\bullet^\bullet\,\mathrm{e}^{(\alpha,\beta)} \\
&= \sum_{i=1}^{m}u_i(0)u_i(-1)\mathrm{e}^{(\alpha,\beta)} \\
&= \sum_{i=1}^{m}\langle u_i,\alpha\rangle u_i(-1)\mathrm{e}^{(\alpha,\beta)} \\
&= \alpha(-1)\mathrm{e}^{(\alpha,\beta)},
\end{aligned}$$
(3.96)

hence we get the action of $L_\Lambda(-1)$ on generating vectors $v$ of the form (3.35) to be:

$$L_\Lambda(-1)v = \sum_{i=1}^{k}m_i\left(\alpha_1(-m_1)\cdots\alpha_i(-1-m_i)\cdots\alpha_k(-m_k)\,\beta_1(-\bar{m}_1)\cdots\beta_{\bar{k}}(-\bar{m}_{\bar{k}})\right)\otimes \mathrm{e}^{(\alpha,\beta)}$$

$$+ \left(\alpha(-1)\alpha_1(-m_1)\cdots\alpha_i(-1-m_i)\cdots\alpha_k(-m_k)\,\beta_1(-\bar{m}_1)\cdots\beta_{\bar{k}}(-\bar{m}_{\bar{k}})\right)\otimes \mathrm{e}^{(\alpha,\beta)}.$$
(3.97)

The proof of the translation property now follows from exact same calculation as in [33, Proposition 2.2].

### 3.3.1 Proof of locality of vertex operators

In this section, we will prove the locality of two vertex operators and defer the proof of product of multiple vertex operators to Appendix B.

**Proposition 3.1.** *The vertex operators $Y_{V_\Lambda}(\mathrm{e}^\lambda, x, \bar{x})$ for $\lambda \in \Lambda_0$ satisfy the locality property 9. More precisely there exists, multi-valued, operator-valued functions $f(z_1, z_2)$ and $g(\bar{z}_1, \bar{z}_2)$ analytic in $z_1, z_2$ and $\bar{z}_1, \bar{z}_2$ respectively with possible singularities at $\{(z_1, z_2) \in \mathbb{C}^2 \,|\, z_1, z_2 \neq 0, z_1 \neq z_2\}$, such that $f(z_1, z_2)g(\bar{z}_1, \bar{z}_2)$ is single-valued when $\bar{z}_1, \bar{z}_2$ are the complex conjugates of $z_1, z_2$ respectively and equals*

$$\begin{aligned}
&Y_{V_\Lambda}(\mathrm{e}^\lambda, z_1, \bar{z}_1)Y_{V_\Lambda}(\mathrm{e}^{\lambda'}, z_2, \bar{z}_2) \text{ when } |z_1| > |z_2|\,, \\
&Y_{V_\Lambda}(\mathrm{e}^{\lambda'}, z_2, \bar{z}_2)Y_{V_\Lambda}(\mathrm{e}^\lambda, z_1, \bar{z}_1) \text{ when } |z_2| > |z_1|\,.
\end{aligned}$$
(3.98)

*Proof.* We will closely follow the proofs of results in [33, Section 2]. We begin by proving that

$$[\alpha(x_1), \alpha'(x_2)] = \langle \alpha, \alpha' \rangle \left[ (x_2 - x_1)^{-2} - (-x_2 + x_1)^{-2} \right]$$
$$[\beta(\bar{x}_1), \beta'(\bar{x}_2)] = \langle \beta, \beta' \rangle \left[ (\bar{x}_2 - \bar{x}_1)^{-2} - (-\bar{x}_2 + \bar{x}_1)^{-2} \right] \tag{3.99}$$

where $\lambda = (\alpha, \beta), \lambda' = (\alpha', \beta')$. We have

$$
\begin{aligned}
[\alpha(x_1), \alpha'(x_2)] &= \sum_{r,s \in \mathbb{Z}} [\alpha(r), \alpha'(s)] x_1^{-r-1} x_2^{-s-1} \\
&= \sum_{r,s \in \mathbb{Z}} \langle \alpha, \alpha' \rangle \, r \, \delta_{r+s,0} \, x_1^{-r-1} x_2^{-s-1} \\
&= -\langle \alpha, \alpha' \rangle \sum_{s \in \mathbb{Z}} s \, x_1^{s-1} x_2^{-s-1} \\
&= -\langle \alpha, \alpha' \rangle \frac{\partial}{\partial x_1} \sum_{s \in \mathbb{Z}} x_1^{s} x_2^{-s-1} \\
&= -\langle \alpha, \alpha' \rangle \frac{\partial}{\partial x_1} \left( (x_1 - x_2)^{-1} - (-x_2 + x_1)^{-1} \right) \\
&= \langle \alpha, \alpha' \rangle \left( (x_1 - x_2)^{-2} - (-x_2 + x_1)^{-2} \right),
\end{aligned}
\tag{3.100}
$$

Note that this commutator is also true for complex variables $x_1 = z_1, x_2 = z_2$. Indeed from (2.10)

$$
\begin{aligned}
\alpha(z_1)\alpha'(z_2) &= {}^{\circ}_{\circ}\alpha(z_1)\alpha(z_2){}^{\circ}_{\circ} + \langle \alpha, \alpha' \rangle \sum_{s \in \mathbb{Z}} s z_1^{-s-1} z_2^{s-1} \\
&= {}^{\circ}_{\circ}\alpha(z_1)\alpha(z_2){}^{\circ}_{\circ} + \frac{\langle \alpha, \alpha' \rangle}{(z_1 - z_2)^2}, \quad |z_1| > |z_2|,
\end{aligned}
\tag{3.101}
$$

and

$$
\begin{aligned}
\alpha'(z_2)\alpha(z_1) &= {}^{\circ}_{\circ}\alpha'(z_2)\alpha(z_1){}^{\circ}_{\circ} - \langle \alpha', \alpha \rangle \sum_{s \in \mathbb{Z}} s z_2^{-s-1} z_1^{s-1} \\
&= {}^{\circ}_{\circ}\alpha'(z_2)\alpha(z_1){}^{\circ}_{\circ} - \frac{\langle \alpha', \alpha \rangle}{(z_2 - z_1)^2}, \quad |z_2| > |z_1|.
\end{aligned}
\tag{3.102}
$$

It is easy to see that

$$
{}^{\circ}_{\circ}\alpha(z_1)\alpha'(z_2){}^{\circ}_{\circ} = {}^{\circ}_{\circ}\alpha'(z_2)\alpha(z_1){}^{\circ}_{\circ}, \tag{3.103}
$$

which gives us the commutator. In particular

$$
[\alpha(z_1), \alpha'(z_2)] = 0. \tag{3.104}
$$

The other Lie bracket in (3.99) can be proved similarly. Using (3.99), we can show that[16]

$$
\left[ \alpha'(x_2)^-, e^{\int \alpha(x_1)^+ dx_1} \right] = \frac{\langle \alpha, \alpha' \rangle}{x_1 - x_2} e^{\int \alpha(x_1)^+ dx_1}. \tag{3.105}
$$

---

[16]we will use exp and e interchangeably.

Integrating both sides of (3.105) gives us

$$\left[ -\int \alpha' \, (x_2)^- \, dx_2, \, e^{\int \alpha(x_1)^+ dx_1} \right]$$

$$= (\langle \alpha, \alpha' \rangle \log (x_1 - x_2) - \langle \alpha, \alpha' \rangle \log x_1) \, e^{\int \alpha(x_1)^+ dx_1}$$

$$= \langle \alpha, \alpha' \rangle \, (\log(x_1 - x_2) - \log(x_1)) \, e^{\int \alpha(x_1)^+ dx_1} = \langle \alpha, \alpha' \rangle \log \left( 1 - \frac{x_2}{x_1} \right) e^{\int \alpha(x_1)^+ dx_1}.$$
(3.106)

One can write analogous formulas for $[\beta'(\bar{x}_1)^\pm, \exp(\int \beta(x_2)^\mp)]$. Using the BCH identity

$$\exp(X) Y \exp(-X) = \sum_{s=0}^{\infty} \frac{[(X)^s, Y]}{s!},$$
(3.107)

where

$$[X^s, Y] = \underbrace{[X \ldots, [X, [X, Y]] \ldots]}_{s \text{ times}}, \quad [X^0, Y] \equiv Y,$$
(3.108)

we get

$$\exp \left( -\int dx_2 \, \alpha'(x_2)^- \right) \exp \left( \int dx_1 \, \alpha(x_1)^+ \right) \exp \left( \int dx_2 \, \alpha'(x_2)^- \right)$$

$$= \left( 1 - \frac{x_2}{x_1} \right)^{\langle \alpha, \alpha' \rangle} \exp \left( \int dx_1 \, \alpha(x_1)^+ \right).$$
(3.109)

To show the locality of vertex operator, we will also require the identities

$$x_1^\alpha e_{\lambda'} = x_1^{\langle \alpha, \alpha' \rangle} e_{\lambda'} x_1^\alpha,$$
$$\bar{x}_2^\beta e_{\lambda'} = \bar{x}_2^{\langle \beta, \beta' \rangle} e_{\lambda'} \bar{x}_2^\beta,$$
(3.110)

the first of which is shown below

$$x_1^\alpha \, e_{\lambda'} (u \otimes e^{\lambda''}) = (-1)^{\epsilon(\lambda', \lambda'')} x_1^\alpha (u \otimes e^{\lambda' + \lambda''}) = (-1)^{\epsilon(\lambda', \lambda'')} x_1^{\langle \alpha, \alpha' \rangle + \langle \alpha, \alpha'' \rangle} (u \otimes e^{\lambda' + \lambda''})$$

$$e_{\lambda'} \, x_1^\alpha (u \otimes e^{\lambda''}) = x_1^{\langle \alpha, \alpha'' \rangle} e_{\lambda'} (u \otimes e^{\lambda''}) = (-1)^{\epsilon(\lambda', \lambda'')} x_1^{\langle \alpha, \alpha'' \rangle} (u \otimes e^{\lambda' + \lambda''}).$$
(3.111)

We now have[17]

$$Y_{V_\Lambda}(e^\lambda, x_1, \bar{x}_1) Y_{V_\Lambda}(e^{\lambda'}, x_2, \bar{x}_2)$$

$$= \exp \left( \int \alpha(x_1)^- \right) \exp \left( \int \alpha(x_1)^+ \right) \exp \left( \int \beta(\bar{x}_1)^- \right) \exp \left( \int \beta(\bar{x}_1)^+ \right) e_\lambda x_1^\alpha \bar{x}_1^\beta$$

$$\exp \left( \int \alpha'(x_2)^- \right) \exp \left( \int \alpha'(x_2)^+ \right) \exp \left( \int \beta'(\bar{x}_2)^- \right) \exp \left( \int \beta'(\bar{x}_2)^+ \right) e_{\lambda'} x_2^{\alpha'} \bar{x}_2^{\beta'}.$$
(3.112)

---

[17]We will often write $\int dx \, \alpha(x) = \int \alpha(x)$ to simplify the expressions.

Now, utilizing (3.49) and the fact that $e_\lambda$, $x_1^\alpha$, and $\bar{x}_1^\beta$ commute with exponential of integrals

$$= \exp\left(\int \alpha(x_1)^-\right)\left[\exp\left(\int \alpha(x_1)^+\right)\exp\left(\int \alpha'(x_2)^-\right)\right]\exp\left(\int \alpha'(x_2)^+\right)$$
$$\exp\left(\int \beta(\bar{x}_1)^-\right)\left[\exp\left(\int \beta(\bar{x}_1)^+\right)\exp\left(\int \beta'(\bar{x}_2)^-\right)\right]\exp\left(\int \beta'(\bar{x}_2)^+\right)e_\lambda x_1^\alpha \bar{x}_1^\beta e_{\lambda'} x_2^{\alpha'} \bar{x}_2^{\beta'}\ .$$
$$(3.113)$$

After which we use (3.109) and (3.110) to write

$$= \left(1 - \frac{x_2}{x_1}\right)^{\langle \alpha,\alpha'\rangle}\left(1 - \frac{\bar{x}_2}{\bar{x}_1}\right)^{\langle \beta,\beta'\rangle} x_1^{\langle \alpha,\alpha'\rangle}\bar{x}_1^{\langle \beta,\beta'\rangle}\exp\left(\int \alpha(x_1)^-\right)\exp\left(\int \alpha'(x_2)^-\right)$$
$$\exp\left(\int \alpha(x_1)^+\right)\exp\left(\int \alpha'(x_2)^+\right)\exp\left(\int \beta(\bar{x}_1)^-\right)\exp\left(\int \beta'(\bar{x}_2)^-\right)$$
$$\exp\left(\int \beta(\bar{x}_1)^+\right)\exp\left(\int \beta'(\bar{x}_2)^+\right)e_\lambda e_{\lambda'} x_1^\alpha \bar{x}_1^\beta x_2^{\alpha'}\bar{x}_2^{\beta'}\ .$$
$$(3.114)$$

Finally we use (2.6) to collect terms to get

$$= (x_1 - x_2)^{\langle \alpha,\alpha'\rangle}(\bar{x}_1 - \bar{x}_2)^{\langle \beta,\beta'\rangle}\exp\left(\int \alpha(x_1)^-\right)\exp\left(\int \alpha'(x_2)^-\right)\exp\left(\int \alpha(x_1)^+\right)$$
$$\exp\left(\int \alpha'(x_2)^+\right)\exp\left(\int \beta(\bar{x}_1)^-\right)\exp\left(\int \beta'(\bar{x}_2)^-\right)\exp\left(\int \beta(\bar{x}_1)^+\right)\exp\left(\int \beta'(\bar{x}_2)^+\right)$$
$$e_\lambda e_{\lambda'} x_1^\alpha \bar{x}_1^\beta x_2^{\alpha'}\bar{x}_2^{\beta'}$$
$$\equiv (x_1 - x_2)^{\langle \alpha,\alpha'\rangle}(\bar{x}_1 - \bar{x}_2)^{\langle \beta,\beta'\rangle} F(x_1,x_2)\bar{F}(\bar{x}_1,\bar{x}_2),$$
$$(3.115)$$

where we used (3.49), (3.109) and (3.110) and $F(x_1,x_2)\bar{F}(\bar{x}_1,\bar{x}_2)$ contains the operator part of $Y_{V_\Lambda}(e^\lambda, x_1, \bar{x}_1)Y_{V_\Lambda}(e^{\lambda'}, x_2, \bar{x}_2)$. Similarly we have

$Y_{V_\Lambda}(e^{\lambda'}, x_2, \bar{x}_2)Y_{V_\Lambda}(e^\lambda, x_1, \bar{x}_1)$

$$= (x_2 - x_2)^{\langle \alpha,\alpha'\rangle}(\bar{x}_2 - \bar{x}_1)^{\langle \beta,\beta'\rangle}\exp\left(\int \alpha'(x_2)^-\right)\exp\left(\int \alpha(x_1)^-\right)\exp\left(\int \alpha'(x_2)^+\right)$$
$$\exp\left(\int \alpha(x_1)^+\right)\exp\left(\int \beta'(\bar{x}_2)^-\right)\exp\left(\int \beta(\bar{x}_1)^-\right)\exp\left(\int \beta'(\bar{x}_2)^+\right)\exp\left(\int \beta(\bar{x}_1)^+\right)$$
$$(-1)^{\lambda\circ\lambda'}e_\lambda e_{\lambda'} x_1^\alpha \bar{x}_1^\beta x_2^{\alpha'}\bar{x}_2^{\beta'}$$
$$= (-1)^{\lambda\circ\lambda'}(x_2 - x_1)^{\langle \alpha,\alpha'\rangle}(\bar{x}_2 - \bar{x}_1)^{\langle \beta,\beta'\rangle} F(x_1,x_2)\bar{F}(\bar{x}_1,\bar{x}_2),$$
$$(3.116)$$

where we used (3.42). Note that $(-x_1 + x_2)^{\langle \alpha,\alpha'\rangle} = (x_2 - x_1)^{\langle \alpha,\alpha'\rangle}$ when[18] $\langle \alpha,\alpha'\rangle \geq 0$. To prove locality, we take complex variables $x_1 = z_1, x_2 = z_2$ and $\bar{x}_1 = \bar{z}_1, \bar{x}_2 = \bar{z}_2$.

---

[18]Recall that when $s \in \mathbb{C}$, $(-x_1 + x_2)^s$ is to be expanded in positive integral powers of $x_2$ as in (2.4).

Note that when we plug complex variable in place of formal variable, we must consider $(x_1 - x_2)^s$ as a formal series so that

$$Y_{V_\Lambda}(e^\lambda, z_1, \bar{z}_1) Y_{V_\Lambda}(e^{\lambda'}, z_2, \bar{z}_2) = \left(\sum_{p \geq 0} (-1)^p z_1^{\langle \alpha, \alpha' \rangle - p} z_2^p\right)\left(\sum_{q \geq 0} (-1)^q \bar{z}_1^{\langle \beta, \beta' \rangle - q} \bar{z}_2^q\right)$$
$$\times F(z_1, z_2) \bar{F}(\bar{z}_1, \bar{z}_2),$$
$$(3.117)$$

and similarly $Y_{V_\Lambda}(e^{\lambda'}, z_2, \bar{z}_2) Y_{V_\Lambda}(e^\lambda, z_1, \bar{z}_1)$. To complete the proof of locality, consider the operator valued functions

$$f(z_1, z_2) = \exp\left(\langle \alpha, \alpha' \rangle \log(z_1 - z_2)\right) F(z_1, z_2),$$
$$g(\bar{z}_1, \bar{z}_2) = \exp\left(\langle \beta, \beta' \rangle \log(\bar{z}_1 - \bar{z}_2)\right) \bar{F}(\bar{z}_1, \bar{z}_2).$$
$$(3.118)$$

Then by (2.10) for $|z_1| > |z_2|$ we see that

$$f(z_1, z_2) g(\bar{z}_1, \bar{z}_2) = \left(\sum_{p \geq 0} (-1)^p z_1^{\langle \alpha, \alpha' \rangle - p} z_2^p\right)\left(\sum_{q \geq 0} (-1)^q \bar{z}_1^{\langle \beta, \beta' \rangle - q} \bar{z}_2^q\right) F(z_1, z_2) \bar{F}(\bar{z}_1, \bar{z}_2).$$
$$(3.119)$$

For $|z_2| > |z_1|$ we have

$$f(z_1, z_2) g(\bar{z}_1, \bar{z}_2) = \exp\left(\langle \alpha, \alpha' \rangle \log(-(z_2 - z_1))\right) \exp\left(\langle \beta, \beta' \rangle \log(-(\bar{z}_2 - \bar{z}_1))\right)$$
$$= e^{i\pi(\langle \alpha, \alpha' \rangle - \langle \beta, \beta' \rangle)} \exp\left(\langle \alpha, \alpha' \rangle \log(z_2 - z_1)\right) \exp\left(\langle \beta, \beta' \rangle \log(\bar{z}_2 - \bar{z}_1)\right) F(z_1, z_2) \bar{F}(\bar{z}_1, \bar{z}_2)$$
$$= (-1)^{\lambda \circ \lambda'} \left(\sum_{p \geq 0} (-1)^p z_2^{\langle \alpha, \alpha' \rangle - p} z_1^p\right)\left(\sum_{q \geq 0} (-1)^q \bar{z}_2^{\langle \beta, \beta' \rangle - q} \bar{z}_1^q\right) F(z_1, z_2) \bar{F}(\bar{z}_1, \bar{z}_2),$$
$$(3.120)$$

where we used the fact that in the principal branch of logarithm to write

$$\log(-z) = \log|z| + i(\pi + \text{Arg}(z)), \quad \log(-\bar{z}) = \log|z| - i(\pi + \text{Arg}(z)) \quad (3.121)$$

with

$$-\pi < \pi + \text{Arg}(z) < \pi. \quad (3.122)$$

$\square$

**Remark 3.2.** From the calculations above, it is easy to see that the following formal commutativity axiom holds for the vertex operators: there exists $K, \bar{K} \in \mathbb{N}$ such that

$$(x_1 - x_2)^K (\bar{x}_1 - \bar{x}_2)^{\bar{K}} \left[Y_{V_\Lambda}(e^\lambda, x_1, \bar{x}_1), Y_{V_\Lambda}(e^{\lambda'}, x_2, \bar{x}_2)\right] = 0. \quad (3.123)$$

Indeed we have

$$\left[Y_{V_\Lambda}(e^\lambda, x_1, \bar{x}_1), \; Y_{V_\Lambda}(e^{\lambda'}, x_2, \bar{x}_2)\right] = \left((x_1 - x_2)^{\langle \alpha, \alpha' \rangle} (\bar{x}_1 - \bar{x}_2)^{\langle \beta, \beta' \rangle}\right.$$
$$\left. - (-1)^{\lambda \circ \lambda'} (x_2 - x_1)^{\langle \alpha, \alpha' \rangle} (\bar{x}_2 - \bar{x}_1)^{\langle \beta, \beta' \rangle}\right) F(x_1, x_2) \bar{F}(\bar{x}_1, \bar{x}_2)$$
$$= \left((x_1 - x_2)^{\langle \alpha, \alpha' \rangle} (\bar{x}_1 - \bar{x}_2)^{\langle \beta, \beta' \rangle}\right.$$
$$\left. - (-x_2 + x_1)^{\langle \alpha, \alpha' \rangle} (-\bar{x}_2 + \bar{x}_1)^{\langle \beta, \beta' \rangle}\right) F(x_1, x_2) \bar{F}(\bar{x}_1, \bar{x}_2)$$
$$(3.124)$$

Since $(\alpha, \beta), (\alpha', \beta') \in \Lambda_0$, we have $\langle \alpha, \alpha' \rangle, \langle \beta, \beta' \rangle \in \mathbb{Z}$ and we can choose $K, \bar{K} \in \mathbb{N}$ large enough such that

$$K + \langle \alpha, \alpha' \rangle \in \mathbb{N}, \quad \bar{K} + \langle \beta, \beta' \rangle \in \mathbb{N}. \tag{3.125}$$

We then get

$$
\begin{aligned}
(x_1 - x_2)^K &(\bar{x}_1 - \bar{x}_2)^{\bar{K}} \left[ Y_{V_\Lambda}(e^\lambda, x_1, \bar{x}_1), Y_{V_\Lambda}(e^{\lambda'}, x_2, \bar{x}_2) \right] \\
= &\Big( (x_1 - x_2)^{K + \langle \alpha, \alpha' \rangle} (\bar{x}_1 - \bar{x}_2)^{\bar{K} + \langle \beta, \beta' \rangle} \\
&\qquad - (-x_2 + x_1)^{K + \langle \alpha, \alpha' \rangle} (-\bar{x}_2 + \bar{x}_1)^{\bar{K} + \langle \beta, \beta' \rangle} \Big) F(x_1, x_2) \bar{F}(\bar{x}_1, \bar{x}_2) \\
= &\Big( (x_1 - x_2)^{K + \langle \alpha, \alpha' \rangle} (\bar{x}_1 - \bar{x}_2)^{\bar{K} + \langle \beta, \beta' \rangle} \\
&\qquad - (x_1 - x_2)^{K + \langle \alpha, \alpha' \rangle} (\bar{x}_1 - \bar{x}_2)^{\bar{K} + \langle \beta, \beta' \rangle} \Big) F(x_1, x_2) \bar{F}(\bar{x}_1, \bar{x}_2) \\
= &\, 0.
\end{aligned}
\tag{3.126}
$$

We now prove the locality for general vertex operators.

**Theorem 3.3.** *The vertex operators $Y_{V_\Lambda}(v, x, \bar{x})$, where $v$ is the general vector of $V_\Lambda$, satisfy the locality property 9. More precisely there exists, multi-valued, operator-valued functions $f(z_1, z_2)$ and $g(\bar{z}_1, \bar{z}_2)$ analytic in $z_1, z_2$ and $\bar{z}_1, \bar{z}_2$ respectively with possible singularities at $\{(z_1, z_2) \in \mathbb{C}^2 \,|\, z_1, z_2 \neq 0, z_1 \neq z_2\}$, such that $f(z_1, z_2) g(\bar{z}_1, \bar{z}_2)$ is single-valued when $\bar{z}_1, \bar{z}_2$ are the complex conjugates of $z_1, z_2$ respectively and equals*

$$
\begin{aligned}
Y_{V_\Lambda}(v, z_1, \bar{z}_1) Y_{V_\Lambda}(w, z_2, \bar{z}_2) \text{ when } |z_1| > |z_2|, \\
Y_{V_\Lambda}(w, z_2, \bar{z}_2) Y_{V_\Lambda}(v, z_1, \bar{z}_1) \text{ when } |z_2| > |z_1|.
\end{aligned}
\tag{3.127}
$$

*Proof.* We will prove the locality for the spanning set of vectors of the form (3.35). Explicitly, we will prove the locality for vertex operators of the form

$$
\begin{aligned}
Y_{V_\Lambda}(v, x, \bar{x}) &= \, {}^\circ_\circ \prod_{r=1}^{k} \prod_{s=1}^{\bar{k}} \left( \frac{1}{(m_r - 1)!} \frac{d^{m_r - 1} \alpha_r(x)}{dx^{m_r - 1}} \right) \left( \frac{1}{(\bar{m}_s - 1)!} \frac{d^{\bar{m}_s - 1} \beta_s(\bar{x})}{d\bar{x}^{\bar{m}_s - 1}} \right) Y_{V_\Lambda}(e^\lambda, x, \bar{x}) {}^\circ_\circ \\
Y_{V_\Lambda}(w, x, \bar{x}) &= \, {}^\circ_\circ \prod_{p=1}^{\ell} \prod_{q=1}^{\bar{\ell}} \left( \frac{1}{(n_p - 1)!} \frac{d^{n_p - 1} \alpha'_p(x)}{dx^{n_p - 1}} \right) \left( \frac{1}{(\bar{n}_q - 1)!} \frac{d^{\bar{n}_q - 1} \beta'_q(\bar{x})}{d\bar{x}^{\bar{n}_q - 1}} \right) Y_{V_\Lambda}(e^\lambda, x, \bar{x}) {}^\circ_\circ,
\end{aligned}
\tag{3.128}
$$

see [34] for a similar calculation. Following the exact same steps as in the proof of [33, Eq. (2.14)] with appropriate modifications, we can show that

$$\left[ \alpha'(x_1)^+, e^{\int \alpha(x_2)^- dx_2} \right] = \left( \frac{\langle \alpha, \alpha' \rangle}{x_1 - x_2} - \frac{\langle \alpha, \alpha' \rangle}{x_1} \right) e^{\int \alpha(x_1)^- dx_1}. \tag{3.129}$$

Differentiating on both the sides of (3.129) with respect to $x_1$ we obtain

$$\left[\frac{1}{s!}\frac{d^s\alpha'(x_1)^+}{dx_1^s}, \mathrm{e}^{\int \alpha(x_2)^- dx_2}\right] = (-1)^s \left(\frac{\langle\alpha,\alpha'\rangle}{(x_1-x_2)^{s+1}} - \frac{\langle\alpha,\alpha'\rangle}{x_1^{s+1}}\right) \mathrm{e}^{\int \alpha(x_1)^- dx_1}. \quad (3.130)$$

Differentiating both sides of (3.105) with respect to $x_2$ we obtain

$$\left[\frac{1}{s!}\frac{d^s\alpha'(x_2)^-}{dx_2^s}, \mathrm{e}^{\int \alpha(x_1)^+ dx_1}\right] = \frac{\langle\alpha,\alpha'\rangle}{(x_1-x_2)^{s+1}}\mathrm{e}^{\int \alpha(x_1)^+ dx_1}. \quad (3.131)$$

Analogous formula holds for $[\beta'(\bar{x}_1)^\pm, \exp\left(\int \beta(x_2)^\mp\right)]$. In addition, we need

$$\alpha(0)\mathrm{e}_{\lambda'}x^{\alpha'} = \langle\alpha,\alpha'\rangle\mathrm{e}_{\lambda'}x^{\alpha'} + \mathrm{e}_{\lambda'}x^{\alpha'}\alpha(0), \quad \lambda' = (\alpha',\beta'). \quad (3.132)$$

This follows from the following calculation: for $u \in S(\hat{\mathfrak{h}}^-)$, $\lambda' = (\alpha',\beta')$, $\lambda'' = (\alpha'',\beta'')$ we have

$$\alpha(0)\mathrm{e}_{\lambda'}x^{\alpha'}\left(u \otimes \mathrm{e}^{\lambda''}\right)$$
$$= (-1)^{\epsilon(\lambda',\lambda'')}x^{\langle\alpha',\alpha''\rangle}\langle\alpha,\alpha'+\alpha''\rangle\left(u \otimes \mathrm{e}^{\lambda'+\lambda''}\right)$$
$$= (-1)^{\epsilon(\lambda',\lambda'')}x^{\langle\alpha',\alpha''\rangle}\langle\alpha,\alpha'\rangle\left(u \otimes \mathrm{e}^{\lambda'+\lambda''}\right) + (-1)^{\epsilon(\lambda',\lambda'')}x^{\langle\alpha',\alpha''\rangle}\langle\alpha,\alpha''\rangle\left(u \otimes \mathrm{e}^{\lambda'+\lambda''}\right)$$
$$= \langle\alpha,\alpha'\rangle\,\mathrm{e}_{\lambda'}x^{\alpha'}\left(u \otimes \mathrm{e}^{\lambda''}\right) + \mathrm{e}_{\lambda'}x^{\alpha'}\alpha(0)\left(u \otimes \mathrm{e}^{\lambda''}\right).$$
$$(3.133)$$

Analogous formulas for $\beta(0)\mathrm{e}_{\lambda'}\bar{x}^{\beta'}$ is

$$\beta(0)\mathrm{e}_{\lambda'}\bar{x}^{\beta'} = \langle\beta,\beta'\rangle\,\mathrm{e}_{\lambda'}\bar{x}^{\beta'} + \mathrm{e}_{\lambda'}\bar{x}^{\beta'}\beta(0), \quad (3.134)$$

which can be proved as follows:

$$\beta(0)\mathrm{e}_{\lambda'}\bar{x}^{\beta'}\left(u \otimes \mathrm{e}^{\lambda''}\right)$$
$$= (-1)^{\epsilon(\lambda',\lambda'')}\bar{x}^{\langle\beta',\beta''\rangle}\langle\beta,\beta'+\beta''\rangle\left(u \otimes \mathrm{e}^{\lambda'+\lambda''}\right)$$
$$= \left((-1)^{\epsilon(\lambda',\lambda'')}\bar{x}^{\langle\beta',\beta''\rangle}\langle\beta,\beta'\rangle\left(u \otimes \mathrm{e}^{\lambda'+\lambda''}\right) + (-1)^{\epsilon(\lambda',\lambda'')}\bar{x}^{\langle\beta',\beta''\rangle}\langle\beta,\beta''\rangle\left(u \otimes \mathrm{e}^{\lambda'+\lambda''}\right)\right)$$
$$= \langle\beta,\beta'\rangle\,\mathrm{e}_{\lambda'}\bar{x}^{\beta'}\left(u \otimes \mathrm{e}^{\lambda''}\right) + \mathrm{e}_{\lambda'}\bar{x}^{\beta'}\beta(0)\left(u \otimes \mathrm{e}^{\lambda''}\right).$$
$$(3.135)$$

Let us now consider the product of two vertex operators, as in (3.127). Using the

normal ordering from (3.54) we have

$$
Y_{V_\Lambda}(v, x_1, \bar{x}_1) Y_{V_\Lambda}(w, x_2, \bar{x}_2) = \exp\left(\int \alpha(x_1)^-\right) \exp\left(\int \beta(\bar{x}_1)^-\right)
$$

$$
\times \, {}_\circ^\circ \prod_{r=1}^{k} \prod_{s=1}^{\bar{k}} \left( \frac{1}{(m_r - 1)!} \frac{d^{m_r - 1} \alpha_r(x_1)}{dx_1^{m_r - 1}} \right) \left( \frac{1}{(\bar{m}_s - 1)!} \frac{d^{\bar{m}_s - 1} \beta_s(\bar{x}_1)}{d\bar{x}_1^{\bar{m}_s - 1}} \right) {}_\circ^\circ
$$

$$
\times \exp\left(\int \alpha(x_1)^+\right) \exp\left(\int \alpha'(x_2)^-\right) \exp\left(\int \beta(\bar{x}_1)^+\right) \exp\left(\int \beta'(\bar{x}_2)^-\right) e_\lambda x_1^\alpha \bar{x}_1^\beta
$$

$$
\times \, {}_\circ^\circ \prod_{p=1}^{\ell} \prod_{q=1}^{\bar{\ell}} \left( \frac{1}{(n_p - 1)!} \frac{d^{n_p - 1} \alpha'_p(x_2)}{dx_2^{n_p - 1}} \right) \left( \frac{1}{(\bar{n}_q - 1)!} \frac{d^{\bar{n}_q - 1} \beta'_q(\bar{x}_2)}{d\bar{x}_2^{\bar{n}_q - 1}} \right) {}_\circ^\circ
$$

$$
\times \exp\left(\int \alpha'(x_2)^+\right) \exp\left(\int \beta'(\bar{x}_2)^+\right) e_{\lambda'} x_2^{\alpha'} \bar{x}_2^{\beta'} ,
$$

(3.136)

where we have used that $e_\lambda x_1^\alpha \bar{x}_1^\beta$ commutes with the exponential of integrals and the exponential of $\alpha$ and $\beta$ commute with each other. Now, using (3.109) we get

$$
Y_{V_\Lambda}(v, x_1, \bar{x}_1) Y_{V_\Lambda}(w, x_2, \bar{x}_2) = \left(1 - \frac{x_2}{x_1}\right)^{\langle \alpha, \alpha' \rangle} \left(1 - \frac{\bar{x}_2}{\bar{x}_1}\right)^{\langle \beta, \beta' \rangle} \exp\left(\int \alpha(x_1)^-\right) \exp\left(\int \beta(\bar{x}_1)^-\right)
$$

$$
\times \, {}_\circ^\circ \prod_{r=1}^{k} \prod_{s=1}^{\bar{k}} \left( \frac{1}{(m_r - 1)!} \frac{d^{m_r - 1} \alpha_r(x_1)}{dx_1^{m_r - 1}} \right) \left( \frac{1}{(\bar{m}_s - 1)!} \frac{d^{\bar{m}_s - 1} \beta_s(\bar{x}_1)}{d\bar{x}_1^{\bar{m}_s - 1}} \right) {}_\circ^\circ
$$

$$
\times \exp\left(\int \alpha'(x_2)^-\right) \exp\left(\int \beta'(\bar{x}_2)^-\right) \exp\left(\int \alpha(x_1)^+\right) \exp\left(\int \beta(\bar{x}_1)^+\right) e_\lambda x_1^\alpha \bar{x}_1^\beta
$$

$$
\times \, {}_\circ^\circ \prod_{p=1}^{\ell} \prod_{q=1}^{\bar{\ell}} \left( \frac{1}{(n_p - 1)!} \frac{d^{n_p - 1} \alpha'_p(x_2)}{dx_2^{n_p - 1}} \right) \left( \frac{1}{(\bar{n}_q - 1)!} \frac{d^{\bar{n}_q - 1} \beta'_q(\bar{x}_2)}{d\bar{x}_2^{\bar{n}_q - 1}} \right) {}_\circ^\circ
$$

$$
\times \exp\left(\int \alpha'(x_2)^+\right) \exp\left(\int \beta'(\bar{x}_2)^+\right) e_{\lambda'} x_2^{\alpha'} \bar{x}_2^{\beta'} .
$$

(3.137)

Further, using (3.130), (3.131), (3.132), and (3.134) successively on the product in

normal order, (3.110), and the formal variable identity (2.6) we get

$$
\begin{aligned}
Y_{V_\Lambda}(v, x_1, \bar{x}_1) Y_{V_\Lambda}(w, x_2, \bar{x}_2) &= (x_1 - x_2)^{\langle \alpha, \alpha' \rangle} (\bar{x}_1 - \bar{x}_2)^{\langle \beta, \beta' \rangle} \\
&\times \exp\left(\int \alpha(x_1)^-\right) \exp\left(\int \alpha'(x_2)^-\right) \exp\left(\int \beta(\bar{x}_1)^-\right) \exp\left(\int \beta'(\bar{x}_2)^-\right) \\
&\times {}^{\circ}_{\circ} \prod_{r=1}^{k} \left[ \frac{1}{(m_r - 1)!} \frac{d^{m_r-1} \alpha_r(x_1)}{dx_1^{m_r-1}} + (-1)^{m_r-1} \left( \frac{\langle \alpha', \alpha_r \rangle}{(x_1 - x_2)^{m_r}} - \frac{\langle \alpha', \alpha_r \rangle}{x_1^{m_r}} \right) \right] \\
&\times \prod_{s=1}^{\bar{k}} \left[ \frac{1}{(\bar{m}_s - 1)!} \frac{d^{\bar{m}_s-1} \beta_s(\bar{x}_1)}{d\bar{x}_1^{\bar{m}_s-1}} + (-1)^{\bar{m}_s-1} \left( \frac{\langle \beta', \beta_s \rangle}{(\bar{x}_1 - \bar{x}_2)^{\bar{m}_s}} - \frac{\langle \beta', \beta_s \rangle}{\bar{x}_1^{\bar{m}_s}} \right) \right] {}^{\circ}_{\circ} \\
&\times {}^{\circ}_{\circ} \prod_{p=1}^{\ell} \left[ \frac{1}{(n_p - 1)!} \frac{d^{n_p-1} \alpha'_p(x_2)}{dx_2^{n_p-1}} - \frac{\langle \alpha, \alpha'_p \rangle}{(x_1 - x_2)^{n_p}} - (-1)^{n_p-1} \frac{\langle \alpha, \alpha'_p \rangle}{x_2^{n_p}} \right] \\
&\times \prod_{q=1}^{\bar{\ell}} \left[ \frac{1}{(\bar{n}_q - 1)!} \frac{d^{\bar{n}_q-1} \beta'_q(\bar{x}_2)}{d\bar{x}_2^{\bar{n}_q-1}} - \frac{\langle \beta, \beta'_q \rangle}{(\bar{x}_1 - \bar{x}_2)^{\bar{n}_q}} - (-1)^{\bar{n}_q-1} \frac{\langle \beta, \beta'_q \rangle}{\bar{x}_2^{\bar{n}_q}} \right] {}^{\circ}_{\circ} \\
&\times \exp\left(\int \alpha(x_1)^+\right) \exp\left(\int \beta(\bar{x}_1)^+\right) \exp\left(\int \alpha'(x_2)^+\right) \exp\left(\int \beta'(\bar{x}_2)^+\right) \\
&\times \mathrm{e}_\lambda \mathrm{e}_{\lambda'} x_1^\alpha \bar{x}_1^\beta x_2^{\alpha'} \bar{x}_2^{\beta'} .
\end{aligned}
\tag{3.138}
$$

Next we have

$$
\begin{aligned}
Y_{V_\Lambda}(w, x_2, \bar{x}_2) Y_{V_\Lambda}(v, x_1, \bar{x}_1) &= (x_2 - x_1)^{\langle \alpha, \alpha' \rangle} (\bar{x}_2 - \bar{x}_1)^{\langle \beta, \beta' \rangle} (-1)^{\lambda \circ \lambda'} \\
&\times \exp\left(\int \alpha(x_1)^-\right) \left(\int \alpha'(x_2)^-\right) \exp\left(\int \beta(\bar{x}_1)^-\right) \exp\left(\int \beta'(\bar{x}_2)^-\right) \\
&\times {}^{\circ}_{\circ} \prod_{p=1}^{\ell} \left[ \frac{1}{(n_p - 1)!} \frac{d^{n_p-1} \alpha'_p(x_2)}{dx_2^{n_p-1}} + (-1)^{n_p-1} \left( \frac{\langle \alpha', \alpha_p \rangle}{(x_2 - x_1)^{n_p}} - \frac{\langle \alpha', \alpha_p \rangle}{x_2^{n_p}} \right) \right] \\
&\times \prod_{q=1}^{\bar{\ell}} \left[ \frac{1}{(\bar{n}_q - 1)!} \frac{d^{\bar{n}_q-1} \beta'_q(\bar{x}_2)}{d\bar{x}_2^{\bar{n}_q-1}} + (-1)^{\bar{n}_q-1} \left( \frac{\langle \beta', \beta_q \rangle}{(\bar{x}_2 - \bar{x}_1)^{\bar{n}_q}} - \frac{\langle \beta', \beta_q \rangle}{\bar{x}_2^{\bar{n}_q}} \right) \right] {}^{\circ}_{\circ} \\
&\times {}^{\circ}_{\circ} \prod_{r=1}^{k} \left[ \frac{1}{(m_r - 1)!} \frac{d^{m_r-1} \alpha_r(x_1)}{dx_1^{m_r-1}} - \frac{\langle \alpha', \alpha_r \rangle}{(x_2 - x_1)^{m_r}} - (-1)^{m_r-1} \frac{\langle \alpha', \alpha_r \rangle}{x_1^{m_r}} \right] \\
&\times \prod_{s=1}^{\bar{k}} \left[ \frac{1}{(\bar{m}_s - 1)!} \frac{d^{\bar{m}_s-1} \beta_s(\bar{x}_1)}{d\bar{x}_1^{\bar{m}_s-1}} - \frac{\langle \beta', \beta_s \rangle}{(\bar{x}_2 - \bar{x}_1)^{\bar{m}_s}} - (-1)^{\bar{m}_s-1} \frac{\langle \beta', \beta_s \rangle}{\bar{x}_1^{\bar{m}_s}} \right] {}^{\circ}_{\circ} \\
&\times \exp\left(\int \alpha(x_2)^+\right) \exp\left(\int \beta(\bar{x}_2)^+\right) \exp\left(\int \alpha'(x_1)^+\right) \exp\left(\int \beta'(\bar{x}_1)^+\right) \\
&\times \mathrm{e}_\lambda \mathrm{e}_{\lambda'} x_1^\alpha \bar{x}_1^\beta x_2^{\alpha'} \bar{x}_2^{\beta'} ,
\end{aligned}
\tag{3.139}
$$

where we used (3.42). Now, let us take $x_1, x_2, \bar{x}_1$ and $\bar{x}_2$ to be complex numbers

$z_1, z_2, \bar{z}_1$ and $\bar{z}_2$ respectively. Then we can rewrite (3.139) as

$$
\begin{aligned}
Y_{V_\Lambda}(w, z_2, \bar{z}_2) Y_{V_\Lambda}(v, z_1, \bar{z}_1) &= (z_2 - z_1)^{\langle \alpha, \alpha' \rangle} (\bar{z}_2 - \bar{z}_1)^{\langle \beta, \beta' \rangle} (-1)^{\lambda \circ \lambda'} \\
&\times \exp\left( \int \alpha(z_1)^- \right) \left( \int \alpha'(z_2)^- \right) \exp\left( \int \beta(\bar{z}_1)^- \right) \exp\left( \int \beta'(\bar{z}_2)^- \right) \\
&\times {}^{\circ}_{\circ} \prod_{r=1}^{k} \left[ \frac{1}{(m_r - 1)!} \frac{d^{m_r - 1} \alpha_r(z_1)}{dz_1^{m_r - 1}} - \frac{\langle \alpha', \alpha_r \rangle}{(z_2 - z_1)^{m_r}} - (-1)^{m_r - 1} \frac{\langle \alpha', \alpha_r \rangle}{z_1^{m_r}} \right] \\
&\times \prod_{s=1}^{\bar{k}} \left[ \frac{1}{(\bar{m}_s - 1)!} \frac{d^{\bar{m}_s - 1} \beta_s(\bar{z}_1)}{d\bar{z}_1^{\bar{m}_s - 1}} - \frac{\langle \beta', \beta_s \rangle}{(\bar{z}_2 - \bar{z}_1)^{\bar{m}_s}} - (-1)^{\bar{m}_s - 1} \frac{\langle \beta', \beta_s \rangle}{\bar{z}_1^{\bar{m}_s}} \right] {}^{\circ}_{\circ} \\
&\times {}^{\circ}_{\circ} \prod_{p=1}^{\ell} \left[ \frac{1}{(n_p - 1)!} \frac{d^{n_p - 1} \alpha'_p(z_2)}{dz_2^{n_p - 1}} + (-1)^{n_p - 1} \left( \frac{\langle \alpha', \alpha_p \rangle}{(z_2 - z_1)^{n_p}} - \frac{\langle \alpha', \alpha_p \rangle}{z_2^{n_p}} \right) \right] \\
&\times \prod_{q=1}^{\bar{\ell}} \left[ \frac{1}{(\bar{n}_q - 1)!} \frac{d^{\bar{n}_q - 1} \beta'_q(\bar{z}_2)}{d\bar{z}_2^{\bar{n}_q - 1}} + (-1)^{\bar{n}_q - 1} \left( \frac{\langle \beta', \beta_q \rangle}{(\bar{z}_2 - \bar{z}_1)^{\bar{n}_q}} - \frac{\langle \beta', \beta_q \rangle}{\bar{z}_2^{\bar{n}_q}} \right) \right] {}^{\circ}_{\circ} \\
&\times \exp\left( \int \alpha(z_2)^+ \right) \exp\left( \int \beta(\bar{z}_2)^+ \right) \exp\left( \int \alpha'(z_1)^+ \right) \exp\left( \int \beta'(\bar{z}_1)^+ \right) \\
&\times e_\lambda e_{\lambda'} z_1^\alpha \bar{z}_1^\beta z_2^{\alpha'} \bar{z}_2^{\beta'} .
\end{aligned}
\tag{3.140}
$$

Here it is important that we understand $(z_1 - z_2)^s$ as the power series since we obtained it by replacing $x_1 \to z_1, x_2 \to z_2$ in $(x_1 - x_2)^s$ which is a formal series. In this step we have used the fact that the two normal ordered products commute. To see this, note that the normal ordered product can be written as the product without normal order plus a multiple of the central element $\mathbf{k}, \bar{\mathbf{k}}$ using (3.29). Then since $\alpha(z_1)$ and $\alpha'(z_2)$ commute by (3.99), hence their derivatives and normal ordered products commute too.

Now the operators in (3.138) and (3.139) are the same. Thus locality follows if we can show that the functions appearing in (3.138) and (3.139) are the expansions of a single smooth function in the domains $|z_1| > |z_2|$ and $|z_2| > |z_1|$ respectively. We have already proved in Proposition 3.1 that the functions $(z_1 - z_2)^{\langle \alpha, \alpha' \rangle} (\bar{z}_1 - \bar{z}_2)^{\langle \beta, \beta' \rangle}$ and $(-1)^{\lambda \circ \lambda'} (z_2 - z_1)^{\langle \alpha, \alpha' \rangle} (\bar{z}_2 - \bar{z}_1)^{\langle \beta, \beta' \rangle}$, understood as power series as explained above, are the expansion of the function

$$
\exp\left( \langle \alpha, \alpha' \rangle \log(z_1 - z_2) \right) \exp\left( \langle \beta, \beta' \rangle \log(\bar{z}_1 - \bar{z}_2) \right). \tag{3.141}
$$

It remains to prove that the functions appearing in the normal ordered products are also expansions of a single smooth function. It can easily be checked that the functions

$$
(-1)^{m_r - 1} \left( \frac{\langle \alpha', \alpha_r \rangle}{(z_1 - z_2)^{m_r}} - \frac{\langle \alpha', \alpha_r \rangle}{z_1^{m_r}} \right), \quad |z_1| > |z_2|, \tag{3.142}
$$

and

$$-\frac{\langle\alpha',\alpha_r\rangle}{(z_2-z_1)^{m_r}}-(-1)^{m_r-1}\frac{\langle\alpha',\alpha_r\rangle}{z_1^{m_r}},\quad |z_2|>|z_1|,\tag{3.143}$$

are the expansions of the function

$$(-1)^{m_r-1}\left(\frac{\langle\alpha',\alpha_r\rangle}{\exp(m_r\log(z_1-z_2))}-\frac{\langle\alpha',\alpha_r\rangle}{z_1^{m_r}}\right),\tag{3.144}$$

in the respective domains except for poles at $z_1=z_2$ and $z_1=0$. □

**Remark 3.3.** A similar calculation as in Remark 3.2 shows that formal commutativity holds for general vertex operators:

$$(x_1-x_2)^K(\bar{x}_1-\bar{x}_2)^{\bar{K}}\left[Y_{V_\Lambda}(v,x_1,\bar{x}_1),Y_{V_\Lambda}(w,x_2,\bar{x}_2)\right]=0\,,\tag{3.145}$$

with $v,w\in V_\Lambda$.

**Remark 3.4.** The proof of locality goes through even if we take vertex operators corresponding to vectors of the form (3.35) with $\mathrm{e}^\lambda\in\mathbb{C}[\Lambda]$ (see Remark 3.1 for definition of such vertex operators). This requires $\Lambda$ to be an integral Lorentzian lattice. It is worth noting that formal commutativity fails to hold for general vertex operators since $\langle\alpha,\alpha\rangle,\langle\beta,\beta\rangle\notin\mathbb{Z}$ in general.

The graded dimension of the LLVOA can be easily computed. Using the structure of the vector space $V_\Lambda$ and the general discussion in [2, Section 1.10], we find that

$$\chi_{V_\Lambda}(\tau,\bar{\tau})=\frac{1}{\eta(\tau)^m\overline{\eta(\tau)}^n}\sum_{(\alpha,\beta)\in\Lambda_0}q^{\frac{\langle\alpha,\alpha\rangle}{2}}\bar{q}^{\frac{\langle\beta,\beta\rangle}{2}},\tag{3.146}$$

where $\eta(\tau)$ is the Dedekind eta function

$$\eta(\tau)=q^{\frac{1}{24}}\prod_{n=1}^{\infty}(1-q^n).\tag{3.147}$$

We now give explicit examples of isomorphisms and automorphisms of the LLVOA.

**Theorem 3.4.** *Let* $(V_\Lambda,Y_\Lambda,\omega_L,\omega_R,\mathbf{1}_{V_\Lambda})$ *and* $(V_{\tilde{\Lambda}},Y_{\tilde{\Lambda}},\tilde{\omega}_L,\tilde{\omega}_R,\mathbf{1}_{V_{\tilde{\Lambda}}})$ *be LLVOAs corresponding to lattices* $\Lambda,\tilde{\Lambda}\subset\mathbb{R}^{m,n}$. *Suppose* $\Lambda$ *and* $\tilde{\Lambda}$ *are related by an* $\mathrm{O}(m,\mathbb{R})\times\mathrm{O}(n,\mathbb{R})$*-transformation, then the two LLVOAs are isomorphic* $(V_\Lambda,Y_\Lambda)\cong(V_{\tilde{\Lambda}},Y_{\tilde{\Lambda}})$.

*Proof.* Suppose $f:\Lambda\longrightarrow\tilde{\Lambda}$ is the isomorphism relating $\Lambda$ and $\tilde{\Lambda}$, then for any $\lambda=(\alpha^\lambda,\beta^\lambda)\in\Lambda$

$$f(\alpha^\lambda,\beta^\lambda)=(O_1\cdot\alpha^\lambda,O_2\cdot\beta^\lambda),\tag{3.148}$$

where $O_1$ and $O_2$ lie in $O(m,\mathbb{R})$ and $O(n,\mathbb{R})$ respectively. Further from the action (3.148), it is clear that $f(\Lambda_1^0)=\tilde{\Lambda}_1^0$ and $f(\Lambda_2^0)=\tilde{\Lambda}_2^0$ and further that the restrictions to $\Lambda_1^0,\Lambda_2^0$ are norm-preserving isomorphisms. Using $f$ we can define the maps

$$f_i:\Lambda_i\longrightarrow\tilde{\Lambda}_i,\quad i=1,2,$$
$$\alpha^\lambda\mapsto O_1\cdot\alpha^\lambda,\quad\beta^\lambda\mapsto O_2\cdot\beta^\lambda,\tag{3.149}$$

which are norm-preserving maps when we consider $\Lambda_1$ and $\tilde{\Lambda}_1$ ($\Lambda_2$ and $\tilde{\Lambda}_2$) as subspaces of $\mathbb{R}^m$ ($\mathbb{R}^n$).

Since an integral basis of $\Lambda$ and $\tilde{\Lambda}$ is also a basis of $\mathbb{R}^{m,n}$, it is clear that the dimension

$$\dim(\mathfrak{h}_1) = \dim(\tilde{\mathfrak{h}}_1) = m, \ \dim(\mathfrak{h}_2) = \dim(\tilde{\mathfrak{h}}_2) = n \,, \tag{3.150}$$

where

$$\mathfrak{h}_i = \Lambda_i \otimes_{\mathbb{Z}} \mathbb{C}, \quad \tilde{\mathfrak{h}}_i = \tilde{\Lambda}_i \otimes_{\mathbb{Z}} \mathbb{C}, \quad i = 1, 2 \,, \tag{3.151}$$

this implies that the central charges of the two LLVOAs are the same. We then extend $f_1 : \mathfrak{h}_1 \to \tilde{\mathfrak{h}}_1$ and $f_2 : \mathfrak{h}_2 \to \tilde{\mathfrak{h}}_2$ by $\mathbb{C}$-linearity and observe that the bilinear form on $\mathfrak{h}_i$'s are preserved under this map. We then extend $f_1$ and $f_2$ to $\hat{\mathfrak{h}}_1, \hat{\mathfrak{h}}_2$, by mapping $\mathbf{k}$ and $\bar{\mathbf{k}}$ back to themselves.

Consider the orthonormal bases, which were chosen when defining the conformal vectors $\omega_L, \omega_R, \tilde{\omega}_L$, and $\tilde{\omega}_R$, i.e. $\{u_i\}_{i=1}^m, \{v_i\}_{i=1}^n$ and $\{\tilde{u}_i\}_{i=1}^m, \{\tilde{v}_i\}_{i=1}^n$ of $\mathfrak{h}_1, \tilde{\mathfrak{h}}_1$ and $\mathfrak{h}_2, \tilde{\mathfrak{h}}_2$ respectively so that

$$\begin{aligned}
\omega_L &= \frac{1}{2} \sum_{i=1}^m \left( u_i(-1)^2 \right) \otimes \mathbf{1}_{V_\Lambda}, \quad \omega_R = \frac{1}{2} \sum_{i=1}^n \left( v_i(-1)^2 \right) \otimes \mathbf{1}_{V_\Lambda} \\
\tilde{\omega}_L &= \frac{1}{2} \sum_{i=1}^m \left( \tilde{u}_i(-1)^2 \right) \otimes \mathbf{1}_{V_{\tilde{\Lambda}}}, \quad \tilde{\omega}_R = \frac{1}{2} \sum_{i=1}^n \left( \tilde{v}_i(-1)^2 \right) \otimes \mathbf{1}_{V_{\tilde{\Lambda}}}.
\end{aligned} \tag{3.152}$$

Define the isomorphism of complex vector spaces

$$\begin{aligned}
\mathfrak{h}_1 &\longrightarrow \tilde{\mathfrak{h}}_1, \quad \mathfrak{h}_2 \longrightarrow \tilde{\mathfrak{h}}_2 \\
u_i &\mapsto \tilde{u}_i, \quad v_j \mapsto \tilde{v}_j, \quad i = 1, \ldots, m, \quad j = 1, \ldots, n.
\end{aligned} \tag{3.153}$$

Denote by $\tilde{\alpha} \in \tilde{\mathfrak{h}}_1, \tilde{\beta} \in \tilde{\mathfrak{h}}_2$ the image of $\alpha \in \mathfrak{h}_1, \beta \in \mathfrak{h}_2$ under the above map. Define the map $\psi : V_\Lambda \longrightarrow V_{\tilde{\Lambda}}$ by

$$\begin{aligned}
\psi\big(&\alpha_1(-m_1) \cdot \alpha_2(-m_2) \cdots \alpha_k(-m_k)\, \beta_1(-\bar{m}_1) \cdot \beta_2(-\bar{m}_2) \cdots \beta_{\bar{k}}(-\bar{m}_{\bar{k}}) \otimes \mathrm{e}^{(\alpha,\beta)}\big) \\
&= \tilde{\alpha}_1(-m_1) \cdot \tilde{\alpha}_2(-m_2) \cdots \tilde{\alpha}_k(-m_k)\, \tilde{\beta}_1(-\bar{m}_1) \cdot \tilde{\beta}_2(-\bar{m}_2) \\
&\qquad\qquad \cdots \tilde{\beta}_{\bar{k}}(-\bar{m}_{\bar{k}}) \otimes \mathrm{e}^{(f_1(\alpha), f_2(\beta))}.
\end{aligned} \tag{3.154}$$

Clearly

$$\psi(\mathbf{1}_{V_\Lambda}) = \mathbf{1}_{V_{\tilde{\Lambda}}} \,, \quad \psi(\omega_i) = \tilde{\omega}_i \,, \ \text{where } i = L, R. \tag{3.155}$$

Since $f_i$ is norm preserving, from (3.55) it is clear that $\psi$ is grading preserving. We now check that (2.94) is satisfied. Let us first check (2.94) for $u = \mathrm{e}^{\lambda'}$ and $v$ of the

form (3.35). From the definition (2.17) we see that

$$Y_{V_\Lambda}(e^{\lambda'}, x, \bar{x})v = \left[\exp\left(-\sum_{s<0}\frac{\alpha^{\lambda'}(s)}{s}x^{-s}\right)\exp\left(-\sum_{s>0}\frac{\alpha^{\lambda'}(s)}{s}x^{-s}\right)\exp\left(-\sum_{s<0}\frac{\beta^{\lambda'}(s)}{s}\bar{x}^{-s}\right)\right.$$

$$\left.\exp\left(-\sum_{s>0}\frac{\beta^{\lambda'}(s)}{s}\bar{x}^{-s}\right)\right]e_{\lambda'}x^{\alpha^{\lambda'}}\bar{x}^{\beta^{\lambda'}}v$$

$$= (-1)^{\epsilon(\lambda',\lambda)}x^{\langle\alpha^{\lambda'},\alpha^\lambda\rangle}\bar{x}^{\langle\beta^{\lambda'},\beta^\lambda\rangle}\left[\exp\left(-\sum_{s<0}\frac{\alpha^{\lambda'}(s)}{s}x^{-s}\right)\exp\left(-\sum_{s>0}\frac{\alpha^{\lambda'}(s)}{s}x^{-s}\right)\right.$$

$$\left.\exp\left(-\sum_{s<0}\frac{\beta^{\lambda'}(s)}{s}\bar{x}^{-s}\right)\exp\left(-\sum_{s>0}\frac{\beta^{\lambda'}(s)}{s}\bar{x}^{-s}\right)\right]$$

$$\alpha_1(-m_1)\cdot\alpha_2(-m_2)\cdots\alpha_k(-m_k)\cdot\beta_1(-\bar{m}_1)\cdot\beta_2(-\bar{m}_2)\cdots\beta_{\bar{k}}(-\bar{m}_{\bar{k}})\otimes e^{\lambda'+(\alpha,\beta)}.$$

$$(3.156)$$

Now expanding the exponentials, we obtain a linear combination of terms of the form

$$\alpha^{\lambda'}(n_1)\cdot\alpha^{\lambda'}(n_2)\cdots\alpha^{\lambda'}(n_p)\cdot\beta^{\lambda'}(\bar{n}_1)\cdot\beta^{\lambda'}(\bar{n}_2)\cdots\beta^{\lambda'}(\bar{n}_{\bar{p}})$$

$$\alpha_1(-m_1)\cdot\alpha_2(-m_2)\cdots\alpha_k(-m_k)\cdot\beta_1(-\bar{m}_1)\cdot\beta_2(-\bar{m}_2)\cdots\beta_{\bar{k}}(-\bar{m}_{\bar{k}})\otimes e^{\lambda'+(\alpha,\beta)}$$

$$(-1)^{\epsilon(\lambda',\lambda)}x^{\langle\alpha^{\lambda'},\alpha^\lambda\rangle}\bar{x}^{\langle\beta^{\lambda'},\beta^\lambda\rangle}x^\ell\bar{x}^{\bar{\ell}}\,,$$

$$(3.157)$$

with $n_i, \bar{n}_i \in \mathbb{Z}, p, \bar{p} \geq 0$ and the sum is over $\ell, \bar{\ell}$ with

$$\ell = -\sum_{i=1}^p n_i, \quad \ell = -\sum_{i=1}^{\bar{p}}\bar{n}_i\,. \tag{3.158}$$

Now we can use the Heisenberg algebra (3.74) to commute the operators $\alpha^{\lambda'}(n_i), \beta^{\lambda'}(\bar{n}_i)$ with $n_j, \bar{n}_j > 0$ past other operators and then annihilate $e^{\lambda'+(\alpha,\beta)}$. The result will be a vector of the form (3.35) with some factors of the form $\langle\alpha^{\lambda'},\alpha^{\lambda'}\rangle, \langle\alpha^{\lambda'},\alpha_i\rangle$ and $\langle\beta^{\lambda'},\beta^{\lambda'}\rangle, \langle\beta^{\lambda'},\beta_j\rangle$. Thus under the map $\psi$ in (3.154), we see that $\psi(Y_{V_\Lambda}(e^{\lambda'}, x, \bar{x})v)$ is a linear combination of terms as on the right hand side of (3.154) with factors of the form

$$(-1)^{\epsilon(\lambda',\lambda)}x^{\langle\alpha^{\lambda'},\alpha^\lambda\rangle}\bar{x}^{\langle\beta^{\lambda'},\beta^\lambda\rangle}, \quad \langle\alpha^{\lambda'},\alpha^{\lambda'}\rangle, \quad \langle\alpha^{\lambda'},\alpha_i\rangle, \quad \langle\beta^{\lambda'},\beta^{\lambda'}\rangle, \quad \langle\beta^{\lambda'},\beta_j\rangle$$

$$= (-1)^{\epsilon(\lambda',\lambda)}x^{\langle f_1(\alpha^{\lambda'}),f_1(\alpha^\lambda)\rangle}\bar{x}^{\langle f_2(\beta^{\lambda'}),f_2(\beta^\lambda)\rangle}, \quad \langle f_1(\alpha^{\lambda'}),f_1(\alpha^{\lambda'})\rangle, \quad \langle f_1(\alpha^{\lambda'}),f_1(\alpha_i)\rangle$$

$$\langle f_2(\beta^{\lambda'}),f_2(\beta^{\lambda'})\rangle, \quad \langle f_2(\beta^{\lambda'}),f_2(\beta_j)\rangle,$$

$$(3.159)$$

where by equality we mean the first term in L.H.S is equal to the first term on R.H.S, and so on. Further, we used the fact that $f_1, f_2$ are norm preserving maps to

show this equality. Consider now the LLVOA obtained from the lattice $\tilde{\Lambda}$ but with the central extension $\hat{\tilde{\Lambda}}$ of $\tilde{\Lambda}$ constructed from the cocycle $\tilde{\epsilon}$ using the integral basis $\{f(\lambda_i)\}_{i=1}^{m+n}$ of $\tilde{\Lambda}$ where $\{\lambda_i\}_{i=1}^{m+n}$ is the integral basis of $\Lambda$ used to define the cocycle for $\hat{\Lambda}$. By Proposition A.2, central extensions corresponding to cocycles defined using different bases of $\tilde{\Lambda}$ are equivalent, thus the LLVOA constructed using those central extensions are isomorphic. Hence, we may assume that the cocycle $\tilde{\epsilon}$ for the central extension $\hat{\tilde{\Lambda}}$ is defined by (3.37) using the basis $\{f(\lambda_i)\}_{i=1}^{m+n}$ of $\tilde{\Lambda}$. Now, following the same calculation as above and using the map $\psi$, it is clear that $Y_{V_{\tilde{\Lambda}}}(e^{f(\lambda')}, x, \bar{x})\psi(v)$ is given by the exact same linear combinations terms of the form of the right hand side of (3.154) as for $Y_{V_{\tilde{\Lambda}}}(e^{f(\lambda')}, x, \bar{x})\psi(v)$ but with factors

$$(-1)^{\tilde{\epsilon}(f(\lambda'),f(\lambda))} x^{\left\langle f_1(\alpha^{\lambda'}), f_1(\alpha^{\lambda}) \right\rangle} \bar{x}^{\left\langle f_2(\beta^{\lambda'}), f_2(\beta^{\lambda}) \right\rangle}, \quad \left\langle f_1(\alpha^{\lambda'}), f_1(\alpha^{\lambda}) \right\rangle, \quad \left\langle f_1(\alpha^{\lambda'}), f_1(\alpha_i) \right\rangle$$
$$\left\langle f_2(\beta^{\lambda'}), f_2(\beta^{\lambda}) \right\rangle, \quad \left\langle f_2(\beta^{\lambda'}), f_2(\beta_j) \right\rangle.$$
$$(3.160)$$

Now, consider any $\lambda' = \sum_i c_i \lambda_i$, $\lambda = \sum_i d_i \lambda_i \in \Lambda$, observe that

$$\tilde{\epsilon}(f(\lambda'), f(\lambda)) = \sum_{i,j=1}^{m+n} c_i d_j \tilde{\epsilon}(f(\lambda_i), f(\lambda_j)) = \sum_{i,j=1}^{m+n} c_i d_j \, \tilde{\epsilon}(f(\lambda_i), f(\lambda_j))$$
$$= \sum_{i,j=1}^{m+n} c_i d_j \, \epsilon(\lambda_i, \lambda_j) = \epsilon(\lambda', \lambda),$$
$$(3.161)$$

where the third equality follows from the fact that $f$ is norm-preserving. Using the fact that $\epsilon(\lambda', \lambda) = \tilde{\epsilon}(f(\lambda'), f(\lambda))$ we see that the factors in (3.160) are equal to the factors on the R.H.S of (3.159), and hence

$$Y_{V_{\tilde{\Lambda}}}(\psi(e^{\lambda'}), x, \bar{x})\psi(v) = \psi(Y_{V_{\Lambda}}(e^{\lambda'}, x, \bar{x})v). \tag{3.162}$$

When we take $u$ to be a more general vector, the corresponding vertex operator has products of operators which can again be expanded and dealt with as above. This completes the proof of the proposition. $\qquad\square$

**Corollary 3.1.** *Let $f \in \text{Aut}(\Lambda)$ such that $f(\Lambda_i^0) = \Lambda_i^0$, $i = 1, 2$. Then $f$ can be extended to an automorphism of the LLVOA associated to $\Lambda$.*

*Proof.* We define the map

$$\psi(e^{\lambda}) = e^{f(\lambda)}, \quad \lambda \in \Lambda_0. \tag{3.163}$$

Then define $\psi : V_{\Lambda} \longrightarrow V_{\Lambda}$ analogous to (3.154) which acts as identity on the factors $\alpha_i(-m_i)$ and $\beta_j(-\bar{m}_j)$ and as (3.163) on $\mathbb{C}[\Lambda_0]$. It can be checked that $\psi$ defines an automorphism of the LLVOA $V_{\Lambda}$ by following the same calculation as in Theorem 3.4. $\qquad\square$

From Corollary 3.1, automorphisms of the lattice whuch preserve $\Lambda_i^0$, $i = 1, 2$ can be extended to automorphisms of the LLVOA. We then propose the following

**Conjecture 1.** Any automorphism $f \in \text{Aut}(\Lambda)$ preserves $\Lambda_i^0$, $i = 1, 2$.

We prove this conjecture for $m = n$ in Appendix D. Although we were not able to prove this conjecture for $m \neq n$ there are physical reasons to believe this conjecture: T-duality group in string theory acts by automorphism of a reference Lorentzian lattice [35]. By definition, T-duality must preserve the chiral and anti-chiral algebra of the CFT. In our formalism, the chiral and anti-chiral algebra is identified with the algebra of modes of the chiral and anti-chiral vertex operators of non-chiral VOA (see Table 1), thus the automorphism of the reference lattice must act as automorphism of the LLVOA. This physical consideration supports the conjecture. We will assume the truth of this conjecture and derive the moduli space of LLVOAs later in Section 5 below.

# 4 Modules and intertwining operators

## 4.1 Modules

We now define modules of a non-chiral VOA.

**Definition 4.1.** Let $(V, Y_V, \omega, \bar{\omega}, \mathbf{1})$ be a non-chiral VOA. A *module* for $V$ is a tuple $(W, Y_W)$ where $W$ is an $(\mathbb{C} \times \mathbb{C})$-graded complex vector space, $Y_W$ is a linear map, called the *module vertex operator map*,

$$
\begin{aligned}
Y_W : V \otimes W &\longrightarrow W\{x^{\pm 1}, \bar{x}^{\pm 1}\} \\
u \otimes w &\longmapsto Y_W(u, x, \bar{x})w
\end{aligned}
\tag{4.1}
$$

or equivalently a map

$$
\begin{aligned}
Y_W : \mathbb{C}^\times \times \mathbb{C}^\times &\longrightarrow \text{Hom}(V \otimes W, \overline{W}) \\
(z, \bar{z}) &\longmapsto Y_W(\cdot, z, \bar{z}) : u \otimes w \longmapsto Y_W(u, z, \bar{z})w,
\end{aligned}
\tag{4.2}
$$

which is multi-valued and analytic if $z, \bar{z}$ are independent complex variables and single valued when $\bar{z}$ is the complex conjugate of $z$. As before the vertex operator $Y_W(u, x, \bar{x})$ for $u \in V_{(h, \bar{h})}$ is expanded as a formal power series

$$
\begin{aligned}
Y_W(u, x, \bar{x}) &= \sum_{\substack{m,n \in \mathbb{C} \\ (m-n) \in \mathbb{Z}}} u_{m,n}^W x^{-m-1} \bar{x}^{-n-1} \\
&= \sum_{\substack{m,n \in \mathbb{C} \\ (m-n) \in \mathbb{Z}}} x_{m,n}^W(u) x^{-m-h} \bar{x}^{-n-\bar{h}} \in \text{End}(W)\{x^{\pm 1}, \bar{x}^{\pm 1}\}.
\end{aligned}
\tag{4.3}
$$

The following properties must be satisfied:

1. *Identity property:* The vertex operator corresponding to the vacuum vector acts as identity, i.e.
$$Y_W(\mathbf{1}, x, \bar{x})w = w, \quad \forall\, w \in W. \tag{4.4}$$

2. *Grading-restriction property:* For every[19] $(h, \bar{h}) \in \mathbb{C} \times \mathbb{C}$,
$$\dim(W_{(h,\bar{h})}) < \infty, \tag{4.5}$$
there exists $M \in \mathbb{R}$, such that
$$W_{(h,\bar{h})} = 0, \quad \text{for } \operatorname{Re}(h) < M \text{ or } \operatorname{Re}(\bar{h}) < M. \tag{4.6}$$

3. *Single-valuedness property:* For every homogenous subspace $W_{(h,\bar{h})}$:
$$h - \bar{h} \in \mathbb{Z}. \tag{4.7}$$

4. *Virasoro property:* The vertex operators $Y_W(\omega, x, \bar{x})$ and $Y_W(\bar{\omega}, x, \bar{x})$, called *conformal vertex operators*, have Laurent series in $x, \bar{x}$ given by
$$Y_W(\omega, x, \bar{x}) = \sum_{n \in \mathbb{Z}} L^W(n) x^{-n-2},$$
$$Y_W(\bar{\omega}, x, \bar{x}) = \sum_{n \in \mathbb{Z}} \bar{L}^W(n) \bar{x}^{-n-2}, \tag{4.8}$$
where $L^W(n), \bar{L}^W(n)$ are operators which satisfy the Virasoro algebra (2.27) with central charge $c, \bar{c}$ respectively.

5. *Grading property:* For $w \in W_{(h,\bar{h})}$
$$L^W(0)w = hw, \quad \bar{L}^W(0)w = \bar{h}w. \tag{4.9}$$

6. *$L^W(0)$-property :*
$$[L^W(0), Y_W(u, x, \bar{x})] = x\frac{\partial}{\partial x} Y_W(u, x, \bar{x}) + Y(L(0)u, x, \bar{x}),$$
$$[\bar{L}^W(0), Y_W(u, x, \bar{x})] = \bar{x}\frac{\partial}{\partial \bar{x}} Y_W(u, x, \bar{x}) + Y_W(\bar{L}(0)u, x, \bar{x}). \tag{4.10}$$

7. *Translation property:* For any $u \in V$
$$\left[L^W(-1), Y_W(u, x, \bar{x})\right] = Y_W\left(L(-1)u, x, \bar{x}\right) = \frac{\partial}{\partial x} Y_W(u, x, \bar{x}),$$
$$\left[\bar{L}^W(-1), Y_W(u; x, \bar{x})\right] = Y_W\left(\bar{L}(-1)u, x, \bar{x}\right) = \frac{\partial}{\partial \bar{x}} Y_W(u, x, \bar{x}). \tag{4.11}$$

---

[19]Note that $h, \bar{h}$ are *not* complex conjugates of each other. We will explicitly specify this when this is the case.

8. *Locality and Duality property:* The module vertex operators must be local, that is given $n$ module vertex operators $Y_W(u_i, z_i, \bar{z}_i)$, $i = 1, \ldots, n$, there exists an operator-valued function $m_n(u_1, \ldots, u_n, z_1, \ldots, z_n, \bar{z}_1, \ldots, \bar{z}_n)$ satisfying the requirements in Property 9 of Definition 2.1. Moreover, for $u_1, u_2 \in V$,

$$
\begin{aligned}
& Y_W\left(u_1, z_1, \bar{z}_1\right) Y_W\left(u_2, z_2, \bar{z}_2\right), \\
& Y_W\left(u_2, z_2, \bar{z}_2\right) Y_W\left(u_1, z_1, \bar{z}_1\right), \\
& Y_W\left(Y_V\left(u_1, z_1 - z_2, \bar{z}_1 - \bar{z}_2\right) u_2, z_2, \bar{z}_2\right),
\end{aligned}
\tag{4.12}
$$

are the expansions of a function $m\left(u_1, u_2, z_1, \bar{z}_1, z_2, \bar{z}_2\right)$ in the sets given by $|z_1| > |z_2| > 0$, $|z_2| > |z_1| > 0$, and $|z_2| > |z_1 - z_2| > 0$, respectively, where $\bar{z}_1, \bar{z}_2$ are the complex conjugates of $z_1$ and $z_2$ respectively. Also $m$ is an End$(W)$-valued function, linear in $u_1, u_2$, defined on

$$
\left\{(z_1, z_2) \in \mathbb{C}^2 \,|\, z_1, z_2 \neq 0, z_1 \neq z_2\right\},
\tag{4.13}
$$

multi-valued and analytic when $\bar{z}_1, \bar{z}_2$ are viewed as independent variables and is single-valued when $\bar{z}_1, \bar{z}_2$ are equal to the complex conjugates of $z_1, z_2$ respectively. We say that the module vertex operators $Y_W(u_1, z_1, \bar{z}_1)$ and $Y_W(u_2, z_2, \bar{z}_2)$ satisfy locality and duality with respect to each other if they satisfy (4.12).

**Remark 4.1.** The module for non-chiral VOA defined here is related to the notion of *module* in [2] and [9], and *ordinary module* in [36].

**Remark 4.2.** The equality of only the first two expressions of (4.12) is the usual locality of two module vertex operators and the equality of first and third expressions in (4.12) is called duality of module vertex operators. From Proposition 2.2, we see that locality implies duality for vertex operators of a non-chiral VOA, while for module vertex operators, Proposition 4.1 below gives a sufficient condition for locality to imply duality in terms of existence of a certain intertwining operator (see Definition 4.2).

Chiral and anti-chiral module vertex operators are defined analogous to chiral and anti-chiral vertex operators. For $v \in V_\Lambda$ with conformal weights $(h, \bar{h})$, we will expand the module vertex operator $Y_W(v, x, \bar{x})$ as in (4.3). As in Lemma 2.4, for chiral and anti-chiral vectors $u \in V_{(h, \bar{h})}, v \in V_{(h', \bar{h}')}$, we expand the module vertex operators as

$$
\begin{aligned}
Y_W(u, x) &= \sum_{m \in \mathbb{Z}} x_m^W(u) x^{-m-(h-\bar{h})}, \\
Y_W(v, \bar{x}) &= \sum_{m \in \mathbb{Z}} \bar{x}_m^W(v) \bar{x}^{-m-(\bar{h}'-h')}.
\end{aligned}
\tag{4.14}
$$

The proof of Theorem 2.1 goes through even for module vertex operators. We record the result for later reference.

**Theorem 4.1.** *Let $u_i \in V_{(h_i, \bar{h}_i)}$ and $v_i \in V_{(h_i', \bar{h}_i')}$ be homogeneous chiral and anti-chiral vectors respectively with corresponding vertex operators*

$$Y_W(u_i, x) = \sum_{n \in \mathbb{Z}} x_n^W(u_i) x^{-n-(h_i - \bar{h}_i)},$$

$$Y_W(v_j, \bar{x}) = \sum_{n \in \mathbb{Z}} \bar{x}_n^W(v_i) \bar{x}^{-n-(\bar{h}_i' - h_i')}. \tag{4.15}$$

*Then we have*

$$[x_n^W(u_i), x_k^W(u_j)] = \sum_{p \geq -(h_i - \bar{h}_i) + 1} \binom{n + (h_i - \bar{h}_i) - 1}{p + (h_i - \bar{h}_i) - 1} x_{k+n}^W(x_p(u_i) \cdot u_j),$$

$$[\bar{x}_n^W(v_i), \bar{x}_k^W(v_j)] = \sum_{p \geq -(\bar{h}_i' - h_i') + 1} \binom{n + (\bar{h}_i' - h_i') - 1}{p + (\bar{h}_i' - h_i') - 1} \bar{x}_{k+n}^W(\bar{x}_p(v_i) \cdot v_j), \tag{4.16}$$

$$[x_n^W(u_i), \bar{x}_k^W(v_j)] = 0.$$

*In particular,*

$$[L^W(n), x_k^W(u_i)] = \sum_{p \geq -1} \binom{n+1}{p+1} x_{k+n}^W(L^W(p) \cdot u_i),$$

$$[L^W(n), \bar{x}_k^W(v_i)] = 0,$$

$$[\bar{L}^W(n), \bar{x}_k^W(v_i)] = \sum_{p \geq -1} \binom{n+1}{p+1} \bar{x}_{k+n}^W(\bar{L}^W(p) \cdot v_i), \tag{4.17}$$

$$[\bar{L}^W(n), x_k^W(u_i)] = 0.$$

*More generally, for $m \in \mathbb{Z}$ we have the Borcherd's identity*

$$\sum_{r \geq 0} \binom{m}{r} \left( (-1)^r x_{n+m-r}^W(u_i) x_{k+r}^W(u_j) - (-1)^{m+r} x_{k+m-r}^W(u_j) x_{n+r}^W(u_i) \right)$$

$$= \sum_{p \geq 1 - (h_i - \bar{h}_i)} \binom{n + (h_i - \bar{h}_i) - 1}{p + (h_i - \bar{h}_i) - 1} x_{k+n+m+\bar{h}_i - \bar{h}_j}^W(x_{p+m}(u_i) \cdot u_j), \tag{4.18}$$

$$\sum_{r \geq 0} \binom{m}{r} \left( (-1)^r \bar{x}_{n+m-r}^W(v_i) \bar{x}_{k+r}^W(v_j) - (-1)^{m+r} \bar{x}_{k+m-r}^W(v_j) \bar{x}_{n+r}^W(v_i) \right)$$

$$= \sum_{p \geq 1 - (\bar{h}_i' - h_i')} \binom{n + (\bar{h}_i' - h_i') - 1}{p + (\bar{h}_i' - h_i') - 1} \bar{x}_{k+n+m+h_i' - h_j'}^W(\bar{x}_{p+m}(v_i) \cdot v_j), \tag{4.19}$$

$$\sum_{r \geq 0} \binom{m}{r} \left( (-1)^r x_{n+m-r}^W(u_i) \bar{x}_{k+r}^W(v_j) - (-1)^{m+r} \bar{x}_{k+m-r}^W(v_j) x_{n+r}^W(u_i) \right) = 0. \tag{4.20}$$

The graded dimension or character of a module $W$ of a non-chiral VOA is defined similar to that of the VOA:

$$\chi_W(\tau, \bar{\tau}) = \operatorname{Tr}_W q^{L^W(0) - \frac{c}{24}} \bar{q}^{\bar{L}^W(0) - \frac{\bar{c}}{24}} = \sum_{(h,\bar{h}) \in \mathbb{C} \times \mathbb{C}} \left(\dim W_{(h,\bar{h})}\right) q^{h - \frac{c}{24}} \bar{q}^{\bar{h} - \frac{\bar{c}}{24}}. \quad (4.21)$$

Let $(W, Y_W)$ be a module of a non-chiral VOA $V$. A $V$-submodule of $W$ is a vector subspace $W_1 \subset W$ such that the vertex operator map restricts to a map on $W_1$:

$$\begin{aligned} Y_W : V \otimes W_1 &\longrightarrow W_1\{x, \bar{x}\} \\ u \otimes w &\longmapsto Y_W(u, x, \bar{x})w \end{aligned} \quad (4.22)$$

and is a $V$-module in its own right. A $V$-module is called *irreducible* if it has no non-zero proper submodules. Irreducible modules are also called *simple* modules. Direct sum of two $V$-modules is another $V$-module with the obvious definition of vertex operator map. A homomorphism between two $V$-modules $(W_1, Y_{W_1})$ and $(W_2, Y_{W_2})$ is a grading preserving linear map $f : W_1 \longrightarrow W_2$ satisfying

$$f(Y_{W_1}(v, x, \bar{x})w) = Y_{W_2}(v, x, \bar{x})f(w), \quad \forall \, v \in V, w \in W_1. \quad (4.23)$$

The notion of isomorphisms and automorphisms are defined analogous to the non-chiral VOA. Again, isomorphic modules have identical graded dimension. A *semi-simple* $V$-module is a $V$-module isomorphic to the direct sum of finitely many simple $V$-modules.

## 4.2 Intertwining operators

In this section, we define intertwining operators and study some of their properties.

**Definition 4.2.** Let $(V, Y_V, \omega, \bar{\omega}, \mathbf{1})$ be a non-chiral vertex operator algebra and let $(W_i, Y_i)$, $(W_j, Y_j)$ and $(W_k, Y_k)$ be three $V$-modules. An *intertwining operator* of type $\begin{pmatrix} W_i \\ W_j W_k \end{pmatrix}$ is a linear map

$$\begin{aligned} \mathcal{Y} : W_j \otimes W_k &\longrightarrow W_i\{x, \bar{x}\} \\ w_{(j)} \otimes w_{(k)} &\mapsto \mathcal{Y}(w_{(j)}, x, \bar{x})w_{(k)}, \end{aligned} \quad (4.24)$$

or equivalently a map

$$\begin{aligned} \mathcal{Y} : \mathbb{C}^\times \times \mathbb{C}^\times &\longrightarrow \operatorname{Hom}(W_j \otimes W_k, \overline{W}_i) \\ (z, \bar{z}) &\longmapsto \mathcal{Y}(\cdot, z, \bar{z}) : w_{(j)} \otimes w_{(k)} \longmapsto \mathcal{Y}(w_{(j)}, z, \bar{z})w_{(k)}, \end{aligned} \quad (4.25)$$

which is multi-valued and analytic if $z, \bar{z}$ are independent complex variables and single valued when $\bar{z}$ is the complex conjugate of $z$. The intertwining operator $\mathcal{Y}(w_{(j)}, x, \bar{x})$ is expanded as

$$\mathcal{Y}(w_{(j)}, x, \bar{x}) = \sum_{n,m \in \mathbb{C}} (w_{(j)})_{n,m} x^{-n-1} \bar{x}^{-m-1} \in \operatorname{Hom}(W_k, W_i)\{x, \bar{x}\}. \quad (4.26)$$

The following properties must be satisfied:

1. *L(0)-property:* For any $w_{(j)} \in W_j$

$$[L(0), \mathcal{Y}(w_{(j)}, x, \bar{x})] = x\frac{\partial}{\partial x}\mathcal{Y}(w_{(j)}, x, \bar{x}) + Y_j(L^{W_j}(0)w_{(j)}, x, \bar{x}),$$
$$[\bar{L}(0), \mathcal{Y}(w_{(j)}, x, \bar{x})] = \bar{x}\frac{\partial}{\partial \bar{x}}\mathcal{Y}(w_{(j)}, x, \bar{x}) + Y_j(\bar{L}^{W_j}(0)w_{(j)}, x, \bar{x}),$$

(4.27)

where the commutator on the LHS is understood to be

$$[L(0), \mathcal{Y}(w_{(j)}, x, \bar{x})] = L^{W_i}(0)\mathcal{Y}(w_{(j)}, x, \bar{x}) - \mathcal{Y}(w_{(j)}, x, \bar{x})L^{W_k}(0),$$
$$[\bar{L}(0), \mathcal{Y}(w_{(j)}, x, \bar{x})] = \bar{L}^{W_i}(0)\mathcal{Y}(w_{(j)}, x, \bar{x}) - \mathcal{Y}(w_{(j)}, x, \bar{x})\bar{L}^{W_k}(0).$$

(4.28)

2. *Translation property:* For any $w_{(j)} \in W_j$

$$[L(-1), \mathcal{Y}(w_{(j)}, x, \bar{x})] = \mathcal{Y}\left(L^{W_j}(-1)w_{(j)}, x, \bar{x}\right) = \frac{\partial}{\partial x}\mathcal{Y}(w_{(j)}, x, \bar{x}),$$
$$[\bar{L}(-1), \mathcal{Y}(w_{(j)}, x, \bar{x})] = \mathcal{Y}\left(\bar{L}^{W_j}(-1)w_{(j)}, x, \bar{x}\right) = \frac{\partial}{\partial \bar{x}}\mathcal{Y}(w_{(j)}, x, \bar{x})$$

(4.29)

where the commutativity is understood as above.

3. *Locality property:* The module vertex operators and the intertwiner must be local, that is given vectors $u_1, \ldots, u_{n-1} \in V, w_{(j)} \in W_j$, there exists an operator-valued function $m_n(u_1, \ldots, u_{n-1}, w_{(j)}, z_1, \ldots, z_n, \bar{z}_1, \ldots, \bar{z}_n)$ satisfying the requirements in Property 9 of Definition 2.1. Here, the product of vertex operators in (2.33) is replaced by

$$Y_i\left(u_{\sigma(1)}, z_{\sigma(1)}, \bar{z}_{\sigma(1)}\right) \cdots Y_i\left(u_{\sigma(a-1)}, z_{\sigma(a-1)}, \bar{z}_{\sigma(a-1)}\right) \mathcal{Y}\left(w_{(j)}, z_a, \bar{z}_a\right)$$
$$Y_k\left(u_{\sigma(a+1)}, z_{\sigma(a+1)}, \bar{z}_{\sigma(a+1)}\right) \cdots Y_k\left(u_{\sigma(n)}, z_{\sigma(n)}, \bar{z}_{\sigma(n)}\right) .$$

(4.30)

We will denote the intertwining operator by

$$\mathcal{Y}^i_{jk} \text{ or } \mathcal{Y}^{W_i}_{W_j W_k},$$

when we need to indicate its type.

**Remark 4.3.** The vertex operator map $Y_V(\cdot, x, \bar{x})$ acting on a non-chiral VOA $V$ is an example of an intertwining operator of type $\binom{V}{VV}$ and $Y_W(\cdot, x, \bar{x})$ acting on a $V$-module $W$ is an example of an intertwining operator of type $\binom{W}{VW}$.

**Remark 4.4.** Following the proof of (2.34) and using the $L(0)$-property 1 along with the grading-restriction property 2 of modules, one can show the following *lower truncation property* for intertwiners: for $w_{(j)} \in W_j$ and $w_{(k)} \in W_k$,

$$(w_{(j)})_{n,m}w_{(k)} = 0 \text{ for } n, m \text{ sufficiently large}.$$

(4.31)

**Proposition 4.1.** *Let $(V, Y_V)$ be a non-chiral VOA and $(W, Y_W)$ be a $V$-module. Suppose there exists an intertwining operator of type $\mathcal{Y}_{WV}^W$ where $(V, Y_V)$ is considered as a module for itself. Suppose further that the intertwining operator satisfies*

$$\lim_{x, \bar{x} \to 0} \mathcal{Y}_{WV}^W(w, x, \bar{x})\mathbf{1} = w, \quad w \in W. \tag{4.32}$$

*Then the locality property of module vertex operators implies the duality property (see Remark 4.2 for terminology):*

$$Y_W(u, z_1, \bar{z}_1)Y_W(v, z_2, \bar{z}_2) = Y_W(Y_V(u, z_1 - z_2, \bar{z}_1 - \bar{z}_2)v, z_2, \bar{z}_2), \quad u, v \in V. \tag{4.33}$$

*In particular, we have the OPE:*

$$Y_W(u, z_1, \bar{z}_1)Y_W(v, z_2, \bar{z}_2) = \sum_{m,n \in \mathbb{C}} Y_W(u_{m,n} \cdot v, z_2, \bar{z}_2)(z_1 - z_2)^{-m-1}(\bar{z}_1 - \bar{z}_2)^{-n-1}, \tag{4.34}$$

*where $u_{m,n} \in \mathrm{End}(V)$ defined using the expansion (2.17).*

*Proof.* The proof is analogous to the proof of Proposition 2.2. Let $w \in W$ be an arbitrary vector. First note that (4.32) along with the translation property 2 implies that Lemma 2.1 is true for the intertwiner $\mathcal{Y}_{WV}^W(w, x, \bar{x})$:

$$\mathcal{Y}_{WV}^W(w, x, \bar{x})\mathbf{1} = e^{\bar{x}\,\bar{L}(-1)}e^{x\,L(-1)}w. \tag{4.35}$$

Then we have

$$
\begin{aligned}
Y_W(u, z_1, \bar{z}_1)&Y_W(v, z_2, \bar{z}_2)e^{\bar{z}_3\,\bar{L}(-1)}e^{z_3\,L(-1)}w \\
&= Y_W(u, z_1, \bar{z}_1)Y_W(v, z_2, \bar{z}_2)\mathcal{Y}_{WV}^W(w, z_3, \bar{z}_3)\mathbf{1} \\
&= \mathcal{Y}_{WV}^W(w, z_3, \bar{z}_3)Y_V(u, z_1, \bar{z}_1)Y_V(v, z_2, \bar{z}_2)\mathbf{1} \\
&= \mathcal{Y}_{WV}^W(w, z_3, \bar{z}_3)Y_V\left(Y_V(u, z_1 - z_2, \bar{z}_1 - \bar{z}_2)v, z_2, \bar{z}_2\right)\mathbf{1} \\
&= Y_W\left(Y_V(u, z_1 - z_2, \bar{z}_1 - \bar{z}_2)v, z_2, \bar{z}_2\right)\mathcal{Y}_{WV}^W(w, z_3, \bar{z}_3)\mathbf{1},
\end{aligned} \tag{4.36}
$$

where we used the locality property of intertwiner $\mathcal{Y}_{WV}^W$ and the duality of vertex operators in Proposition 2.2. Now taking the limit $z_3, \bar{z}_3 \to 0$ gives the required result. $\qquad\square$

### 4.3 Non-chiral CFT and modular invariance

Given a non-chiral VOA $(V, Y_V)$ with the set of all isomorphism classes of simple modules $\{(W_i, Y_{W_i})\}$, one can construct a non-chiral CFT by taking the non-chiral VOA and a subset of the simple modules[20]. Note that one is allowed to choose copies of the same module.

---

[20]A crucial requiremnt in choosing what subset of simple modules to include is to make sure that the OPE of appropriate intertwiners between modules closes in the sense that the right hand side of the OPE contains interwiners and vertex operators for modules which are included in the subset we choose. We will explore these *fusion rules* for intertwiners in a future work. For the purposes of this paper, we will work with the simplistic definition given below.

**Definition 4.3.** A non-chiral VOA $(V, Y_V)$ along with a subset $\{(W_\alpha, Y_{W_\alpha})\}_{\alpha \in I}$ of simple modules, with possibly $(W_\alpha, Y_{W_\alpha}) \cong (W_\beta, Y_{W_\beta})$ for some $\alpha, \beta \in I$, will be called a non-chiral CFT.

Two non-chiral CFTs are said to be equivalent if the underlying non-chiral VOAs and their simple modules are isomorphic. Note that the isomorphism is allowed to permute the (non-trivial) modules but not the non-chiral VOA which is considered as a module for itself.

Let $\{W_\alpha\}_{\alpha \in I}$ with $W_0 \cong V$ being the non-chiral VOA $V$ considered as a module for itself be a non-chiral CFT. The *torus partition function* of the non-chiral CFT is defined by

$$Z_V(\tau, \bar{\tau}) := \sum_{i \in I} \chi_{W_i}(\tau, \bar{\tau}). \tag{4.37}$$

It is clear that equivalent non-chiral CFTs have identical partition function.

A non-chiral CFT is called *modular invariant* if its torus partition function is modular invariant:

$$Z_V(\gamma\tau, \gamma\bar{\tau}) = Z_V(\tau, \bar{\tau}), \quad \gamma \in \mathrm{SL}(2, \mathbb{Z}), \tag{4.38}$$

where

$$\gamma\tau = \frac{a\tau + b}{c\tau + d}, \quad \gamma\bar{\tau} = \frac{a\bar{\tau} + b}{c\bar{\tau} + d}, \quad \gamma = \begin{pmatrix} a & b \\ c & d \end{pmatrix} \in \mathrm{SL}(2, \mathbb{Z}). \tag{4.39}$$

Given a non-chiral CFT, its torus partition function need not be modular invariant. Modular invariance is a physical requirement and it puts strong constraints on which modules of a non-chiral VOA is allowed to construct the non-chiral CFT.

# 5 Moduli space of non-chiral CFTs over Lorentzian lattices

## 5.1 Construction of modules of LLVOA

Given any $[\mu] = [(\mu_1, \mu_2)] \in \Lambda/\Lambda_0$ be a coset. We will construct a module for the LLVOA corresponding to this coset. First observe that there is a one-to-one correspondence between cosets $\Lambda/\Lambda_0$ and $\mathbb{C}[\Lambda]/\mathbb{C}[\Lambda_0]$ given by the map

$$[\mu] \mapsto \mathbb{C}[\mu + \Lambda_0] = e^\mu \cdot \mathbb{C}[\Lambda_0] = \mathrm{Span}_{\mathbb{C}}\{e^{\mu + \lambda} : \lambda \in \Lambda_0\}. \tag{5.1}$$

Using the coset $\mathbb{C}[\mu + \Lambda_0]$, define the vector space

$$W_\mu := S(\mathfrak{h}^-) \otimes \mathbb{C}[\mu + \Lambda_0]. \tag{5.2}$$

Note that $W_\mu$ is generated by elements of the form

$$w := (\alpha_1(-m_1) \cdot \alpha_2(-m_2) \cdots \alpha_k(-m_k) \, \beta_1(-\bar{m}_1) \cdot \beta_2(-\bar{m}_2) \cdots \beta_{\bar{k}}(-\bar{m}_{\bar{k}})) \otimes e^{(\mu_1, \mu_2) + (\alpha, \beta)} \tag{5.3}$$

for $m_i, \bar{m}_{\bar{\imath}} > 0, k, \bar{k} \geq 0, (\alpha, \beta) \in \Lambda_0$. The vertex operator map is exactly the same as for $(Y_{V_\Lambda}, V_\Lambda)$:

$$Y_{W_\mu}(\cdot, x, \bar{x}) = Y_{V_\Lambda}(\cdot, x, \bar{x}). \tag{5.4}$$

Note that $Y_{W_\mu}(\cdot, x, \bar{x})$ acts on $W_\mu$ for which the required action of $\hat{\mathfrak{h}}^0, \mathbf{k}, \bar{\mathbf{k}}$ on $\mathbb{C}[\mu + \Lambda_0]$ is defined in (3.36). The action of $x^\alpha, \bar{x}^\beta$ is defined exactly the same as in (3.47) and the action of $\hat{\Lambda}_0$ on $\mathbb{C}[\mu + \Lambda]$ is given in (3.44). The grading on $W_\mu$ is given by defining the *conformal weights* of $w$ in (5.3) to be

$$h = \frac{\langle \mu_1 + \alpha, \mu_1 + \alpha \rangle}{2} + \sum_{i=1}^{k} m_i, \quad \bar{h} = \frac{\langle \mu_2 + \beta, \mu_2 + \beta \rangle}{2} + \sum_{j=1}^{\bar{k}} \bar{m}_j. \tag{5.5}$$

**Remark 5.1.** In view of (5.4), the modes $x_{m,n}^{W_\mu}(u)$ of the vertex operator $Y_{W_\mu}(u, x, \bar{x})$ is the same as the mode $x_{m,n}(u)$ of $Y_{V_\Lambda}(u, x, \bar{x})$ but now acting on $W_\mu$, see Remark 3.1.

**Theorem 5.1.** *For every $[\mu] \in \Lambda/\Lambda_0$, the tuple $(W_\mu, Y_{W_\mu})$ is a $V_\Lambda$-module.*

*Proof.* We prove the properties in Section 4 to show $(W_\mu, Y_{W_\mu})$ is a $V_\Lambda$-module. The proof of Properties 1 through 7, except Property 2 are exactly the same as in the case of the LLVOA, which we had shown in Subsection 3.3.

Now, it can be seen that $h$ and $\bar{h}$ in (5.5) are both positive numbers, as $m_i$ are positive integers and the bilinear forms, inherited from $\mathbb{R}^m$ and $\mathbb{R}^n$, on $\Lambda_1$ and $\Lambda_2$ are positive definite. Hence, $M = 0$ for Property 2. To show $\dim(W_{(h,\bar{h})}) < \infty$, we show that for any $h, \bar{h} \in \mathbb{R}$ the number of distinct $\lambda = (\alpha, \beta) \in \Lambda_0$ satisfying

$$\langle \mu_1 + \alpha, \mu_1 + \alpha \rangle \leq 2h \text{ and } \langle \mu_2 + \beta, \mu_2 + \beta \rangle \leq 2\bar{h}, \text{ where } \alpha \in \Lambda_1, \beta \in \Lambda_2, \tag{5.6}$$

can be only finitely many. We basically mimic the proof for the grading restriction property for the LLVOA, noting that $\mu_1 + \Lambda_1$ and $\mu_2 + \Lambda_2$ are also discrete.
The proof of the locality of module vertex operators is same as for the LLVOA case. We prove the duality property. We will show that there exists an intertwining operator of type $\mathcal{Y}_{W_\mu V_\Lambda}^{W_\mu}$ satisfying the hypothesis of Proposition 4.1. Indeed, for $w \in W_\mu$ of the form (5.3), consider the operator

$$\mathcal{Y}_{W_\mu V_\Lambda}^{W_\mu}(w, x, \bar{x}) = {}^\circ_\circ \prod_{r=1}^{k} \prod_{s=1}^{\bar{k}} \left( \frac{1}{(m_r - 1)!} \frac{d^{m_r - 1}\alpha_r(x)}{dx^{m_r - 1}} \right) \left( \frac{1}{(\bar{m}_s - 1)!} \frac{d^{\bar{m}_s - 1}\beta_s(\bar{x})}{d\bar{x}^{\bar{m}_s - 1}} \right) \tag{5.7}$$
$$\mathcal{Y}_{W_\mu V_\Lambda}^{W_\mu}(e^{\mu + (\alpha, \beta)}, x, \bar{x}){}^\circ_\circ,$$

where

$$\mathcal{Y}_{W_\mu V_\Lambda}^{W_\mu}(e^{\mu + (\alpha, \beta)}, x, \bar{x}) = Y_{V_\Lambda}(e^{\mu + (\alpha, \beta)}, x, \bar{x}). \tag{5.8}$$

The operators appearing in the intertwiner above act on $V_\Lambda$ in the obvious way. The axioms of intertwiners along with the hypothesis of Proposition 4.1 for $\mathcal{Y}_{W_\mu V_\Lambda}^{W_\mu}$ follows from the general proofs in Subsection 3.3. $\qquad\square$

We now show that these modules are irreducible.

**Proposition 5.1.** *Any $V_\Lambda$-module $(W, Y_W)$ is also an $(\hat{\mathfrak{h}}_1^\star \oplus \hat{\mathfrak{h}}_2^\star)$-module, where $\hat{\mathfrak{h}}_i^\star$ are Heisenberg algebras associated to $\mathfrak{h}_i$.*

*Proof.* For any $\alpha \in \mathfrak{h}_1$, consider $\alpha(-1) \otimes \mathbf{1} \in V_\Lambda$. The corresponding vertex operator is

$$Y_W(\alpha(-1) \otimes \mathbf{1}, x, \bar{x}) := \alpha^W(x) := \sum_{n \in \mathbb{Z}} \alpha^W(n) x^{-n-1}. \tag{5.9}$$

This implies that $W$ is also an $\hat{\mathfrak{h}}_1^\star$-module. Similarly considering the vector $\beta(-1) \otimes \mathbf{1} \in V_\Lambda$ and its vertex operator, we see that $W$ is also an $\hat{\mathfrak{h}}_2^\star$-module. $\square$

**Remark 5.2.** When $W = W_\mu$ is the module of the LLVOA corresponding to the coset $[\mu] \in \Lambda/\Lambda_0$, then $\alpha^W(x) = \alpha(x)$ and $\alpha^W(n) = \alpha(n)$, see Remark 5.1.

**Theorem 5.2.** *For $[\mu] \in \Lambda/\Lambda_0$, the $V_\Lambda$-module $(W_\mu, Y_{W_\mu})$ is irreducible.*

*Proof.* Suppose $W \subset W_\mu$ is a $V_\Lambda$-submodule. Then $W$ is also an $(\hat{\mathfrak{h}}_1^\star \oplus \hat{\mathfrak{h}}_2^\star)$-module. By Theorem C.2

$$W \cong S(\hat{\mathfrak{h}}_1^-) \otimes S(\hat{\mathfrak{h}}_2^-) \otimes \Omega_W \cong S(\hat{\mathfrak{h}}^-) \otimes \Omega_W, \tag{5.10}$$

where $\Omega_W$ is the vacuum space of $W$, see Appendix C for definition. Since the vacuum space of $W_\mu$ is $\mathbb{C}[\mu + \Lambda_0]$ we have

$$\Omega_W \subset \mathbb{C}[\mu + \Lambda_0] = \bigoplus_{\lambda \in \mu + \Lambda_0} \mathbb{C}e^\lambda. \tag{5.11}$$

Since $W$ is invariant under $\alpha(0), \beta(0)$ for all $\alpha \in \mathfrak{h}_1, \beta \in \mathfrak{h}_2$ and $\mathbb{C}e^\lambda$ are eigenspaces for $\alpha(0), \beta(0)$, we must have

$$\Omega_W = \mathbb{C}[M], \tag{5.12}$$

for some non-empty subspace $M \subset \mu + \Lambda_0$. Finally note that for any $\lambda \in \Lambda_0$ we have

$$e_\lambda = \exp\left(-\int dx \, \alpha^\lambda(x)^-\right) \exp\left(-\int d\bar{x} \, \beta^\lambda(\bar{x})^-\right) Y_{V_\Lambda}(e^\lambda, x, \bar{x})$$
$$\times \exp\left(-\int dx \, \alpha^\lambda(x)^+\right) \exp\left(-\int d\bar{x} \, \beta^\lambda(\bar{x})^+\right) x^{-\alpha^\lambda} \bar{x}^{-\beta^\lambda}. \tag{5.13}$$

Noting that $x^{-\alpha^\lambda}, \bar{x}^{-\beta^\lambda}$ acts as $x^{-\alpha^\lambda(0)}, \bar{x}^{-\beta^\lambda(0)}$ respectively, we see that $W$ must be invariant under $e_\lambda$ for all $\lambda \in \Lambda_0$. This means that $M = \mu + \Lambda_0$ since $\{e_\lambda : \lambda \in \Lambda_0\}$ acts transitively on $\mathbb{C}[\mu + \Lambda_0]$. $\square$

The graded dimension for the module $W_\mu$ for any $[\mu] \in \Lambda/\Lambda_0$ can be easily calculated. As for LLVOA, we obtain

$$\chi_{W_\mu}(\tau, \bar\tau) = \frac{1}{\eta(\tau)^m \overline{\eta(\tau)}^n} \sum_{(\alpha,\beta) \in \mu + \Lambda_0} q^{\frac{\langle \alpha, \alpha \rangle}{2}} \bar{q}^{\frac{\langle \beta, \beta \rangle}{2}}. \tag{5.14}$$

Using (3.146) and (5.14), we see that the partition function of the non-chiral CFT consisting of the LLVOA $(V_\Lambda, Y_{V_\Lambda})$ and its modules [21] $\{(W_\mu, Y_{W_\mu})\}_{[\mu] \in \Lambda/\Lambda_0}$ is given by

$$\begin{aligned}
Z_{V_\Lambda}(\tau, \bar\tau) &= \sum_{[\mu] \in \Lambda/\Lambda_0} \chi_{W_\mu}(\tau, \bar\tau) \\
&= \frac{1}{\eta(\tau)^m \overline{\eta(\tau)}^n} \sum_{(\alpha,\beta) \in \Lambda} q^{\frac{\langle \alpha, \alpha \rangle}{2}} \bar{q}^{\frac{\langle \beta, \beta \rangle}{2}}.
\end{aligned} \tag{5.15}$$

## 5.2 Moduli space of modular invariant non-chiral CFTs over Lorentzian lattices

Given a Lorentzian lattice $\Lambda \subset \mathbb{R}^{m,n}$, we have constructed a non-chiral vertex operator algebra based on $\Lambda$ and constructed a set of its irreducible modules. In general, these non-chiral CFTs, consisting of the LLVOA and its irreducible modules, are not modular invariant. To construct a modular invariant non-chiral CFT we restrict to even self-dual lattices and only consider the irreducible modules constructed here which are in 1-1 correspondence with the cosets $\Lambda/\Lambda_0$. Indeed, we have the following theorem.

**Theorem 5.3.** *Let $\Lambda \in \mathbb{R}^{m,n}$ be an even self-dual lattice such that $m - n \equiv 0$ mod 24. Then the non-chiral CFT consisting of the LLVOA $V_\Lambda$ and its modules $\{W_\mu\}_{[\mu] \in \Lambda/\Lambda_0}$ is a modular invariant non-chiral CFT.*

*Proof.* The partition function of the non-chiral CFT in the statement of the theorem is given by [22] (5.15):

$$\begin{aligned}
Z_{V_\Lambda}^{\mathrm{mod}}(\tau, \bar\tau) &:= \frac{1}{\eta(\tau)^m \overline{\eta(\tau)}^n} \sum_{(\alpha,\beta) \in \Lambda} q^{\frac{\langle \alpha, \alpha \rangle}{2}} \bar{q}^{\frac{\langle \beta, \beta \rangle}{2}} \\
&= \frac{1}{\eta(\tau)^m \overline{\eta(\tau)}^n} \Theta_\Lambda(\tau, \bar\tau),
\end{aligned} \tag{5.16}$$

where $\Theta_\Lambda(\tau, \bar\tau)$ is the Siegel-Narain theta function [37, 38] associated to the lattice $\Lambda$. Invariance under $\tau \to \tau + 1$:

$$Z_{V_\Lambda}^{\mathrm{mod}}(\tau + 1, \bar\tau + 1) = Z_{V_\Lambda}^{\mathrm{mod}}(\tau, \bar\tau), \tag{5.17}$$

---

[21] We stress that these are not all the modules of the LLVOA. But if we want the non-chiral CFT to be modular invariant, we need to restrict to this set of modules, see Theorem 5.3.

[22] We are using a different notation for the partition to emphasize modular invariance.

follows from $m - n \equiv 0 \bmod 24$ because

$$\eta(\tau + 1) = e^{2\pi i/24}\eta(\tau)\,, \tag{5.18}$$

Invariance under the modular transformation:

$$
\begin{aligned}
Z_{V_\Lambda}^{\mathrm{mod}}\left(-\frac{1}{\tau}, -\frac{1}{\bar\tau}\right) &= \frac{e^{-i\pi\frac{m-n}{4}}}{\sqrt{|\det \mathcal{G}_\Lambda|}}\tau^{m/2}\bar\tau^{n/2}(-i\tau)^{-m/2}(i\bar\tau)^{-n/2}\frac{1}{\eta(\tau)^m\overline{\eta(\tau)}^n}\sum_{(\alpha,\beta)\in\Lambda^\star} q^{\frac{\langle\alpha,\alpha\rangle}{2}}\bar q^{\frac{\langle\beta,\beta\rangle}{2}}\\
&= Z_{V_\Lambda}^{\mathrm{mod}}(\tau, \bar\tau)\,,
\end{aligned}
$$
$$\tag{5.19}$$

follows from the fact that $\Lambda$ is unimodular and self-dual. Here we used the modular transformation of the Dedekind eta function:

$$\eta\left(-\frac{1}{\tau}\right) = \sqrt{-i\tau}\eta(\tau)\,, \tag{5.20}$$

and the modular transformation of the Siegel-Narain theta function [37, 39]. $\qquad\square$

We now want to classify all non-chiral CFTs based on Lorentzian lattices of signature $(m, n)$ upto isomorphism. Following the physics convention, we call the set of isomorphism classes the moduli space of modular invariant non-chiral CFTs over Lorentzian lattices and denote it by $\mathcal{M}_{m,n}$.

**Theorem 5.4.** *Under the assumptions of Theorem 3.4, the non-chiral CFTs based on $\Lambda, \tilde\Lambda$ are isomorphic.*

*Proof.* Let $(W_\mu, Y_{W_\mu})_{[\mu]\in\Lambda/\Lambda_0}$ and $(\tilde W_\mu, \tilde Y_{\tilde W_\mu})_{[\mu]\in\tilde\Lambda/\tilde\Lambda_0}$ be the isomorphism classes of irreducible modules of the corresponding LLVOAs $(V_\Lambda, Y_{V_\Lambda})$ and $(V_{\tilde\Lambda}, Y_{V_{\tilde\Lambda}})$. By Theorem 3.4 the two LLVOAs are isomorphic. It now suffices to show that for $0 \neq [\mu] \in \Lambda/\Lambda_0$, there exists $0 \neq [\nu] \in \tilde\Lambda/\tilde\Lambda_0$ such that

$$(W_\mu, Y_{W_\mu}) \cong (\tilde W_\nu, \tilde Y_{\tilde W_\nu})\,. \tag{5.21}$$

Pick a representative $\mu \in [\mu]$ and let $\nu = f(\mu)$. Then define the map

$$
\begin{aligned}
\varphi : W_\mu &\longrightarrow \tilde W_\nu\\
(\alpha_1(-m_1) \cdot \alpha_2(-m_2) &\cdots \alpha_k(-m_k)\,\beta_1(-\bar m_1) \cdot \beta_2(-\bar m_2) \cdots \beta_{\bar k}(-\bar m_{\bar k})) \otimes \mathrm{e}^{(\mu_1,\mu_2)+(\alpha,\beta)}\\
&\mapsto \left(\tilde\alpha_1(-m_1) \cdot \tilde\alpha_2(-m_2) \cdots \tilde\alpha_k(-m_k)\,\tilde\beta_1(-\bar m_1) \cdot \tilde\beta_2(-\bar m_2) \cdots \tilde\beta_{\bar k}(-\bar m_{\bar k})\right)\\
&\hspace{4cm}\otimes \mathrm{e}^{(f_1(\mu_1+\alpha),f-2(\mu_2+\beta))}\,,
\end{aligned}
$$
$$\tag{5.22}$$

where $f_i : \Lambda_i \longrightarrow \tilde\Lambda_i,\quad i = 1, 2$ is defined as in (3.149) and extended to $f_i : \hat{\mathfrak{h}}_i \longrightarrow \hat{\tilde{\mathfrak{h}}}_i$ by $\mathbb{C}$-linearity. Here $\hat{\tilde{\mathfrak{h}}}_i$ is constructed as in (3.26) and (3.27) for $\tilde\Lambda$. This map is grading preserving since $f_1, f_2$ are norm-preserving on $\Lambda_1, \Lambda_2$ respectively. One can now show that $\varphi$ now defines an isomorphism of modules of isomorphic LLVOA by following the same calculations as in the proof of Theorem 3.4. $\qquad\square$

It is known that all even self-dual Lorentzian lattices of signature $(m, n)$ are related by an $O(m, n, \mathbb{R})$ transformation [32]. Thus the set of all non-chiral CFTs based on Lorentzian lattices in signature $(m, n)$ can be identified with $O(m, n, \mathbb{R})$. But in view of Theorem 5.4, many of the lattices determine isomorphic non-chiral CFTs. Moreover, for any non-chiral CFT based on $\Lambda$, from Corollary 3.1, one can identify a discrete subgroup of $O(m, n, \mathbb{R})$ which acts as automorphisms of the CFT. If we believe the truth of Conjecture 1, then we can identify this discrete subgroup of $O(m, n, \mathbb{R})$ as the automorphism group of the lattice. More precisely, from Theorem 3.2 it is easy to see that the discrete subgroup is isomorphic to $\mathcal{G}_\Lambda O_\Lambda(m, n, \mathbb{Z}) \mathcal{G}_\Lambda^{-1} \subset O(m, n, \mathbb{R})$. This subgroup, somewhat inaccurately, but conventionally [35, 40] is denoted by $O(m, n, \mathbb{Z})$. Thus we have the following theorem.

**Theorem 5.5.** *Assuming Conjecture 1, the moduli space $\mathcal{M}_{m,n}$ of modular invariant non-chiral CFTs based on Lorentzian lattices of signature $(m, n)$ is isomorphic to*

$$\mathcal{M}_{m,n} \cong \frac{O(m, n, \mathbb{R})}{O(m, \mathbb{R}) \times O(n, \mathbb{R}) \times O(m, n, \mathbb{Z})} \, , \qquad (5.23)$$

*where $O(m, \mathbb{R}) \times O(n, \mathbb{R})$ acts on $O(m, n, \mathbb{R})$ by right multiplication and $O(m, n, \mathbb{Z})$ acts by left multiplication.*

*Proof.* Choose a reference Lorentzian lattice $\Lambda_{\text{ref}}$ with generator matrix $\mathcal{G}_{\text{ref}}$. Then the set of all Lorentzian lattices can be identified with $O(m, n, \mathbb{R})$ under the map[23]

$$\mathcal{O} \mapsto \mathcal{G}_{\text{ref}} \mathcal{O} \, . \qquad (5.24)$$

From above discussion, the non-chiral CFT based on $\mathcal{G}_{\text{ref}} \mathcal{O}$ and $\mathcal{G}_{\text{ref}} \mathcal{O} O$ with $O \in O(m, \mathbb{R}) \times O(n, \mathbb{R})$ is isomorphic. Thus we must quotient out by the right action of $O(m, \mathbb{R}) \times O(n, \mathbb{R})$ on $O(m, n, \mathbb{R})$. Also automorphisms of $\Lambda_{\text{ref}}$, which is isomorphic to a discrete subgroup of $O(m, n, \mathbb{R})$ and denoted by $O(m, n, \mathbb{Z})$, act by right multiplication on $\mathcal{G}_{\text{ref}}$. So we must quotient by the left action of $O(m, n, \mathbb{Z})$ on $O(m, n, \mathbb{R})$. This gives the required structure of the moduli space. □

**Remark 5.3.** In deriving the moduli space of non-chiral CFTs based on Lorentzian lattices, we imposed modular invariance as a requirement. This was crucial in restricting the lattices to self-dual ones. If we lift the modular invariance requirement, we obtain more general non-chiral CFTs recently discussed in [41] and called *generalised Narain theories* [24].

**Remark 5.4.** At a general point in the moduli space (5.23), the sublattice $\Lambda_0$ is trivial and the LLVOA is simply $S(\hat{\mathfrak{h}}^-) \cong S(\hat{\mathfrak{h}}_1^-) \otimes S(\hat{\mathfrak{h}}_2^-)$. The chiral and anti-chiral

---

[23]Recall that in our convention, lattice vectors are written as rows rather than columns.
[24]We thank Masahito Yamazaki for discussion on this point.

algebra is then generated by the vertex operators

$$u_i(x) = \sum_{r \in \mathbb{Z}} u_i(r) x^{-r-1}, \quad v_j(\bar{x}) = \sum_{r \in \mathbb{Z}} v_i(r) \bar{x}^{-r-1}, \quad i = 1, \ldots m, \quad j = 1, \ldots, n \,,$$

(5.25)

corresponding to states $\{u_i(-1) \cdot \mathbf{1}\}_{i=1}^m$ and $\{v_i(-1) \cdot \mathbf{1}\}_{i=1}^n$ where $\{u_i\}, \{v_i\}$ are orthonormal basis of $\hat{\mathfrak{h}}_1, \hat{\mathfrak{h}}_2$ respectively. All other chiral vertex operators are given by products of derivatives of $u_i(x), v_j(\bar{x})$ (see (3.53)). In physics, these are called *Kac-Moody currents* and their modes generate $\mathrm{U}(1)^m \times \mathrm{U}(1)^n$ *Kac-Moody algebra* in the non-chiral CFT. Thus at a generic point in the moduli space, the chiral and anti-chiral algebra is extended from Virasoro to $\mathrm{U}(1)^m \times \mathrm{U}(1)^n$ Kac-Moody algebra. At certain points in the moduli space where $\Lambda_0 \neq 0$, the chiral and anti-chiral algebra is further extended to some enhanced symmetry algebra [25]. It would be interesting to identify the chiral and anti-chiral algebra at these special points in the moduli space with known algebras.

# 6   Narain CFT

Narain CFTs are a large class of conformal field theories which are constructed by compactifying free bosons on a torus and coupling them to a background antisymmetric $B$-field. Narain CFTs naturally appear in string theory when we perform toroidal compactification of strings in multiple directions. In this section, we will describe these CFTs and explain how they provide physical examples of the non-chiral VOA we constructed in Section 3. We restrict to $m = n$ case for this discussion.

## 6.1   Construction of Narain CFTs

We describe the construction of Narain CFTs. The exposition is based on [42, Section 4.1] and [35, Section 8.4].

Let $\Gamma \subset \mathbb{R}^n$ be an $n$-dimensional Euclidean lattice and $2\pi\Gamma$ be the rescaled lattice. Let $\mathbb{T}^n \equiv \mathbb{R}^n/(2\pi\Gamma)$ be the $n$-dimensional torus obtained by imposing the equivalence relation

$$\boldsymbol{x} \sim \boldsymbol{x}' \iff \boldsymbol{x} - \boldsymbol{x}' \in 2\pi\Gamma \,.$$

(6.1)

We then consider $n$ bosons $X^\mu$, $\mu = 1, \ldots, n$ on a two dimensional surface (worldsheet) moving on the torus $\mathbb{T}^n$ (target space). Alternatively, $X^\mu$ can be considered as coordinates on the torus $\mathbb{T}^n$. Note that

$$X^\mu \sim X^\mu + 2\pi e^\mu, \quad \vec{e} \in \Gamma \,.$$

(6.2)

---

[25]We thank Anatoly Dymarsky for discussions on this point.

Let us parameterize the worldsheet by $\sigma, t$. Then the action for the CFT is given by

$$S = \frac{1}{4\pi\alpha'} \int dt \int d\sigma \, (\dot{X}^2 - X'^2 - 2B_{\mu\nu}\dot{X}^\mu X'^\nu) \,, \tag{6.3}$$

where

$$\dot{X}^2 = \sum_{\mu=1}^n \dot{X}^\mu \dot{X}^\mu, \quad X'^2 = \sum_{\mu=1}^n X'^\mu X'^\mu \,, \tag{6.4}$$

with dot indicating derivative with respect to $t$ and prime with respect to $\sigma$, $B_{\mu\nu}$ is an anti-symmetric matrix and $\alpha'$ is a coupling constant (called *Regge slope* in string theory). The equation of motion for $X^\mu$ is given by

$$\ddot{X}^\mu - X''^\mu = 0 \,, \tag{6.5}$$

which is the wave equation in 2 dimensions with solutions

$$X(\sigma, t) = X_L^\mu(t + \sigma) + X_R^\mu(t - \sigma) \,. \tag{6.6}$$

Here $X_L, X_R$ are called the left moving and right moving components. We now take the $\sigma$ coordinate on the worldsheet to be periodic, $\sigma \sim \sigma + 2\pi$ so that

$$X^\mu(t, \sigma + 2\pi) = X^\mu(t, \sigma) + 2\pi e^\mu, \quad \vec{e} \in \Gamma. \tag{6.7}$$

The periodicity implies that we have a Fourier expansion of the form

$$\begin{aligned} X_L^\mu(t + \sigma) &= \frac{x^\mu}{2} + \alpha' \frac{p_L^\mu}{2}(t + \sigma) + \frac{i}{2} \sum_{n \neq 0} \frac{a_n^\mu}{n} e^{-in(t+\sigma)} \,, \\ X_R^\mu(t - \sigma) &= \frac{x^\mu}{2} + \alpha' \frac{p_R^\mu}{2}(t - \sigma) + \frac{i}{2} \sum_{n \neq 0} \frac{b_n^\mu}{n} e^{-in(t-\sigma)} \,, \end{aligned} \tag{6.8}$$

where

$$\frac{\alpha'}{2} (\vec{p}_L - \vec{p}_R) = \vec{e} \in \Gamma \,, \tag{6.9}$$

so that

$$\begin{aligned} X^\mu(t, \sigma) &= X_L^\mu(t + \sigma) + X_R^\mu(t - \sigma) \\ &= x^\mu + \frac{\alpha'}{2}(p_L^\mu + p_R^\mu) t + \vec{e}\sigma + \frac{i}{2} \sum_{n \neq 0} \left( \frac{a_n^\mu}{n} e^{-in(t+\sigma)} + \frac{b_n^\mu}{n} e^{-in(t-\sigma)} \right) \,, \end{aligned} \tag{6.10}$$

satisfies the periodicity (6.7). Note that the total momenta given by

$$P^\mu = \frac{1}{2\pi\alpha'} \int_0^{2\pi} d\sigma \left( \dot{X}^\mu - B^{\mu\nu} X'_\nu \right) = \frac{\alpha'}{2}(p_L^\mu + p_R^\mu) - B^{\mu\nu}e_\nu \,, \tag{6.11}$$

must be a vector of the dual lattice $\Gamma^\star$ defined as

$$\Gamma^\star := \left\{ \vec{e} \in \mathbb{R}^n \mid \vec{e} \cdot \vec{e'} \in \mathbb{Z}, \forall \, \vec{e'} \in \Gamma \right\} \,, \tag{6.12}$$

since $X(t, \sigma)$ is only defined up to arbitrary shifts by $2\pi \vec{e}$ for $\vec{e} \in \Gamma$. We have

$$p_L^\mu = \frac{\alpha' P^\mu + (B^{\mu\nu} + \delta^{\mu\nu}) e_\nu}{\alpha'}, \quad p_R^\mu = \frac{\alpha' P^\mu + (B^{\mu\nu} - \delta^{\mu\nu}) e_\nu}{\alpha'}. \tag{6.13}$$

Thus the set of vectors[26] $\lambda = (\vec{p}_L, \vec{p}_R) \in \mathbb{R}^{n,n}$ form a lattice $\Lambda \subset \mathbb{R}^{n,n}$. Moreover

$$\lambda \circ \lambda := \|p_L\|^2 - \|p_R\|^2 = \frac{4}{\alpha'} \vec{P} \cdot \vec{e}. \tag{6.14}$$

We now fix $\alpha' = 2$, so that $\lambda \circ \lambda \in 2\mathbb{Z}$. Next for any $\lambda, \lambda' \in \Lambda$

$$\begin{aligned} \lambda \circ \lambda' &= \vec{p}_L \cdot \vec{p'}_L - \vec{p}_R \cdot \vec{p'}_R \\ &= \vec{P}' \cdot \vec{e} + \vec{P} \cdot \vec{e}' \in \mathbb{Z}. \end{aligned} \tag{6.15}$$

So $\Lambda$ is an even[27] Lorentzian lattice. In fact, $\Lambda$ is self dual [42]. A generator matrix for $\Lambda$ in the coordinates [43]

$$\lambda = (\alpha, \beta), \quad \alpha = \frac{\vec{p}_L + \vec{p}_R}{\sqrt{2}}, \quad \beta = \frac{\vec{p}_L - \vec{p}_R}{\sqrt{2}}, \tag{6.16}$$

is

$$\mathcal{G}_\Lambda = \frac{1}{\sqrt{2}} \left( \begin{array}{c|c} 2\gamma^\star & 0 \\ \hline \gamma B & \gamma \end{array} \right), \tag{6.17}$$

where $\gamma$ and $\gamma^\star = (\gamma^{-1})^T$ are the generator matrices for $\Gamma$ and $\Gamma^\star$ respectively. Upon quantisation, we impose the commutators

$$\begin{aligned} &[a_n^\mu, a_m^\nu] = n\delta_{n+m,0}\delta^{\mu\nu}, \quad [b_n^\mu, b_m^\nu] = n\delta_{n+m,0}\delta^{\mu\nu}, \quad [a_n^\mu, b_m^\nu] = 0, \\ &[x^\mu, p_L^\mu] = [x^\mu, p_R^\nu] = i\delta^{\mu\nu}. \end{aligned} \tag{6.18}$$

For every $(\vec{p}_L, \vec{p}_R) \in \Lambda$ we have a primary operator given by

$$V_{p_L, p_R}(z, \bar{z}) = {}_{\,\bullet}^{\,\bullet} e^{i\vec{p}_L \cdot \vec{X}_L(z) + i\vec{p}_R \cdot \vec{X}_R(\bar{z})} {}_{\,\bullet}^{\,\bullet}, \tag{6.19}$$

where $z = e^{i(it+\sigma)}, \bar{z} = e^{i(it-\sigma)}$ which is obtained by Wick rotating $t \to it$. The normal ordering is defined via

$$\begin{aligned} {}_{\,\bullet}^{\,\bullet} a_n^\mu a_m^\nu {}_{\,\bullet}^{\,\bullet} = {}_{\,\bullet}^{\,\bullet} a_m^\nu a_n^\mu {}_{\,\bullet}^{\,\bullet} &= \begin{cases} a_m^\nu a_n^\mu & m \le n, \\ a_n^\mu a_m^\nu & m \ge n, \end{cases} \\ {}_{\,\bullet}^{\,\bullet} a_m^\mu p_L^\nu {}_{\,\bullet}^{\,\bullet} = {}_{\,\bullet}^{\,\bullet} p_L^\nu a_m^\mu {}_{\,\bullet}^{\,\bullet} &= a_m^\mu p_L^\nu, \\ {}_{\,\bullet}^{\,\bullet} x^\mu a_n^\nu {}_{\,\bullet}^{\,\bullet} = {}_{\,\bullet}^{\,\bullet} a_n^\nu x^\mu {}_{\,\bullet}^{\,\bullet} &= a_n^\nu x^\mu, \\ {}_{\,\bullet}^{\,\bullet} x^\mu p_L^\nu {}_{\,\bullet}^{\,\bullet} = {}_{\,\bullet}^{\,\bullet} p_L^\nu x^\mu {}_{\,\bullet}^{\,\bullet} &= x^\mu p_L^\nu, \end{aligned} \tag{6.20}$$

---

[26] We again take the lattice vectors to be rows in $\mathbb{R}^{n,n}$.

[27] Note that an even lattice is necessarily integral.

and similarly for $b_n^\mu, p_R^\mu$. The Virasoro generators are given by

$$L_n = \frac{1}{2} \sum_{m \in \mathbb{Z}} {}^{\bullet}_{\bullet} a_m \cdot a_{n-m} {}^{\bullet}_{\bullet}, \quad \bar{L}_n = \frac{1}{2} \sum_{m \in \mathbb{Z}} {}^{\bullet}_{\bullet} b_m \cdot b_{n-m} {}^{\bullet}_{\bullet} . \tag{6.21}$$

It is an easy exercise to show that $L_n, \bar{L}_n$ satisfy the Virasoro algebra with central charge $(c, \bar{c}) = (n, n)$. The OPE of these primary operators takes the form [35, Section 8.4]

$$V_{p_L, p_R}(z, \bar{z}) V_{p'_L, p'_R}(w, \bar{w}) \sim (z - w)^{p_L \cdot p'_L} (\bar{z} - \bar{w})^{p_R \cdot p'_R} V_{p_L + p'_L, p_R + p'_R}(w, \bar{w}) . \tag{6.22}$$

From the OPE we see that as the first vertex operator circles around the second, it picks up a factor of $e^{2\pi i (p_L \cdot p'_L - p_R \cdot p'_R)}$. So for the OPE to be single valued, one requires the lattice $\Lambda$ to be integral.

The torus partition function of the theory is

$$Z(\tau, \bar{\tau}) = \frac{1}{|\eta(\tau)|^{2n}} \sum_{(\vec{p}_L, \vec{p}_R) \in \Lambda} q^{\|\vec{p}_L\|^2/2} \bar{q}^{\|\vec{p}_R\|^2/2}, \quad q = e^{2\pi i \tau}, \bar{q} = e^{2\pi i \bar{\tau}}, \tag{6.23}$$

where $\eta(\tau)$ is the Dedekind eta function

$$\eta(\tau) = q^{\frac{1}{24}} \prod_{n=1}^{\infty} (1 - q^n), \tag{6.24}$$

and $\tau$ is the moduli of the torus. Here $\tau, \bar{\tau}$ are complex conjugates of each other. The partition function (6.23) is modular invariant since $\Lambda$ is self-dual. This construction gives a CFT for any even, self-dual Lorentzian lattice $\Lambda \subset \mathbb{R}^{n,n}$ of signature $(n, n)$, this is called the Narain CFT associated to the lattice $\Lambda$.

It is easy to see that the partition function is invariant under an orthogonal action on $\vec{p}_L, \vec{p}_R$ separately. Thus two Narain CFT associated on lattices $\Lambda, \Lambda'$ are equivalent if $\Lambda, \Lambda'$ is related by the right action of $O(n, \mathbb{R}) \times O(n, \mathbb{R})$, where $O(n, \mathbb{R}) \subset GL(n, \mathbb{R})$ acts on $\mathbb{R}^n$ and preserves the Euclidean inner product. Next, any two Lorentzian lattices are related by the left action of $O(n, n, \mathbb{R})$, where $O(n, n, \mathbb{R}) \subset GL(2n, \mathbb{R})$ acts on $\mathbb{R}^{n,n}$ and preserves the Lorentzian inner product of signature $(n, n)$. But note that if two lattices are related by an $O(n, n, \mathbb{Z})$-transformation then the two lattices are the same. Thus the moduli space of Narain CFTs is given by the quotient

$$\frac{O(n, n, \mathbb{R})}{O(n, \mathbb{R}) \times O(n, \mathbb{R}) \times O(n, n, \mathbb{Z})}, \tag{6.25}$$

where the first two factors in the denominator act on the right and relate physically equivalent lattices to each other while the last factor acts on the left.

It is immediate to see from the dictionary between conformal field theory in physics and our notion of non-chiral CFT described in Subection 1.1 that Narain CFTs are simply non-chiral CFTs based on a Lorentzian lattice of signature $(n, n)$. The moduli space structure (6.25) is then a special case of Theorem 5.5. The general case $m \neq n$ also appears in toroidal compactification of heterotic string theory, see [38, 44–46] for details. Our general result in Theorem 5.5 is a more mathematical statement of the moduli space of Narain compactifications for general signature, see [47] for more details from physical viewpoint.

Finally we note that [42] gives a construction of Lorentzian lattices of signature $(n, n)$ from stabiliser codes. Our construction of LLVOA then completes the parallel with the construction of VOAs from codes. The code-Narain CFT correspondence has been explored extensively recently [48–52]. It would be interesting to study their implications on LLVOAs.

# 7 Conclusion and Future Directions

In this paper, we have initiated a mathematically rigorous study of non-chiral VOAs and presented the construction of a non-trivial example of our definition, namely the Lorentzian lattice vertex operator algebra and its modules. We also showed the relevance of the construction by demonstrating that Narain CFTs which appear in string compactifications are physical examples of our construction. In this section, we sketch some future directions of the study.

(1) Rationality, Regularity, and $C_2$-Cofiniteness: We have defined the notion of non-chiral VOA. It is natural to introduce the notion of rationality , regularity, and strong regularity as in the theory of vertex operator algebras. One would also like to define the notion of $C_2$-cofiniteness and and see if $C_2$-cofiniteness and rationality implies regularity as in [36]. The main result that we would like to prove is the theorem of Zhu [53]: the graded dimensions of the irreducible modules of a $C_2$-cofinite VOA with certain additional properties is a representation of $\mathrm{SL}(2, \mathbb{Z})$. To be more precise, the set of graded charaters is a vector-valued modular form. We would like to prove a similar theorem for non-chiral VOA in the generalised sense of a bi-modular form:

$$\chi_{W_i} \left( \frac{a\tau + b}{c\tau + d}, \frac{a\bar{\tau} + b}{c\bar{\tau} + d} \right) = \sum_i \rho(\gamma)_{ij} \chi_{W_j}(\tau, \bar{\tau}), \quad \gamma = \begin{pmatrix} a & b \\ c & d \end{pmatrix} \in \mathrm{SL}(2, \mathbb{Z}) \qquad (7.1)$$

where $\rho(\gamma)_{ij}$ is the representation matrix of $\gamma$. Furthermore, we would like to classify all irreducible modules of the LLVOA on the lines of Dong [54] and obtain conditions on the Lorentzian lattice so that the associated LLVOA is, rational, regular

and strongly regular. The first result in this direction is due to Katrin Wendland [55], see also [43] for some recent progress from physical viewpoint.

**(2) Modular Tensor Categories and the Verlinde Conjecture:** In conformal field theory in physics, one expects the Verlinde conjecture [56] to hold even for non-chiral CFTs. In [12], Moore and Seiberg showed that Verlinde conjecture follows from the axioms of a *rational conformal field theory* which they defined. Their axioms had an important *associativity* assumption. Huang [57] established the associativity (axiom in [12]) from the axioms of vertex operator algebra and its modules. It required the introduction of the notion of tensor product of modules [58]. Huang also proved the Verlinde conjecture using the definition of tensor product of modules of a VOA and the associativity theorem.

We would like to define a similar notion of tensor product of modules for non-chiral VOA and then prove the associativity theorem and Verlinde conjecture. Additionally, one of the main results of [12] was the realisation that conformal field theories can be understood as generalisation of group theory - the chiral (anti-chiral) algebra and its modules along with intertwining operators forms a category called a modular tensor category [59]. Huang proved that the vertex operator algebras and their modules are examples of *braided tensor categories* [60]. We would like to follow a similar approach and establish these results for non-chiral VOAs.

**Acknowledgments.** The authors would like to thank Anatoly Dymarsky, Yi-Zhi Huang, Gregory Moore, Ananda Roy, Siddhartha Sahi, Hubert Saleur, Ashoke Sen, and Masahito Yamazaki for some useful correspondence and discussions. We especially thank Anatoly Dymarsky, Yi-Zhi Huang, and Hubert Saleur for comments on the manuscript and raising some interesting questions. We also thank Runkai Tao for proof-reading the manuscript. The work of R.K.S is supported by the US Department of Energy under grant DE-SC0010008.

## A  Lattice central extensions

In this appendix, we collect some results about central extensions and refer the reader to [2, Chapter 5] for more details. Recall that a central extension of $G$ by $A$ is a short exact sequence

$$0 \longrightarrow A \overset{\iota}{\longrightarrow} \hat{G} \overset{-}{\longrightarrow} G \longrightarrow 0 \,, \tag{A.1}$$

such that $\iota(A)$ is contained in the center $Z(\hat{G})$ of $\hat{G}$. Here $-$ denotes the surjective (projection) map. We sometimes call $\hat{G}$ as the central extension of $G$ by $A$. Two central extensions $\hat{G}$ and $\hat{G}'$ are said to be equivalent if there exists an isomorphism $\psi : \hat{G} \longrightarrow \hat{G}'$ such that the following diagrams commute

$$
\begin{array}{ccccccccc}
0 & \longrightarrow & A & \overset{\iota}{\longrightarrow} & \hat{G} & \overset{-}{\longrightarrow} & G & \longrightarrow & 0 \\
& & \downarrow & & \downarrow{\psi} & & \downarrow & & \\
0 & \longrightarrow & A & \overset{\iota}{\longrightarrow} & \hat{G}' & \overset{-}{\longrightarrow} & G & \longrightarrow & 0
\end{array}
$$

We now specialize to an abelian group $G$, written additively, and $A = \mathbb{Z}_2 = \{\pm 1\}$. A *2-cocycle* is a bilinear map $\epsilon : G \times G \longrightarrow \mathbb{Z}_2$ satisfying

$$\epsilon(a,b) + \epsilon(a+b,c) = \epsilon(b,c) + \epsilon(a,b+c). \tag{A.2}$$

A 2-cocycle $\epsilon$ is called a *2-coboundary* if there exists $\eta : G \longrightarrow \mathbb{Z}_2$ such that

$$\epsilon(a,b) = \eta(a+b) - \eta(a) - \eta(b), \quad a,b \in G. \tag{A.3}$$

Two 2-cocycles differing by a 2-coboundary are said to be *cohomologous*. A 2-cocycle determines a central extension $\hat{G}$ of $G$ by $\mathbb{Z}_2$ as follows:

$$\hat{G} = \mathbb{Z}_2 \times G = \{(\theta, a) : \theta \in \mathbb{Z}_2, a \in G\} \tag{A.4}$$

with group operation

$$(\theta, a) \cdot (\tau, b) = \left(\theta\tau\,(-1)^{\epsilon(a,b)}, a+b\right). \tag{A.5}$$

A 2-coboundary determines a central extension equivalent to the trivial extension $\mathbb{Z}_2 \times G$ (*i.e.* the direct product of $\mathbb{Z}_2$ and $G$) and two cohomologous 2-cocycles determine equivalent central extensions. It turns out that equivalence classes of central extensions is classified by the group cohomology $H^2(G, \mathbb{Z}_2)$ which is the quotient of 2-cocycles by 2-coboundaries ([2, Proposition 5.1.2]).

We define the commutator map $c : G \times G \longrightarrow \mathbb{Z}_2$ of a central extension $\hat{G}$ by the relation

$$(-1)^{c(\bar{a},\bar{b})} = aba^{-1}b^{-1}, \quad a,b \in \hat{G}. \tag{A.6}$$

The following proposition will be useful.

**Proposition A.1.** [2, Proposition 5.2.3] *Two central extensions of $G$ by $\mathbb{Z}_2$ are equivalent if and only if their commutator maps are the same.*

Let us now consider the Lorentzian lattice $\Lambda$ as an abelian group. Let $\hat{\Lambda}$ be the central extension of $\Lambda$ by $\mathbb{Z}_2$ determined by the cocycle $\epsilon$ defined in (3.37).

**Lemma A.1.** *The commutator function $c : \Lambda \times \Lambda \to \mathbb{Z}_2$ for the central extension $\hat{\Lambda}$ in (3.39) is given by $c(\mu_1, \mu_2) = \mu_1 \circ \mu_2 \bmod 2\mathbb{Z}$.*

*Proof.* We use the notation $e_\lambda = (1, \lambda)$, like in Section 3. Using the group operation (A.5), it can be shown that

$$
\begin{aligned}
e_{\mu_1} e_{\mu_2} &= (-1)^{\epsilon(\mu_1, \mu_2)} e_{\mu_1 + \mu_2}, \\
e_{\mu_2} e_{\mu_1} &= (-1)^{\epsilon(\mu_2, \mu_1)} e_{\mu_1 + \mu_2}.
\end{aligned}
\tag{A.7}
$$

Using the above equations, the commutator function is seen to be

$$
(-1)^{c(\mu_1, \mu_2)} = e_{\mu_1} e_{\mu_2} e_{\mu_1}^{-1} e_{\mu_2}^{-1} = (-1)^{\epsilon(\mu_1, \mu_2) - \epsilon(\mu_2, \mu_1)}.
\tag{A.8}
$$

If we take the basis of the lattice to be given by $\{\lambda_i\}_{i=1}^{m+n}$, then we have say

$$
\mu_1 = \sum_{i=1}^{m+n} c_i \lambda_i, \quad \mu_2 = \sum_{i=1}^{m+n} d_i \lambda_i, \quad c_i, d_i \in \mathbb{Z}.
\tag{A.9}
$$

Using this, we can write

$$
(-1)^{\epsilon(\mu_1, \mu_2) - \epsilon(\mu_2, \mu_1)} = (-1)^{\sum_{i,j=1}^{m+n} c_i d_j \epsilon(\lambda_i, \lambda_j) - c_i d_j \epsilon(\lambda_j, \lambda_i)} = \prod_{i,j=1}^{m+n} (-1)^{c_i d_j (\epsilon(\lambda_i, \lambda_j) - \epsilon(\lambda_j, \lambda_i))}.
\tag{A.10}
$$

Using (3.37), each term in the above product can be written as

$$
(-1)^{c_i d_j (\epsilon(\lambda_i, \lambda_j) - \epsilon(\lambda_j, \lambda_i))} = \begin{cases} (-1)^{c_i d_j \, \lambda_i \circ \lambda_j}, & i > j, \\ (-1)^{-c_i d_j \, \lambda_i \circ \lambda_j} = (-1)^{c_i d_j \, \lambda_i \circ \lambda_j}, & i < j, \\ 1 = (-1)^{c_i d_j \, \lambda_i \circ \lambda_i}, & i = j, \end{cases}
\tag{A.11}
$$

where the equality in $i < j$ case follows as $(-1)^n = (-1)^{-n}$, when $n \in \mathbb{Z}$, and the equality in case $i = j$ follows as the lattice is even. Hence, we can simplify (A.10) as

$$
(-1)^{\epsilon(\mu_1, \mu_2) - \epsilon(\mu_2, \mu_1)} = \prod_{i,j=1}^{m+n} (-1)^{c_i d_j \, \lambda_i \circ \lambda_j} = (-1)^{\mu_1 \circ \mu_2} = (-1)^{c(\mu_1, \mu_2)}.
\tag{A.12}
$$

Hence, we conclude $c(\mu_1, \mu_2) = \mu_1 \circ \mu_2 \bmod 2\mathbb{Z}$. $\qquad\square$

**Proposition A.2.** *Let $\{\tilde{\lambda}_i\}_{i=1}^{m+n}$ be a basis of $\Lambda$ different from $\{\lambda_i\}_{i=1}^{m+n}$. Then the cocycle $\tilde{\epsilon} : \Lambda \times \Lambda \longrightarrow \mathbb{Z}_2$ defined as in* (3.37)

$$\tilde{\epsilon}(\tilde{\lambda}_i, \tilde{\lambda}_j) = \begin{cases} \tilde{\lambda}_i \circ \tilde{\lambda}_j, & i > j, \\ 0, & otherwise, \end{cases} \tag{A.13}$$

*and extended to $\Lambda$ by $\mathbb{Z}$-bilinearity, determines a central extension equivalent to the one determined by $\epsilon$.*

*Proof.* From Lemma A.1, the commutator maps of $\epsilon$ and $\tilde{\epsilon}$ are the same. Thus by Proposition A.1, the two central extensions are equivalent. $\square$

# B   Locality of product of multiple vertex operators

In this appendix, we show that that the product of multiple vertex operators of LLVOA and its modules exists and is local in the sense of the locality property 9.

We first prove the existence and locality for product of three vertex operators. For three vectors $e^\lambda, e^{\lambda'}, e^{\lambda''} \in \mathbb{C}[\Lambda]$, we want to calculate the product

$$Y_{V_\Lambda}(e^\lambda, x_1, \bar{x}_1) Y_{V_\Lambda}(e^{\lambda'}, x_2, \bar{x}_2) Y_{V_\Lambda}(e^{\lambda''}, x_3, \bar{x}_3). \tag{B.1}$$

From (3.115) and (3.52), we have

$$Y_{V_\Lambda}(e^\lambda, x_1, \bar{x}_1) Y_{V_\Lambda}(e^{\lambda'}, x_2, \bar{x}_2) Y_{V_\Lambda}(e^{\lambda''}, x_3, \bar{x}_3)$$

$$= (x_2 - x_3)^{\langle \alpha', \alpha'' \rangle} (\bar{x}_2 - \bar{x}_3)^{\langle \beta', \beta'' \rangle} Y_{V_\Lambda}(e^\lambda, x_1, \bar{x}_1) \exp\left(\int \alpha'(x_2)^-\right) \exp\left(\int \alpha''(x_3)^-\right)$$

$$\exp\left(\int \alpha'(x_2)^+\right) \quad \exp\left(\int \alpha''(x_3)^+\right) \exp\left(\int \beta'(\bar{x}_2)^-\right) \exp\left(\int \beta''(\bar{x}_3)^-\right)$$

$$\exp\left(\int \beta'(\bar{x}_2)^+\right) \exp\left(\int \beta''(\bar{x}_3)^+\right) e_{\lambda'} e_{\lambda''} x_2^{\alpha'} \bar{x}_2^{\beta'} x_3^{\alpha''} \bar{x}_3^{\beta''}$$

$$= (x_2 - x_3)^{\langle \alpha', \alpha'' \rangle} (\bar{x}_2 - \bar{x}_3)^{\langle \beta', \beta'' \rangle} \exp\left(\int \alpha(x_1)^-\right) \exp\left(\int \alpha(x_1)^+\right)$$

$$\exp\left(\int \beta(\bar{x}_1)^-\right) \exp\left(\int \beta(\bar{x}_1)^+\right) e_\lambda x_1^\alpha \bar{x}_1^\beta \exp\left(\int \alpha'(x_2)^-\right) \exp\left(\int \alpha''(x_3)^-\right)$$

$$\exp\left(\int \alpha'(x_2)^+\right) \quad \exp\left(\int \alpha''(x_3)^+\right) \exp\left(\int \beta'(\bar{x}_2)^-\right) \exp\left(\int \beta''(\bar{x}_3)^-\right)$$

$$\exp\left(\int \beta'(\bar{x}_2)^+\right) \exp\left(\int \beta''(\bar{x}_3)^+\right) e_{\lambda'} e_{\lambda''} x_2^{\alpha'} \bar{x}_2^{\beta'} x_3^{\alpha''} \bar{x}_3^{\beta''}. \tag{B.2}$$

Now using (3.109) and (3.110) we get

$$Y_{V_\Lambda}(e^\lambda, x_1, \bar{x}_1) Y_{V_\Lambda}(e^{\lambda'}, x_2, \bar{x}_2) Y_{V_\Lambda}(e^{\lambda''}, x_3, \bar{x}_3)$$

$$= (x_1 - x_2)^{\langle \alpha, \alpha' \rangle} (\bar{x}_1 - \bar{x}_2)^{\langle \beta, \beta' \rangle} (x_1 - x_3)^{\langle \alpha, \alpha'' \rangle} (\bar{x}_1 - \bar{x}_3)^{\langle \beta, \beta'' \rangle} (x_2 - x_3)^{\langle \alpha', \alpha'' \rangle} (\bar{x}_2 - \bar{x}_3)^{\langle \beta', \beta'' \rangle}$$

$$\exp\left(\int \alpha(x_1)^-\right) \exp\left(\int \alpha'(x_2)^-\right) \exp\left(\int \alpha''(x_3)^-\right) \exp\left(\int \alpha(x_1)^+\right)$$

$$\exp\left(\int \alpha'(x_2)^+\right) \quad \exp\left(\int \alpha''(x_3)^+\right) \exp\left(\int \beta(\bar{x}_1)^-\right) \exp\left(\int \beta'(\bar{x}_2)^-\right)$$

$$\exp\left(\int \beta''(\bar{x}_3)^-\right) \exp\left(\int \beta(\bar{x}_1)^+\right) \exp\left(\int \beta'(\bar{x}_2)^+\right) \exp\left(\int \beta''(\bar{x}_3)^+\right)$$

$$e_\lambda e_{\lambda'} e_{\lambda''} x_1^\alpha \bar{x}_1^\beta x_2^{\alpha'} \bar{x}_2^{\beta'} x_3^{\alpha''} \bar{x}_3^{\beta''}$$

$$\equiv (x_1 - x_2)^{\langle \alpha, \alpha' \rangle} (\bar{x}_1 - \bar{x}_2)^{\langle \beta, \beta' \rangle} (x_1 - x_3)^{\langle \alpha, \alpha'' \rangle} (\bar{x}_1 - \bar{x}_3)^{\langle \beta, \beta'' \rangle} (x_2 - x_3)^{\langle \alpha', \alpha'' \rangle} (\bar{x}_2 - \bar{x}_3)^{\langle \beta', \beta'' \rangle}$$

$$F(x_1, x_2, x_3) G(\bar{x}_1, \bar{x}_2, \bar{x}_3). \tag{B.3}$$

Substituting complex variables $x_1 = z_1, x_2 = z_2$ and $x_3 = z_3$, it is easy to show that the right hand side of (B.3) is the expansion of the function

$$
\exp\left(\langle \alpha, \alpha' \rangle \log(z_1 - z_2)\right) \exp\left(\langle \beta, \beta' \rangle \log(\bar{z}_1 - \bar{z}_2)\right) \exp\left(\langle \alpha, \alpha'' \rangle \log(z_1 - z_3)\right)
$$
$$
\exp\left(\langle \beta, \beta'' \rangle \log(\bar{z}_1 - \bar{z}_3)\right) \exp\left(\langle \alpha', \alpha'' \rangle \log(z_2 - z_3)\right) \exp\left(\langle \beta', \beta'' \rangle \log(\bar{z}_2 - \bar{z}_3)\right) \quad \text{(B.4)}
$$
$$
F(z_1, z_2, z_3) G(\bar{z}_1, \bar{z}_2, \bar{z}_3)
$$

in the region $|z_1| > |z_2| > |z_3|$. Following the same arguments as in Section 3.3.1, we see that the three vertex operators satisfy locality for transpositions $(12), (23) \in S_3$. For the transposition $(13) \in S_3$, we get a sign $(-1)^{\lambda \circ \lambda' + \lambda' \circ \lambda'' + \lambda \circ \lambda''}$ from the exchange of operators $e_{\lambda''} e_{\lambda'} e_{\lambda} \to e_{\lambda} e_{\lambda'} e_{\lambda''}$. This sign can be used to show that the functions appearing in (B.3) after the permutation $z_1 \leftrightarrow z_3$ is the expansion of the function (B.4) in the region $|z_3| > |z_2| > |z_1|$. Since $S_3$ is generated by the three permutations $(12), (23), (13)$, the locality property holds for any permutation in $S_3$.

The proof for an arbitrary number of vertex operator is entirely similar.

We now prove locality for general vectors of the form (3.35). We first want to calculate

$$
Y_{V_\Lambda}(u, x_1, \bar{x}_1) Y_{V_\Lambda}(v, x_2, \bar{x}_2) Y_{V_\Lambda}(w, x_3, \bar{x}_3), \quad \text{(B.5)}
$$

with vectors $u, v, w$ of the form (3.35):

$$
u = \left(\alpha_1(-l_1) \cdot \alpha_2(-l_2) \cdots \alpha_k(-l_k) \cdot \beta_1(-\bar{l}_1) \cdot \beta_2(-\bar{l}_2) \cdots \beta_{\bar{k}}(-\bar{l}_{\bar{k}})\right) \otimes e^{(\alpha, \beta)},
$$
$$
v = \left(\alpha'_1(-m_1) \cdot \alpha'_2(-m_2) \cdots \alpha'_\ell(-m_\ell) \cdot \beta'_1(-\bar{m}_1) \cdot \beta'_2(-\bar{m}_2) \cdots \beta'_{\bar{\ell}}(-\bar{m}_{\bar{\ell}})\right) \otimes e^{(\alpha', \beta')},
$$
$$
w = \left(\alpha''_1(-n_1) \cdot \alpha''_2(-n_2) \cdots \alpha''_m(-n_m) \cdot \beta''_1(-\bar{n}_1) \cdot \beta''_2(-\bar{n}_2) \cdots \beta''_{\bar{m}}(-\bar{n}_{\bar{m}})\right) \otimes e^{(\alpha'', \beta'')},
$$
$$
\quad \text{(B.6)}
$$

and we further use the notation that $\lambda = (\alpha, \beta)$, $\lambda' = (\alpha', \beta')$, and $\lambda'' = (\alpha'', \beta'')$. Using (3.138), we have

$$
Y_{V_\Lambda}(u, x_1, \bar{x}_1) Y_{V_\Lambda}(v, x_2, \bar{x}_2) Y_{V_\Lambda}(w, x_3, \bar{x}_3) = (x_1 - x_2)^{\langle \alpha, \alpha' \rangle} (\bar{x}_1 - \bar{x}_2)^{\langle \beta, \beta' \rangle} Y_{V_\Lambda}(u, x_1, \bar{x}_1)
$$
$$
\times \exp\left(\int \alpha'(x_2)^-\right) \exp\left(\int \alpha''(x_3)^-\right) \exp\left(\int \beta'(\bar{x}_2)^-\right) \exp\left(\int \beta''(\bar{x}_3)^-\right)
$$
$$
\times {}_\circ^\circ \prod_{r=1}^{\ell} \left[ \frac{1}{(m_r - 1)!} \frac{d^{m_r - 1} \alpha'_r(x_2)}{dx_2^{m_r - 1}} + (-1)^{m_r - 1} \left( \frac{\langle \alpha'', \alpha'_r \rangle}{(x_2 - x_3)^{m_r}} - \frac{\langle \alpha'', \alpha'_r \rangle}{x_2^{m_r}} \right) \right]
$$
$$
\times \prod_{s=1}^{\bar{\ell}} \left[ \frac{1}{(\bar{m}_s - 1)!} \frac{d^{\bar{m}_s - 1} \beta'_s(\bar{x}_2)}{d\bar{x}_2^{\bar{m}_s - 1}} + (-1)^{\bar{m}_s - 1} \left( \frac{\langle \beta'', \beta'_s \rangle}{(\bar{x}_2 - \bar{x}_3)^{\bar{m}_s}} - \frac{\langle \beta'', \beta'_s \rangle}{\bar{x}_2^{\bar{m}_s}} \right) \right] {}_\circ^\circ
$$

$$\times \; {\overset{\bullet}{\underset{\bullet}{}}} \prod_{p=1}^{m} \left[ \frac{1}{(n_p-1)!} \frac{d^{n_p-1}\alpha_p''(x_3)}{dx_3^{n_p-1}} - \frac{\langle \alpha', \alpha_p'' \rangle}{(x_2-x_3)^{n_p}} - (-1)^{n_p-1} \frac{\langle \alpha', \alpha_p'' \rangle}{x_3^{n_p}} \right]$$

$$\times \prod_{q=1}^{\bar{m}} \left[ \frac{1}{(\bar{n}_q-1)!} \frac{d^{\bar{n}_q-1}\beta_q''(\bar{x}_3)}{d\bar{x}_3^{\bar{n}_q-1}} - \frac{\langle \beta', \beta_q'' \rangle}{(\bar{x}_2-\bar{x}_3)^{\bar{n}_q}} - (-1)^{\bar{n}_q-1} \frac{\langle \beta', \beta_q'' \rangle}{\bar{x}_3^{\bar{n}_q}} \right] {\overset{\bullet}{\underset{\bullet}{}}} \qquad \text{(B.7)}$$

$$\times \exp\left( \int \alpha'(x_2)^+ \right) \exp\left( \int \beta'(\bar{x}_2)^+ \right) \exp\left( \int \alpha''(x_3)^+ \right) \exp\left( \int \beta''(\bar{x}_3)^+ \right)$$

$$\times \; \mathrm{e}_{\lambda'} \mathrm{e}_{\lambda''} x_2^{\alpha'} \bar{x}_2^{\beta'} x_3^{\alpha''} \bar{x}_3^{\beta''} \, .$$

Now using the general expression for vertex operator (2.17) and the relations (3.130), (3.131), (3.132), and (3.134) successively on the product in normal order, along with relations (3.109),(3.110) and the formal variable identity (2.6), we obtain

$$Y_{V_\Lambda}(u, x_1, \bar{x}_1) Y_{V_\Lambda}(v, x_2, \bar{x}_2) Y_{V_\Lambda}(w, x_3, \bar{x}_3)$$
$$= (x_1-x_2)^{\langle \alpha, \alpha' \rangle} (\bar{x}_1-\bar{x}_2)^{\langle \beta, \beta' \rangle} (x_1-x_3)^{\langle \alpha, \alpha'' \rangle} (\bar{x}_1-\bar{x}_3)^{\langle \beta, \beta'' \rangle} (x_2-x_3)^{\langle \alpha', \alpha'' \rangle} (\bar{x}_2-\bar{x}_3)^{\langle \beta', \beta'' \rangle}$$

$$\times \exp\left( \int \alpha(x_1)^- \right) \exp\left( \int \alpha'(x_2)^- \right) \exp\left( \int \alpha''(x_3)^- \right)$$

$$\times \exp\left( \int \beta(\bar{x}_1)^- \right) \exp\left( \int \beta'(\bar{x}_2)^- \right) \exp\left( \int \beta''(\bar{x}_3)^- \right)$$

$$\times \; {\overset{\bullet}{\underset{\bullet}{}}} \prod_{i=1}^{k} \left[ \frac{1}{(l_i-1)!} \frac{d^{l_i-1}\alpha_i(x_1)}{dx_1^{l_i-1}} + (-1)^{l_i-1} \left( \frac{\langle \alpha', \alpha_i \rangle}{(x_1-x_2)^{l_i}} - \frac{\langle \alpha', \alpha_i \rangle}{x_1^{l_i}} + \frac{\langle \alpha'', \alpha_i \rangle}{(x_1-x_3)^{l_i}} - \frac{\langle \alpha'', \alpha_i \rangle}{x_1^{l_i}} \right) \right]$$

$$\times \prod_{j=1}^{\bar{k}} \left[ \frac{1}{(\bar{l}_j-1)!} \frac{d^{\bar{l}_j-1}\beta_j(\bar{x}_1)}{d\bar{x}_1^{\bar{l}_j-1}} + (-1)^{\bar{l}_j-1} \left( \frac{\langle \beta', \beta_j \rangle}{(\bar{x}_1-\bar{x}_2)^{\bar{l}_j}} - \frac{\langle \beta', \beta_j \rangle}{\bar{x}_1^{\bar{l}_j}} + \frac{\langle \beta'', \beta_j \rangle}{(\bar{x}_1-\bar{x}_3)^{\bar{l}_j}} - \frac{\langle \beta'', \beta_j \rangle}{\bar{x}_1^{\bar{l}_j}} \right) \right] {\overset{\bullet}{\underset{\bullet}{}}}$$

$$\times \; {\overset{\bullet}{\underset{\bullet}{}}} \prod_{r=1}^{\ell} \left[ \frac{1}{(m_r-1)!} \frac{d^{m_r-1}\alpha_r'(x_2)}{dx_2^{m_r-1}} + (-1)^{m_r-1} \left( \frac{\langle \alpha'', \alpha_r' \rangle}{(x_2-x_3)^{m_r}} - \frac{\langle \alpha'', \alpha_r' \rangle}{x_2^{m_r}} - \frac{\langle \alpha, \alpha_r' \rangle}{x_2^{m_r}} \right) - \frac{\langle \alpha, \alpha_r' \rangle}{(x_1-x_2)^{m_r}} \right]$$

$$\times \prod_{s=1}^{\bar{\ell}} \left[ \frac{1}{(\bar{m}_s-1)!} \frac{d^{\bar{m}_s-1}\beta_s'(\bar{x}_2)}{d\bar{x}_2^{\bar{m}_s-1}} + (-1)^{\bar{m}_s-1} \left( \frac{\langle \beta'', \beta_s' \rangle}{(\bar{x}_2-\bar{x}_3)^{\bar{m}_s}} - \frac{\langle \beta'', \beta_s' \rangle}{\bar{x}_2^{\bar{m}_s}} - \frac{\langle \beta, \beta_s' \rangle}{\bar{x}_2^{\bar{m}_s}} \right) - \frac{\langle \beta, \beta_s' \rangle}{(\bar{x}_1-\bar{x}_2)^{\bar{m}_s}} \right] {\overset{\bullet}{\underset{\bullet}{}}}$$

$$\times \; {\overset{\bullet}{\underset{\bullet}{}}} \prod_{p=1}^{m} \left[ \frac{1}{(n_p-1)!} \frac{d^{n_p-1}\alpha_p''(x_3)}{dx_3^{n_p-1}} - \frac{\langle \alpha', \alpha_p'' \rangle}{(x_2-x_3)^{n_p}} - \frac{\langle \alpha, \alpha_p'' \rangle}{(x_1-x_3)^{n_p}} - (-1)^{n_p-1} \left( \frac{\langle \alpha', \alpha_p'' \rangle}{x_3^{n_p}} + \frac{\langle \alpha, \alpha_p'' \rangle}{x_3^{n_p}} \right) \right]$$

$$\times \prod_{q=1}^{\bar{m}} \left[ \frac{1}{(\bar{n}_q-1)!} \frac{d^{\bar{n}_q-1}\beta_q''(\bar{x}_3)}{d\bar{x}_3^{\bar{n}_q-1}} - \frac{\langle \beta', \beta_q'' \rangle}{(\bar{x}_2-\bar{x}_3)^{\bar{n}_q}} - \frac{\langle \beta, \beta_q'' \rangle}{(\bar{x}_1-\bar{x}_3)^{\bar{n}_q}} - (-1)^{\bar{n}_q-1} \left( \frac{\langle \beta', \beta_q'' \rangle}{\bar{x}_3^{\bar{n}_q}} + \frac{\langle \beta, \beta_q'' \rangle}{\bar{x}_3^{\bar{n}_q}} \right) \right] {\overset{\bullet}{\underset{\bullet}{}}}$$

$$\times \exp\left( \int \alpha(x_1)^+ \right) \exp\left( \int \alpha'(x_2)^+ \right) \exp\left( \int \alpha''(x_3)^+ \right)$$

$$\times \exp\left( \int \beta(\bar{x}_1)^+ \right) \exp\left( \int \beta'(\bar{x}_2)^+ \right) \exp\left( \int \beta''(\bar{x}_3)^+ \right)$$

$$\times \; \mathrm{e}_{\lambda} \mathrm{e}_{\lambda'} \mathrm{e}_{\lambda''} x_1^{\alpha} \bar{x}_1^{\beta} x_2^{\alpha'} \bar{x}_2^{\beta'} x_3^{\alpha''} \bar{x}_3^{\beta''} \, .$$

$$\text{(B.8)}$$

To prove locality, we substitute formal variables with complex numbers $x_i \to z_i, \bar{x}_i \to \bar{z}_i, \; i = 1, 2, 3$. We can write

$$Y_{V_\Lambda}(u, z_1, \bar{z}_1) Y_{V_\Lambda}(v, z_2, \bar{z}_2) Y_{V_\Lambda}(w, z_3, \bar{z}_3)$$

$$= f(z_1, z_2, z_3, \bar{z}_1, \bar{z}_2, \bar{z}_3) \times \exp\left(\int \alpha(z_1)^-\right) \exp\left(\int \alpha'(z_2)^-\right) \exp\left(\int \alpha''(z_3)^-\right)$$

$$\times \exp\left(\int \beta(\bar{z}_1)^-\right) \exp\left(\int \beta'(\bar{z}_2)^-\right) \exp\left(\int \beta''(\bar{z}_3)^-\right)$$

$$\times \; {}_\circ^\circ \prod_{i=1}^{k} \left[ \frac{1}{(l_i - 1)!} \frac{d^{l_i - 1} \alpha_i(z_1)}{dz_1^{l_i - 1}} + f_{1,i}(z_1, z_2, z_3) \right] \prod_{j=1}^{\bar{k}} \left[ \frac{1}{(\bar{l}_j - 1)!} \frac{d^{\bar{l}_j - 1} \beta_j(\bar{z}_1)}{d\bar{z}_1^{\bar{l}_j - 1}} + g_{1,j}(\bar{z}_1, \bar{z}_2, \bar{z}_3) \right] {}_\circ^\circ$$

$$\times \; {}_\circ^\circ \prod_{r=1}^{\ell} \left[ \frac{1}{(m_r - 1)!} \frac{d^{m_r - 1} \alpha'_r(z_2)}{dz_2^{m_r - 1}} + f_{2,r}(z_1, z_2, z_3) \right] \prod_{s=1}^{\bar{\ell}} \left[ \frac{1}{(\bar{m}_s - 1)!} \frac{d^{\bar{m}_s - 1} \beta'_s(\bar{z}_2)}{d\bar{z}_2^{\bar{m}_s - 1}} + g_{2,s}(\bar{z}_1, \bar{z}_2, \bar{z}_3) \right] {}_\circ^\circ$$

$$\times \; {}_\circ^\circ \prod_{p=1}^{m} \left[ \frac{1}{(n_p - 1)!} \frac{d^{n_p - 1} \alpha''_p(z_3)}{dz_3^{n_p - 1}} + f_{3,p}(z_1, z_2, z_3) \right] \prod_{q=1}^{\bar{m}} \left[ \frac{1}{(\bar{n}_q - 1)!} \frac{d^{\bar{n}_q - 1} \beta''_q(\bar{z}_3)}{d\bar{z}_3^{\bar{n}_q - 1}} + g_{3,q}(\bar{z}_1, \bar{z}_2, \bar{z}_3) \right] {}_\circ^\circ$$

$$\times \exp\left(\int \alpha(z_1)^+\right) \exp\left(\int \alpha'(z_2)^+\right) \exp\left(\int \alpha''(z_3)^+\right)$$

$$\times \exp\left(\int \beta(\bar{z}_1)^+\right) \exp\left(\int \beta'(\bar{z}_2)^+\right) \exp\left(\int \beta''(\bar{z}_3)^+\right) e_\lambda e_{\lambda'} e_{\lambda''} z_1^\alpha \bar{z}_1^\beta z_2^{\alpha'} \bar{z}_2^{\beta'} z_3^{\alpha''} \bar{z}_3^{\beta''} \, ,$$

$$\tag{B.9}$$

in the domain $|z_1| > |z_2| > |z_3|$, where

$$f(z_1, z_2, z_3, \bar{z}_1, \bar{z}_2, \bar{z}_3)$$
$$= \exp\left(\langle \alpha, \alpha' \rangle \log(z_1 - z_2)\right) \exp\left(\langle \beta, \beta' \rangle \log(\bar{z}_1 - \bar{z}_2)\right) \exp\left(\langle \alpha, \alpha'' \rangle \log(z_1 - z_3)\right)$$
$$\quad \exp\left(\langle \beta, \beta'' \rangle \log(\bar{z}_1 - \bar{z}_3)\right) \exp\left(\langle \alpha', \alpha'' \rangle \log(z_2 - z_3)\right) \exp\left(\langle \beta', \beta'' \rangle \log(\bar{z}_2 - \bar{z}_3)\right),$$
$$f_{1,i}(z_1, z_2, z_3) = (-1)^{l_i - 1} \left( \frac{\langle \alpha', \alpha_i \rangle}{\exp(l_i \log(z_1 - z_2))} - \frac{\langle \alpha', \alpha_i \rangle}{\exp(l_i \log z_1)} \right.$$
$$\left. + \frac{\langle \alpha'', \alpha_i \rangle}{\exp(l_i \log(z_1 - z_3))} - \frac{\langle \alpha'', \alpha_i \rangle}{\exp(l_i \log z_1)} \right)$$
$$g_{1,j}(\bar{z}_1, \bar{z}_2, \bar{z}_3) = (-1)^{\bar{l}_j - 1} \left( \frac{\langle \beta', \beta_j \rangle}{\exp(\bar{l}_j \log(\bar{z}_1 - \bar{z}_2))} - \frac{\langle \beta', \beta_j \rangle}{\exp(\bar{l}_j \log \bar{z}_1)} \right.$$
$$\left. + \frac{\langle \beta'', \beta_j \rangle}{\exp(\bar{l}_j \log(\bar{z}_1 - \bar{z}_3))} - \frac{\langle \beta'', \beta_j \rangle}{\exp(\bar{l}_j \log \bar{z}_1)} \right),$$

$$\tag{B.10}$$

$$f_{2,r}(z_1, z_2, z_3) = (-1)^{m_r-1} \left( \frac{\langle \alpha'', \alpha'_r \rangle}{\exp(m_r \log(z_2 - z_3))} - \frac{\langle \alpha'', \alpha'_r \rangle}{\exp(m_r \log z_2)} - \frac{\langle \alpha, \alpha'_r \rangle}{\exp(m_r \log z_2)} \right)$$
$$- \frac{\langle \alpha, \alpha'_r \rangle}{\exp(m_r \log(z_1 - z_2))},$$

$$g_{2,s}(\bar{z}_1, \bar{z}_2, \bar{z}_3) = (-1)^{\bar{m}_s-1} \left( \frac{\langle \beta'', \beta'_s \rangle}{\exp(\bar{m}_s \log(\bar{z}_2 - \bar{z}_3))} - \frac{\langle \beta'', \beta'_s \rangle}{\exp(m_r \log(\bar{z}_2))} - \frac{\langle \beta, \beta'_s \rangle}{\exp(m_r \log(\bar{z}_2))} \right)$$
$$- \frac{\langle \beta, \beta'_s \rangle}{\exp(\bar{m}_s \log(\bar{z}_1 - \bar{z}_2))},$$

$$f_{3,p}(z_1, z_2, z_3) = - \frac{\langle \alpha', \alpha''_p \rangle}{\exp(n_p \log(z_2 - z_3))} - \frac{\langle \alpha, \alpha''_p \rangle}{\exp(n_p \log(z_1 - z_3))}$$
$$- (-1)^{n_p-1} \left( \frac{\langle \alpha', \alpha''_p \rangle}{\exp(n_p \log z_3)} + \frac{\langle \alpha, \alpha''_p \rangle}{\exp(n_p \log z_3)} \right),$$

$$g_{3,q}(\bar{z}_1, \bar{z}_2, \bar{z}_3) = - \frac{\langle \beta', \beta''_q \rangle}{\exp(\bar{n}_q \log(\bar{z}_2 - \bar{z}_3))} - \frac{\langle \beta, \beta''_q \rangle}{\exp(\bar{n}_q \log(\bar{z}_1 - \bar{z}_3))}$$
$$- (-1)^{\bar{n}_q-1} \left( \frac{\langle \beta', \beta''_q \rangle}{\exp(\bar{n}_q \log \bar{z}_3)} + \frac{\langle \beta, \beta''_q \rangle}{\exp(\bar{n}_q \log \bar{z}_3)} \right).$$

$$\text{(B.11)}$$

One can check that under any permutation $\sigma \in S_3$ of the vertex operators, the product is given by the right hand side of (B.9) in the domain $|z_{\sigma(1)}| > |z_{\sigma(2)}| > |z_{\sigma(3)}|$.

For more than three vertex operators, the locality can be proved in a similar way.

# C   Heisenberg algebras and their representations

In this appendix, we prove some properties of Heisenberg algebras which is used in the paper. See [2, Section 1.7] for definitions. We start with some results about Lie algebras.

## C.1   Preliminaries

Let $U$ be a left $R$-module and $D = \text{End}_R(U)$ be the ring of $R$-linear endomorphisms of $U$. One can consider $U$ as a left $D$-module by defining

$$\varphi \cdot u = \varphi(u), \quad \varphi \in D, \ \ u \in U. \tag{C.1}$$

The $D$-linear (in)dependence of a subset of $U$ is defined in an obvious way. A $D$-linear operator on $U$ is a map $f : U \longrightarrow U$ such that

$$f(\varphi \cdot u - v) = \varphi \cdot f(u) - f(v) = \varphi(f(u)) - f(v), \quad \varphi \in D, \ \ u, v \in U. \tag{C.2}$$

**Theorem C.1.** (Jacobson Density Theorem, [61, Theorem 13.14]) *Let $U$ be a simple left $R$-module and write $D = \text{End}_R(U)$. Let $\alpha$ be any $D$-linear operator on $U$ and let $X \subseteq U$ be any finite $D$-linearly independent subset. Then there exists an element $r \in R$ such that $rx = \alpha x$ for all $x \in X$.*

Finally, we will need a generalisation of Schur's lemma to countably-infinite dimensional representations.

**Lemma C.1.** (Dixmier's Lemma, [62, Lemma 100]) *Suppose $V$ is a vector space over $\mathbb{C}$ of countable dimension and that $S \subset \text{End}(V)$ acts irreducibly. If $T \in \text{End}(V)$ commutes with every element of $S$, then $T$ is a scalar multiple of the identity operator.*

**Proposition C.1.** *Let $\mathfrak{g}_1, \mathfrak{g}_2$ be two complex Lie algebras (possibly countably infinite dimensional). Then every irreducible $(\mathfrak{g}_1 \oplus \mathfrak{g}_2)$-module which contains an irreducible $\mathfrak{g}_1$ or $\mathfrak{g}_2$-module is isomorphic to the tensor product of two irreducible $\mathfrak{g}_1, \mathfrak{g}_2$-modules. Conversely, the tensor product of irreducible $\mathfrak{g}_1, \mathfrak{g}_2$-modules of countable dimension is an irreducible $(\mathfrak{g}_1 \oplus \mathfrak{g}_2)$-module.*

*Proof.* Let $V$ be an irreducible $(\mathfrak{g}_1 \oplus \mathfrak{g}_2)$-module and suppose $Y \subseteq V$ is an irreducible $\mathfrak{g}_2$-module. $X = \text{Hom}_{\mathfrak{g}_2}(Y, V)$ is the space of $\mathfrak{g}_2$-linear maps from $Y$ to $V$, i.e.

$$X = \{\phi : Y \to V \mid \phi(g_2 \cdot v - w) = g_2 \cdot \phi(v) - \phi(w), \ \forall g_2 \in \mathfrak{g}_2, v, w \in Y\}. \tag{C.3}$$

Then $X$ is a $\mathfrak{g}_1$-module under the bilinear map

$$(g_1, \phi) \mapsto g_1 \cdot \phi : y \mapsto g_1 \cdot (\phi(y)). \tag{C.4}$$

It is easy to see that $X \otimes Y$ is a $(\mathfrak{g}_1 \oplus \mathfrak{g}_2)$-module with the action

$$(g_1, g_2) \cdot (\phi \otimes y) = (g_1 \cdot \phi_g) \otimes y + \phi \otimes (g_2 \cdot y) . \tag{C.5}$$

Moreover the natural map from $X \otimes Y$ to $V$

$$\phi \otimes y \mapsto \phi(y) , \tag{C.6}$$

is a $(\mathfrak{g}_1 \oplus \mathfrak{g}_2)$-module map. We first show the map (C.6) is non-zero. Take $\phi = \text{id} :$ $Y \to V$, which clearly lies in $X$, then $\text{id} \otimes y \mapsto \text{id}(y) = y$. Hence, the (non-zero) image of this $(\mathfrak{g}_1 \oplus \mathfrak{g}_2)$-module map, (C.6), is a $(\mathfrak{g}_1 \oplus \mathfrak{g}_2)$-module. As $V$ is an irreducible $(\mathfrak{g}_1 \oplus \mathfrak{g}_2)$-module, the image is entire $V$, and the map (C.6) is surjective. Finally, we show this map is injective. Indeed suppose $\phi \otimes v \mapsto \phi(v) = 0$. Then it suffices to show that either $v = 0$ or $\phi = 0$. Suppose $v \neq 0$. By $\mathfrak{g}_2$-linearity of $\phi$, we see that

$$0 = g_2 \cdot \phi(v) = \phi(g_2 \cdot v), \quad \forall g_2 \in \mathfrak{g}_2 . \tag{C.7}$$

By irreducibility of $Y$, we see that $\phi = 0$. This implies that $V \cong X \otimes Y$ as $(\mathfrak{g}_1 \oplus \mathfrak{g}_2)$-modules, and that the latter is also an irreducible module. As $X \otimes Y$ is an irreducible $(\mathfrak{g}_1 \oplus \mathfrak{g}_2)$-module, $X$ must be an irreducible $\mathfrak{g}_1$ module. Similar arguments apply when $Y$ is an irreducible $\mathfrak{g}_1$-module.

To prove the converse, we follow [63]. We want to show that if $X, Y$ are irreducible $\mathfrak{g}_1, \mathfrak{g}_2$-modules, then $X \otimes Y$ with $(\mathfrak{g}_1 \oplus \mathfrak{g}_2)$-action defined by

$$(g_1, g_2) \cdot (x \otimes y) = (g_1 \cdot x) \otimes y + x \otimes (g_2 \cdot y) , \tag{C.8}$$

is an irreducible $(\mathfrak{g}_1 \oplus \mathfrak{g}_2)$-module. Suppose $M \subset X \otimes Y$ be a non-trivial $(\mathfrak{g}_1 \oplus \mathfrak{g}_2)$-submodule. It suffices to show that $M$ contains non-trivial pure tensors and then the irreducibility of $X, Y$ will imply that $M = X \otimes Y$. Let us assume that $x \otimes y$ is a pure tensor in $M$. The irreducibility of $X, Y$ guarantees that $\mathfrak{g}_1 \cdot x = X$ and $\mathfrak{g}_2 \cdot y = Y$, which guarantees that $x \otimes Y$ and $X \otimes y \subset M$. Consider any pure tensor $\tilde{x} \otimes \tilde{y} \in V$, we know that $\tilde{x} \otimes y$ belongs in $M$. Applying only $\mathfrak{g}_2$ on this vector, we can show that $\tilde{x} \otimes \tilde{y} \in M$, which implies that $M = X \otimes Y$, since pure tensors span $X \otimes Y$.

We now show that $M$ contains a pure tensor. Since any $\mathfrak{g}$-module is also a $U(\mathfrak{g})$-module, our strategy is to produce a pure tensor from an arbitrary vector in $M$ using the action of an element of $U(\mathfrak{g}_1 \oplus \mathfrak{g}_2)$. Start with an arbitrary non-zero vector

$$v = \sum_{i=1}^{n} x_i \otimes y_i \in M . \tag{C.9}$$

Without loss of generality, we may choose $\{x_i\}$ to be linearly independent over $\mathbb{C}$ and $y_i \neq 0$ for all $i$. By Dixmier's Lemma (or Schur's lemma when $X, Y$ are finite dimensional), we have

$$\text{End}_{U(\mathfrak{g}_1)}(X) \cong \mathbb{C}, \quad \text{End}_{U(\mathfrak{g}_2)}(Y) \cong \mathbb{C} . \tag{C.10}$$

In the statement of Jacobson Density Theorem, choose $U = X, R = U(\mathfrak{g}_1)$, hence $D = \mathrm{End}_{U(\mathfrak{g}_1)}(X) = \mathbb{C}$, due to (C.10). Thus, let us take $\{x_1, x_2, \ldots, x_n\}$, which is a finite $D$-linearly independent subset. Consider a $\mathbb{C}$- linear map $\alpha$, such that

$$\alpha x_i = \begin{cases} x_1 & i = 1, \\ 0 & i > 1. \end{cases} \tag{C.11}$$

We can then use Jacobson Density Theorem to conclude that there exists $u \in U(\mathfrak{g}_1)$ such that

$$\alpha x_i = u x_i, \tag{C.12}$$

$\forall 1 \le i \le n$. Hence, for $u \otimes 1 \in U(\mathfrak{g}_1) \otimes U(\mathfrak{g}_2) \cong U(\mathfrak{g}_1 \oplus \mathfrak{g}_2)$,

$$(u \otimes 1)v = x_1 \otimes y_1, \tag{C.13}$$

which is a pure tensor. $\qquad\square$

## C.2 Modules of direct sum of Heisenberg algebras

We now prove some basic results about the modules of direct sum of Heisenberg algebras. We will use the notations of [2, Chapter 1] in this section.

For a $\mathbb{Z}$-graded vector space

$$\mathfrak{g} = \bigoplus_{n \in \mathbb{Z}} \mathfrak{g}_n, \tag{C.14}$$

we will write

$$\mathfrak{g}^+ := \bigoplus_{n > 0} \mathfrak{g}_n, \quad \mathfrak{g}^- := \bigoplus_{n < 0} \mathfrak{g}_n, \quad \mathfrak{g}^0 := \mathfrak{g}_0. \tag{C.15}$$

Let $\mathfrak{g}_1, \mathfrak{g}_2$ be two Heisenberg algebras with central elements $\mathbf{k}$ and $\bar{\mathbf{k}}$ respectively. Analogous to the $\mathfrak{C}_k$ condition for a module of a Heisenberg algebra (see [2, Page 23]), we define the $\mathfrak{C}_{k,\bar{k}}$ condition for an $(\mathbb{R} \times \mathbb{R})$- graded module $V$ of the direct sum $\mathfrak{g}_1 \oplus \mathfrak{g}_2$ of two Heisenberg algebras $\mathfrak{g}_1, \mathfrak{g}_2$. We say that $V$ satisfies the $\mathfrak{C}_{k,\bar{k}}$ condition if

1. $\mathbf{k}$ and $\bar{\mathbf{k}}$ act on $V$ by multiplication with $k$ and $\bar{k}$ respectively.

2. There exist $M_1, M_2 \in \mathbb{R}$, such that $V_{m,n} = 0$, if either $m > M_1$ and $n > M_2$.

The *vacuum space* of $V$, denoted by $\Omega_V \subset V$, is a subspace of non-zero vectors such that any $v \in \Omega_V$ satisfies $(\mathfrak{g}_1^+ \oplus \mathfrak{g}_2^+) \cdot v = 0$. Let $M(k), M(\bar{k})$ denote the unique (up to isomorphism) irreducible module of $\mathfrak{g}_1, \mathfrak{g}_2$ respectively. In particular, we have [2, Section 1.7]

$$M(k) \cong S(\mathfrak{g}_1^-), \quad M(\bar{k}) \cong S(\mathfrak{g}_2^-). \tag{C.16}$$

Then we have the following proposition.

**Proposition C.2.** *Let $\mathfrak{g}_1, \mathfrak{g}_2$ be two Heisenberg algebras with central elements $\mathbf{k}, \bar{\mathbf{k}}$ respectively. Then the following are true:*

1. *$M(k) \otimes M(\bar{k})$ is the unique (up to isomorphism) irreducible $(\mathfrak{g}_1 \oplus \mathfrak{g}_2)$-module satisfying condition $\mathfrak{C}_{k,\bar{k}}$.*

2. *The vacuum space $\Omega_{M(k) \otimes M(\bar{k})}$ is one-dimensional and*

$$\Omega_{M(k) \otimes M(\bar{k})} = \mathbb{C}(1_{M(k)} \otimes 1_{M(\bar{k})}) . \tag{C.17}$$

3. *For any $(\mathfrak{g}_1 \oplus \mathfrak{g}_2)$-module satisfying condition $\mathfrak{C}_{k,\bar{k}}$, the $(\mathfrak{g}_1 \oplus \mathfrak{g}_2)$-module generated by a vacuum vector is equivalent to $M(k) \otimes M(\bar{k})$.*

*Proof.* (1) Any $(\mathfrak{g}_1 \oplus \mathfrak{g}_2)$-module is in particular a $\mathfrak{g}_2$-module and hence completely reducible by [2, Proposition 1.7.2]. Thus it contains an irreducible $\mathfrak{g}_2$-module. By Proposition C.1, every such irreducible $(\mathfrak{g}_1 \oplus \mathfrak{g}_2)$-module is isomorphic to a tensor product $X \otimes Y$ where $X, Y$ are irreducible $\mathfrak{g}_1, \mathfrak{g}_2$-module respectively. By [2, Proposition 1.7.2] we have $X \otimes Y \cong M(k) \otimes M(\bar{k})$.

(2) Since $M(k), M(\bar{k})$ satisfies the conditions $\mathfrak{C}_k, \mathfrak{C}_{\bar{k}}$ of [2, Page 23], it is clear that $M(k) \otimes M(\bar{k})$ satisfies the condition $\mathfrak{C}_{k,\bar{k}}$. The vacuum space of $M(k) \otimes M(\bar{k})$ is easily seen to be the tensor product of the vacuum spaces of $M(k)$ and $M(\bar{k})$ respectively and hence is one-dimensional.

(3) Let $V$ be any $(\mathfrak{g}_1 \oplus \mathfrak{g}_2)$-module satisfying condition $\mathfrak{C}_{k,\bar{k}}$. Let $v \in V$ and consider the $(\mathfrak{g}_1 \oplus \mathfrak{g}_2)$-submodule of $V$

$$\{(g_1, g_2) \cdot v : (g_1, g_2) \in \mathfrak{g}_1 \oplus \mathfrak{g}_2\} . \tag{C.18}$$

It is easily seen that this gives an irreducible $(\mathfrak{g}_1 \oplus \mathfrak{g}_2)$-module and is isomorphic to $M(k) \otimes M(\bar{k})$ by (1). $\qquad\square$

We now have the following theorem.

**Theorem C.2.** *Any $(\mathfrak{g}_1 \oplus \mathfrak{g}_2)$-module is completely reducible and is isomorphic to copies of $M(k) \otimes M(\bar{k})$. More precisely, for any such module $V$, the (well-defined) canonical linear map*

$$\begin{aligned} f : U(\mathfrak{g}_1 \oplus \mathfrak{g}_2) \otimes_{U(\mathfrak{b}_1 \oplus \mathfrak{b}_2)} \Omega_V &\longrightarrow V , \\ u \otimes v \mapsto u \cdot v , \quad u \in U(\mathfrak{g}_1 \oplus \mathfrak{g}_2), v \in \Omega_V , \end{aligned} \tag{C.19}$$

*is a $(\mathfrak{g}_1 \oplus \mathfrak{g}_2)$-module isomorphism. In particular, the linear map*

$$\begin{aligned} M(k) \otimes M(\bar{k}) \otimes_{\mathbb{C}} \Omega_V &\cong U(\mathfrak{g}_1^- \oplus \mathfrak{g}_2^-) \otimes_{\mathbb{C}} \Omega_V \longrightarrow V , \\ u \otimes v \mapsto u \cdot v , \quad u \in U(\mathfrak{g}_1^- \oplus \mathfrak{g}_2^-), v \in \Omega_V , \end{aligned} \tag{C.20}$$

*defines a $(\mathfrak{g}_1 \oplus \mathfrak{g}_2)$-module isomorphism, $\Omega_V$ now regarded as a trivial $(\mathfrak{g}_1 \oplus \mathfrak{g}_2)$-module.*

*Proof.* We closely follow [2] for this proof. First, we show the $f$ in (C.19) is an injective map. Note that from the action of $f$, it is clear that $f$ is injective on $1 \otimes \Omega_V \hookrightarrow U(\mathfrak{g}_1 \oplus \mathfrak{g}_2) \otimes_{U(\mathfrak{b}_1 \oplus \mathfrak{b}_2)} \Omega_V$. Let $K$ be the kernel of $f$. Then it is easy to see that $K$ is a $(\mathfrak{g}_1 \oplus \mathfrak{g}_2)$-submodule of $U(\mathfrak{g}_1 \oplus \mathfrak{g}_2) \otimes_{U(\mathfrak{b}_1 \oplus \mathfrak{b}_2)} \Omega_V$ and has a grading induced from $U(\mathfrak{g}_1 \oplus \mathfrak{g}_2) \otimes_{U(\mathfrak{b}_1 \oplus \mathfrak{b}_2)} \Omega_V$. Thus $K$ satisfies the condition $\mathfrak{C}_{k,\bar{k}}$. It follows that it has a vacuum vector, say $v \in K$. But then $v \in \Omega_V$ because the vacuum space of $U(\mathfrak{g}_1 \oplus \mathfrak{g}_2) \otimes_{U(\mathfrak{b}_1 \oplus \mathfrak{b}_2)} \Omega_V$ is precisely $\Omega_V$. Since $f(v) = 0$, it contradicts the fact that $f$ is injective on $\Omega_V$.

We now prove the surjectivity of $f$. Suppose $V/\mathrm{Im}(f) \neq 0$. Since $\mathrm{Im}(f)$ is a $(\mathfrak{g}_1 \oplus \mathfrak{g}_2)$-submodule of $V$, $V/\mathrm{Im}(f)$ is naturally a $(\mathfrak{g}_1 \oplus \mathfrak{g}_2)$-module and satisfies the condition $\mathfrak{C}_{k,\bar{k}}$ since $V$ does. Then there exists a vacuum vector $w \in V/\mathrm{Im}(f)$. Let $w = [v]$ for some $v \in V$. It follows that $v \notin \mathrm{Im}(f)$ and

$$x_{1i} \cdot v, x_{2i} \cdot v \in \mathrm{Im}(f), \quad i \in \mathbb{Z}_+ , \tag{C.21}$$

since $x_{1i} \cdot [v], x_{2i} \cdot [v] = 0$. Moreover, due to the $\mathfrak{C}_{k,\bar{k}}$ property, there exists $i_0, j_0 \in \mathbb{Z}_+$ such that

$$x_{1i} \cdot v, x_{2j} \cdot v = 0 \ \text{ for all } \ i > i_0, j > j_0 . \tag{C.22}$$

We will now show that there exists $t \in \mathrm{Im}(f)$ such that

$$x_{ki} \cdot t = x_{ki} \cdot v, \ \text{ for all } \ i \in \mathbb{Z}_+ \text{ and } k \in \{1, 2\} . \tag{C.23}$$

This will imply that $t - v$ is a vacuum vector but $t - v \in V \setminus \Omega_V$, since $t - v \notin \mathrm{Im}(f)$ and $\Omega_V \subset \mathrm{Im}(f)$, which is a contradiction. To this end, choose a basis $\{\omega_\gamma\}_{\gamma \in \Gamma}$ ($\Gamma$ an index set) of $\Omega_V$. Then by injectivity of $f$ and the first isomorphism theorem

$$\mathrm{Im}(f) \cong \coprod_{\gamma \in \Gamma} U(\mathfrak{g}_1 \oplus \mathfrak{g}_2) \otimes_{U(\mathfrak{b}_1 \oplus \mathfrak{b}_2)} \mathbb{C}\omega_\gamma . \tag{C.24}$$

Let $s^1_{i\gamma}, s^2_{j\gamma}$ be the component of $x_{1i} \cdot v, x_{2j} \cdot v$ respectively under this decomposition. Then for any $i, i', j, j' \in \mathbb{Z}_+$ we have

$$x_{1i} x_{1i'} \cdot v = x_{1i'} x_{1i} \cdot v, \quad x_{2i} x_{2i'} \cdot v = x_{2i'} x_{2i} \cdot v. \tag{C.25}$$

as the Lie Bracket is zero on $\mathfrak{g}_1^+$ and $\mathfrak{g}_2^+$. This implies that for all $\gamma \in \Gamma$

$$x_{1i} \cdot s^1_{i'\gamma} = x_{1i'} \cdot s^1_{i\gamma}, \quad x_{2j} \cdot s^2_{j'\gamma} = x_{2j'} \cdot s^2_{j\gamma}. \tag{C.26}$$

It is also clear from (C.22) that

$$s^1_{i\gamma} = s^2_{j\gamma} = 0, \quad \text{for all } \ i > i_0, j > j_0 . \tag{C.27}$$

Moreover there is a finite subset $\Gamma_0 \subset \Gamma$ such that

$$s^1_{i\gamma} = s^2_{j\gamma} = 0, \quad \text{for all } \ \gamma \in \Gamma \setminus \Gamma_0 , \tag{C.28}$$

since any vector in $\text{Im}(f)$, in particular $x_{1i} \cdot v$ and $x_{2i} \cdot v$, has a finite decomposition in (C.24). Now, fix a $\gamma \in \Gamma_0$ and identify $U(\mathfrak{g}_1 \oplus \mathfrak{g}_2) \otimes_{U(\mathfrak{b}_1 \oplus \mathfrak{b}_2)} \mathbb{C}\omega_\gamma$ as the polynomial algebra over generators $y_{1i}, y_{2i}$. Then (C.26) implies that

$$\frac{\partial}{\partial y_{1i}} s^1_{i'\gamma} = \frac{\partial}{\partial y_{1i'}} s^1_{i\gamma}, \quad \frac{\partial}{\partial y_{2j}} s^2_{j'\gamma} = \frac{\partial}{\partial y_{2j'}} s^2_{j\gamma}, \quad i, i', j, j' \in \mathbb{Z}_+ . \tag{C.29}$$

Since $s^1_{i\gamma}, s^2_{j\gamma} = 0$ for $i > i_0, j > j_0$, it is clear from (C.29) that $s^1_{i\gamma}, s^2_{j\gamma}$ lie in the polynomial algebra generated by finitely many $y_{1i}, y_{2j}$ respectively for $i \leq i_0, j \leq j_0$. Thus there exists $s^1, s^2$ in these algebras such that

$$k \frac{\partial}{\partial y_{1i}} s^1 = s^1_{i\gamma}, \quad \bar{k} \frac{\partial}{\partial y_{2i}} s^2 = s^2_{i\gamma} , \tag{C.30}$$

for $i \leq i_0, j \leq j_0$ and hence for all $i, j \in \mathbb{Z}_+$. We then take $t^1_\gamma = s^1, t^2_\gamma = s^2$ and put

$$t := \sum_{\gamma \in \Gamma} (t^1_\gamma + t^2_\gamma) . \tag{C.31}$$

Note that $x_{2j} \cdot t^1_\gamma = x_{1i} \cdot t^2_\gamma = 0$ and hence (C.23) holds. $\qquad \square$

# D Proof of Conjecture 1 for $m = n$ Case

Suppose $\Lambda \subset \mathbb{R}^{m,m}$ is a Lorentzian lattice. We want to show that for any $f \in \mathrm{Aut}(\Lambda)$ we have

$$f(\Lambda_1^0) = \Lambda_1^0, \quad f(\Lambda_2^0) = \Lambda_2^0. \tag{D.1}$$

We will prove this in a series of results.

**Lemma D.1.** *Let $\Lambda$ and $\tilde{\Lambda}$ be two Lorentzian lattice related by an $\mathrm{O}(m,\mathbb{R}) \times \mathrm{O}(m,\mathbb{R})$ transformation $O$, i.e.*

$$\mathcal{G}_{\tilde{\Lambda}} = \mathcal{G}_\Lambda O. \tag{D.2}$$

*Then we have*

$$\mathrm{Aut}(\tilde{\Lambda}) = O\mathrm{Aut}(\Lambda)O^{-1}. \tag{D.3}$$

*Moreover $\mathrm{Aut}(\Lambda)$ preserves $\Lambda_i^0$ if and only if $\mathrm{Aut}(\tilde{\Lambda})$ preserves $\tilde{\Lambda}_i^0$, where $i \in \{1,2\}$.*

*Proof.* Let $f \in \mathrm{Aut}(\Lambda)$, then $\tilde{f} = OfO^{-1} \in \mathrm{Aut}(\tilde{\Lambda})$. Thus $O\mathrm{Aut}(\Lambda)O^{-1} \subseteq \mathrm{Aut}(\tilde{\Lambda})$. The reverse containment is similar.

Now suppose $\mathrm{Aut}(\tilde{\Lambda})$ preserves $\tilde{\Lambda}_1^0$. For any $(\alpha, 0) \in \Lambda_1^0$, we have $f(\alpha, 0) = (O^{-1}\tilde{f}O)(\alpha, 0)$ for some $\tilde{f} \in \mathrm{Aut}(\tilde{\Lambda})$. Since $O, O^{-1}$ preserves $\langle \alpha, \alpha \rangle$ and $\langle \alpha, \alpha \rangle$ for any $(\alpha, \beta) \in \Lambda$ it is clear that

$$f(\alpha, 0) = (\alpha', 0) \in \Lambda_1^0. \tag{D.4}$$

Similarly, $f$ preserves $\Lambda_2^0$. The converse is analogous. $\qquad\square$

**Theorem D.1.** *Let $\Lambda \subset \mathbb{R}^{m,m}$ be any Lorentzian lattice, then their exists an $\mathrm{O}(m,\mathbb{R}) \times \mathrm{O}(m,\mathbb{R})$ transformation which relates $\Lambda$ to a Lorentzian lattice $\Lambda_S$ with generator matrix of the form*

$$\mathcal{G}_{\Lambda_S} = \begin{pmatrix} \gamma^\star & \gamma^\star \\ \gamma\frac{B+\mathbb{1}}{2} & \gamma\frac{B-\mathbb{1}}{2} \end{pmatrix}, \tag{D.5}$$

*where $B$ is an anti-symmetric matrix, $\gamma$ is the generator matrix for a lattice $\Gamma$ in $\mathbb{R}^m$, and $\gamma^\star$ is the generator matrix for the dual lattice $\Gamma^\star$.*

*Proof.* We will use the result of [42, Appendix C] for the proof of this theorem. Let $\{e_i\}_{i=1}^{2m}$ be the standard basis of $\mathbb{R}^{m,m}$. Then we change the basis of $\mathbb{R}^{m,m}$ from standard basis to the basis $\{f_i\}_{i=1}^{2m}$:

$$\begin{aligned} f_i &= \frac{e_i + e_{m+i}}{\sqrt{2}}, \\ f_{m+i} &= \frac{e_i - e_{m+i}}{\sqrt{2}}, \end{aligned} \tag{D.6}$$

where $i \in \{1, \ldots, m\}$. Let $\mathcal{G}_\Lambda$ and $\widetilde{\mathcal{G}}_\Lambda$ be the generator matrix for $\Lambda$ in the $\{e_i\}$ and $\{f_i\}$ basis respectively. Now from Appendix C of [42], by an $O(m, \mathbb{R}) \times O(m, \mathbb{R})$ transformation, we can transform $\Lambda$ into the lattice $\Lambda_S$ with generator matrix[28]

$$\widetilde{\mathcal{G}}_{\Lambda_S} = \frac{1}{\sqrt{2}} \left( \begin{array}{c|c} 2\gamma^\star & 0 \\ \hline \gamma B & \gamma \end{array} \right), \tag{D.7}$$

in the $\{f_i\}$ basis for some antisymmetric matrix $B$. Changing the basis of $\mathbb{R}^{m,m}$ back to the standard basis $\{e_i\}$ amounts to right multiplying the generator matrix (D.7) by

$$\begin{pmatrix} \frac{\mathbb{1}}{\sqrt{2}} & \frac{\mathbb{1}}{\sqrt{2}} \\ \frac{\mathbb{1}}{\sqrt{2}} & -\frac{\mathbb{1}}{\sqrt{2}} \end{pmatrix}, \tag{D.8}$$

where $\mathbb{1}$ is the $m \times m$ identity matrix. This gives the generator matrix in (D.5). $\quad\square$

We now have the following important theorem.

**Theorem D.2.** *Let $\Lambda_S \subset \mathbb{R}^{m,m}$ be the Lorentzian lattice with generator matrix $\mathcal{G}_{\Lambda_S}$ given in (D.5). Then $\mathrm{Aut}(\Lambda_S)$ preserves $(\Lambda_S)_i^0$, $i = 1, 2$.*

*Proof.* Let $\{\alpha_i\}_{i=1}^m$ be an integral basis of $\Gamma$ and $\{\alpha_i^\star\}_{i=1}^m$ be the basis of $\Gamma^\star$ dual to $\{\alpha_i\}_{i=1}^m$, i.e.

$$\sum_{k=1}^m (\alpha_i)^k (\alpha_j^\star)^k = \delta_i^j, \quad \sum_{k=1}^m (\alpha_k^\star)^i (\alpha_k)^j = \delta_i^j. \tag{D.9}$$

Here the superscript $j$ over the vector denotes the $j$th component of the vector in the standard basis of $\mathbb{R}^m$. A general vector $\lambda \in \Lambda_S$ can be written as $\lambda = (\alpha, \beta)$ where

$$\begin{aligned}
\alpha^j &= \sum_{i=1}^m m_i (\alpha_i^\star)^j + \frac{1}{2} \sum_{i=1}^m \left( \sum_{k=1}^m B_{jk}(\alpha_i)^k + (\alpha_i)^j \right) n_i, \\
\beta^j &= \sum_{i=1}^m m_i (\alpha_i^\star)^j + \frac{1}{2} \sum_{i=1}^m \left( \sum_{k=1}^m B_{jk}(\alpha_i)^k - (\alpha_i)^j \right) n_i,
\end{aligned} \tag{D.10}$$

and $m_i, n_i \in \mathbb{Z}$. In vector notation we have

$$\begin{aligned}
\vec{\alpha} &= \vec{m}^T \gamma^\star + \vec{n}^T \left( \gamma \frac{B + \mathbb{1}}{2} \right), \\
\vec{\beta} &= \vec{m}^T \gamma^\star + \vec{n}^T \left( \gamma \frac{B - \mathbb{1}}{2} \right).
\end{aligned} \tag{D.11}$$

---

[28]Note that in our convention, the rows of the generator matrix are basis for the lattice while in [42] it is the columns.

We then have

$$
\begin{aligned}
\langle \alpha, \alpha \rangle = \sum_{i,j=1}^{m} & \left[ m_i m_j \langle \alpha_i^\star, \alpha_j^\star \rangle + 2 \times \frac{1}{2} m_i n_j \sum_{k,\ell=1}^{m} B_{k\ell}(\alpha_i^\star)^k (\alpha_j)^\ell + 2 \times \frac{1}{2} m_i n_j \langle \alpha_i^\star, \alpha_j \rangle \right. \\
& \left. + \frac{1}{4} \sum_{k,\ell,p=1}^{m} n_i n_p B_{jk}(\alpha_i)^k B_{j,\ell}(\alpha_p)^\ell + 2 \times \frac{1}{4} \sum_{k,\ell=1}^{m} n_i n_\ell B_{jk}(\alpha_i)^k (\alpha_l)^j + \frac{1}{4} n_i n_j \langle \alpha_i, \alpha_j \rangle \right] \\
= \vec{m}^T \mathbf{g}^{-1} \vec{m} & + \sum_{i,j,k,\ell,p,q=1}^{m} m_i n_j (\alpha_i^\star)^p (\alpha_q^\star)^p (\alpha_q)^k B_{k\ell}(\alpha_j)^\ell + \vec{m}^T \vec{n} \\
+ \frac{1}{4} \sum_{i,j,k,\ell,p,u,v,s,t=1}^{m} & n_i n_p (\alpha^\star)_u^j (\alpha_u)^s B_{sk}(\alpha_i)^k (\alpha^\star)_v^j (\alpha_v)^t B_{t\ell}(\alpha_p)^\ell + \frac{1}{4} \vec{n}^T \mathbf{g} \vec{n} \\
= \vec{m}^T \mathbf{g}^{-1} \vec{m} & + \vec{m}^T \mathbf{b} \mathbf{g}^{-1} \vec{n} + \vec{m}^T \vec{n} - \frac{1}{4} \vec{n}^T \mathbf{b} \mathbf{g}^{-1} \mathbf{b} \vec{n} + \frac{1}{4} \vec{n}^T \mathbf{g} \vec{n} \\
= \vec{m}^T \mathbf{g}^{-1} \vec{m} & + \vec{m}^T \mathbf{b} \mathbf{g}^{-1} \vec{n} + \vec{m}^T \vec{n} + \frac{1}{4} \vec{n}^T (\mathbf{g} - \mathbf{b} \mathbf{g}^{-1} \mathbf{b}) \vec{n},
\end{aligned}
\tag{D.12}
$$

where $\vec{m}, \vec{n} \in \mathbb{Z}^m$ are column vectors, $\mathbf{g}_{ij} = \langle \alpha_i, \alpha_j \rangle$ and $\mathbf{g}_{ij}^\star = \mathbf{g}_{ij}^{-1} = \langle \alpha_i^\star, \alpha_j^\star \rangle$ are the Gram matrices of $\Gamma$ and $\Gamma^\star$ respectively and $\mathbf{b}_{ij} = (\alpha_i)^k B_{k\ell}(\alpha_j)^\ell$ is the antisymmetric matrix $B$ in the $\{\alpha_i\}$ basis of $\mathbb{R}^m$. In the last step we used (D.9) and the fact that $B$ and $\mathbf{b}$ are antisymmetric. In terms of matrices we have

$$
\begin{aligned}
\mathbf{g} = \gamma \gamma^T, \quad \mathbf{g}^\star &= \gamma^\star (\gamma^\star)^T = (\gamma^T)^{-1} \gamma^{-1} = \mathbf{g}^{-1}, \\
\mathbf{b} = \gamma B \gamma^T, \quad B &= \gamma^{-1} \mathbf{b} \gamma^\star.
\end{aligned}
\tag{D.13}
$$

Similarly we have

$$
\langle \beta, \beta \rangle = \vec{m}^T \mathbf{g}^{-1} \vec{m} + \vec{m}^T \mathbf{b} \mathbf{g}^{-1} \vec{n} - \vec{m}^T \vec{n} + \frac{1}{4} \vec{n}^T (\mathbf{g} - \mathbf{b} \mathbf{g}^{-1} \mathbf{b}) \vec{n}.
\tag{D.14}
$$

From [64, Chapter 10], we have that the following transformations generate the group $\mathrm{O}(m, m, \mathbb{Z})$:

$$
\begin{aligned}
(1) \quad \vec{m} \leftrightarrow \vec{n} \quad \text{and} \quad 2\mathbf{g}^{-1} &\leftrightarrow \frac{1}{2}\left( \mathbf{g} - \mathbf{b} \mathbf{g}^{-1} \mathbf{b} \right) \\
\mathbf{b} \mathbf{g}^{-1} &\leftrightarrow -\mathbf{g}^{-1} \mathbf{b}, \\
(2) \quad \vec{m} \rightarrow \vec{m} - \mathbf{N} \vec{n} \quad \text{and} \quad \mathbf{b} &\rightarrow \mathbf{b} + 2\mathbf{N},
\end{aligned}
\tag{D.15}
$$

where $\mathbf{N}$ is an arbitrary anti-symmetric matrix with integer entries. These transformations must be understood in the following way: a transformation on the lattice can be implemented in two ways, first by action on the integer coordinates $(\vec{m}^T, \vec{n}^T)$ and second, by acting on the generator matrix $\mathcal{G}_{\Lambda_S}$ from the left. The transformations in (D.15) is a composition of both of these. To show that these transformations indeed generate the automorphism group $\mathrm{O}_{\Lambda_S}(m, m, \mathbb{Z})$ we need to show that these transformations leave the inner product invariant and preserves the lattice. It is easy

to check that under these transformations the $\langle \alpha, \alpha \rangle$ and $\langle \beta, \beta \rangle$ are preserved [64]. In particular these transformations preserve the Lorentzian inner product. We now show that these transformations preserve the lattice. Let us start with (1). It is clear that $\vec{m} \leftrightarrow \vec{n}$ preserves the lattice. We now check that the transformation on the generator matrix preserves the lattice. First note that from (D.15) we get[29]

$$
\begin{aligned}
\mathbf{g}^{-1} &\leftrightarrow \frac{1}{4} \left( \mathbf{g} - \mathbf{b}\mathbf{g}^{-1}\mathbf{b} \right) , \\
\mathbf{b} &\leftrightarrow 4 \left( \mathbf{b} - \mathbf{g}\mathbf{b}^{-1}\mathbf{g} \right)^{-1} .
\end{aligned}
\tag{D.16}
$$

From these transformations, we want to find how $\gamma$ and $\gamma^\star$ transform. Using the fact that $\mathbf{b}$ is antisymmetric, we guess the transformation to be

$$
\begin{aligned}
\gamma &\leftrightarrow \pm 2 \left( \gamma \pm \mathbf{b}\gamma^\star \right)^\star =: \pm\gamma_\pm , \\
\gamma^\star &\leftrightarrow \pm \frac{1}{2} \left( \gamma \pm \mathbf{b}\gamma^\star \right) =: \pm\gamma_\pm^\star .
\end{aligned}
\tag{D.17}
$$

It is easy to check that (D.17) reproduces the first equation in (D.16). These are all the solutions to the first equation in (D.15) but as we will show below, only two of them preserve the lattice. We also see that under the second equation of (D.16) we have

$$
B = \gamma^{-1}\mathbf{b}\gamma^\star \leftrightarrow \left( \gamma + \mathbf{b}\gamma^\star \right)^T \left( \mathbf{b} - \mathbf{g}\mathbf{b}^{-1}\mathbf{g} \right)^{-1} \left( \gamma + \mathbf{b}\gamma^\star \right) =: B'.
\tag{D.18}
$$

We claim that the following two transformations on the generator matrix preserve the lattice:

$$
\begin{aligned}
\text{(i)} \quad \mathcal{G}_{\Lambda_S} &= \begin{pmatrix} \gamma^\star & \gamma^\star \\ \gamma\frac{B+\mathbb{1}}{2} & \gamma\frac{B-\mathbb{1}}{2} \end{pmatrix} \to \mathcal{G}_{\Lambda_S}^{(i)} := \begin{pmatrix} \gamma_+^\star & -\gamma_-^\star \\ \gamma_+\frac{B'+\mathbb{1}}{2} & -\gamma_-\frac{B'-\mathbb{1}}{2} \end{pmatrix} , \\
\text{(ii)} \quad \mathcal{G}_{\Lambda_S} &= \begin{pmatrix} \gamma^\star & \gamma^\star \\ \gamma\frac{B+\mathbb{1}}{2} & \gamma\frac{B-\mathbb{1}}{2} \end{pmatrix} \to \mathcal{G}_{\Lambda_S}^{(ii)} := \begin{pmatrix} -\gamma_+^\star & \gamma_-^\star \\ -\gamma_+\frac{B'+\mathbb{1}}{2} & \gamma_-\frac{B'-\mathbb{1}}{2} \end{pmatrix} .
\end{aligned}
\tag{D.19}
$$

We now do some manipulations to prove our claim. We have

$$
\gamma_+^\star = \frac{1}{2} \left( \gamma + \mathbf{b}\gamma^\star \right) = \frac{1}{2}\gamma \left( B + \mathbb{1} \right).
\tag{D.20}
$$

Similarly

$$
\gamma_-^\star = -\frac{1}{2}\gamma \left( B - \mathbb{1} \right).
\tag{D.21}
$$

---

[29]Note that we are assuming that $\mathbf{b}$ and $B$ are invertible in writing this transformation but one can also write the transformation in a nonsingular way even when $\mathbf{b}$ is not invertible [46]. The manipulations below will also be modified in a nonsingular way.

Next we have

$$\gamma_+ \frac{B' + \mathbb{1}}{2} = 2\left(\gamma + \mathbf{b}\gamma^\star\right)^\star \left[\frac{\left(\gamma + \mathbf{b}\gamma^\star\right)^T \left(\mathbf{b} - \mathbf{g}\mathbf{b}^{-1}\mathbf{g}\right)^{-1} \left(\gamma + \mathbf{b}\gamma^\star\right) + \mathbb{1}}{2}\right] \tag{D.22}$$

$$= \left(\mathbf{b} - \mathbf{g}\mathbf{b}^{-1}\mathbf{g}\right)^{-1} \left(\gamma + \mathbf{b}\gamma^\star\right) + \left(\gamma + \mathbf{b}\gamma^\star\right)^\star.$$

Observe that

$$
\begin{aligned}
\left(\mathbf{b} - \mathbf{g}\mathbf{b}^{-1}\mathbf{g}\right)^{-1} \left(\gamma + \mathbf{b}\gamma^\star\right) &= \left(\mathbf{b} - \mathbf{g}\mathbf{b}^{-1}\mathbf{g}\right)^{-1} \left(\gamma + \gamma B\right) \\
&= \left(\mathbf{b} - \mathbf{g}\mathbf{b}^{-1}\mathbf{g}\right)^{-1} \gamma(\mathbb{1} + B) \\
&= \left(\gamma^{-1}\mathbf{b} + \gamma^{-1}\mathbf{g}\mathbf{b}^{-1}\mathbf{g}\right)^{-1} (\mathbb{1} + B) \\
&= \left(B\gamma^T - \gamma^T\mathbf{b}^{-1}\gamma\gamma^T\right)^{-1} (\mathbb{1} + B) \\
&= \left(B\gamma^T - (\gamma B)^{-1}\gamma\gamma^T\right)^{-1} (\mathbb{1} + B) \\
&= \left(\left(B - B^{-1}\right)\gamma^T\right)^{-1} (\mathbb{1} + B) \\
&= \gamma^\star \left(B - B^{-1}\right) (\mathbb{1} + B).
\end{aligned} \tag{D.23}
$$

Next note that

$$B - B^{-1} = \left(\mathbb{1} + B\right)\left(\mathbb{1} - B^{-1}\right). \tag{D.24}$$

Then we get

$$
\begin{aligned}
\left(\mathbf{b} - \mathbf{g}\mathbf{b}^{-1}\mathbf{g}\right)^{-1} \left(\gamma + \mathbf{b}\gamma^\star\right) &= \gamma^\star \left(\mathbb{1} - B^{-1}\right)^{-1} \\
&= \gamma^\star \left((B - \mathbb{1})B^{-1}\right)^{-1} \\
&= -\gamma^\star B(\mathbb{1} - B)^{-1}.
\end{aligned} \tag{D.25}
$$

We also have

$$
\begin{aligned}
\left(\gamma + \mathbf{b}\gamma^\star\right)^\star &= \left(\gamma(\mathbb{1} + B)\right)^\star \\
&= \gamma^\star(\mathbb{1} - B)^{-1}.
\end{aligned} \tag{D.26}
$$

Putting all this together we get

$$\gamma_+ \frac{B' + \mathbb{1}}{2} = \gamma^\star. \tag{D.27}$$

Similarly we have

$$\gamma_- \frac{B' - \mathbb{1}}{2} = \gamma^\star \left[\left(B - B^{-1}\right)^{-1} (\mathbb{1} - B) - (\mathbb{1} + B)^{-1}\right]. \tag{D.28}$$

We now use

$$B - B^{-1} = -(\mathbb{1} - B)\left(\mathbb{1} + B^{-1}\right), \tag{D.29}$$

to get

$$\gamma_- \frac{B' - \mathbb{1}}{2} = -\gamma^\star. \tag{D.30}$$

From (D.20), (D.21), (D.27) and (D.30) we see that

$$\mathcal{G}_{\Lambda_S}^{(i)} = \begin{pmatrix} \gamma\frac{B+\mathbb{1}}{2} & \gamma\frac{B-\mathbb{1}}{2} \\ \gamma^\star & \gamma^\star \end{pmatrix} = \begin{pmatrix} 0 & \mathbb{1} \\ \mathbb{1} & 0 \end{pmatrix} \mathcal{G}_{\Lambda_S} \tag{D.31}$$

Since $\left(\begin{smallmatrix} 0 & 1 \\ 1 & 0 \end{smallmatrix}\right)$ is determinant $-1$ and integral, the transformation (i) of (D.19) preserves the lattice. Similarly we have

$$\mathcal{G}_{\Lambda_S}^{(ii)} = \begin{pmatrix} -\gamma\frac{B+\mathbb{1}}{2} & \gamma\frac{B-\mathbb{1}}{2} \\ -\gamma^\star & \gamma^\star \end{pmatrix} = \begin{pmatrix} 0 & -\mathbb{1} \\ -\mathbb{1} & 0 \end{pmatrix} \mathcal{G}_{\Lambda_S} \tag{D.32}$$

and hence preserves the lattice. The full transformation (1) of (D.15) acting on the integer coordinates as well as the generator matrix can be written as a transformation acting only on the integer coordinates as follows:

$$\text{(i)} \ \ (\vec{m}^T, \vec{n}^T) \to (\vec{m}^T, \vec{n}^T), \quad \text{(ii)} \ \ (\vec{m}^T, \vec{n}^T) \to (-\vec{m}^T, -\vec{n}^T). \tag{D.33}$$

This generates a $\mathbb{Z}_2$ subgroup of the automorphism group $\mathrm{O}_{\Lambda_S}(m, m, \mathbb{Z})$. Obviously these transformations preserve $\langle \alpha, \alpha \rangle$ and $\langle \beta, \beta \rangle$ and hence the full Lorentzian norm. One could check directly that other choices in (D.17) does not preserve the lattice. Now, we will show that the second transformation in (D.15) preserves the lattice too.

Using the transformation of $\mathbf{b} \to \mathbf{b} + 2\mathbf{N}$, we obtain

$$\gamma B = \mathbf{b}\gamma^\star \to \mathbf{b}\gamma^\star + 2\mathbf{N}\gamma^\star = \gamma B + 2\mathbf{N}\gamma^\star. \tag{D.34}$$

We can input the above relation into the generator matrix to see how it transforms:

$$\mathcal{G}_{\Lambda_S} = \begin{pmatrix} \gamma^\star & \gamma^\star \\ \gamma\frac{B+\mathbb{1}}{2} & \gamma\frac{B-\mathbb{1}}{2} \end{pmatrix} \to \mathcal{G}'_{\Lambda_S} := \begin{pmatrix} \gamma^\star & \gamma^\star \\ \gamma\frac{B+\mathbb{1}}{2} + \mathbf{N}\gamma^\star & \gamma\frac{B-\mathbb{1}}{2} + \mathbf{N}\gamma^\star \end{pmatrix}. \tag{D.35}$$

From the above equation it follows that

$$\mathcal{G}'_{\Lambda_S} = \begin{pmatrix} \mathbb{1} & 0 \\ \mathbf{N} & \mathbb{1} \end{pmatrix} \mathcal{G}_{\Lambda_S}. \tag{D.36}$$

Again since $\left(\begin{smallmatrix} \mathbb{1} & 0 \\ \mathbf{N} & \mathbb{1} \end{smallmatrix}\right)$ is unimodular and integral, this transforamtion preserves the lattice. The transformation on integer coordinates in (2) of (D.15) can also be nicely represented as

$$(\vec{m}^T, \vec{n}^T) \to ((\vec{m} - \mathbf{N}\vec{n})^T, \vec{n}^T) = (\vec{m} + \vec{n}^T\mathbf{N}, \vec{n}^T) = (\vec{m}^T, \vec{n}^T) \begin{pmatrix} \mathbb{1} & 0 \\ \mathbf{N} & \mathbb{1} \end{pmatrix}, \tag{D.37}$$

where we have used that $\mathbf{N}$ is an anti-symmetric integral matrix. To get the complete transformation on the integer coordinates fixing without transforming the generator matrix, we compose these two transformations to see that

$$(\vec{m}^T, \vec{n}^T) \to (\vec{m}^T, \vec{n}^T) \begin{pmatrix} \mathbb{1} & 0 \\ \mathbf{N} & \mathbb{1} \end{pmatrix} \begin{pmatrix} \mathbb{1} & 0 \\ \mathbf{N} & \mathbb{1} \end{pmatrix} = (\vec{m}^T + 2\vec{n}^T\mathbf{N}, \vec{n}^T). \tag{D.38}$$

As $\mathbf{N}$ is integral, we see that the transformed vector lies on the lattice again. Checking that this transformation preserves the norm is a straightforward computation.

Thus all automorphisms of $\Lambda_S$ will preserve $\langle \alpha, \alpha \rangle$ and $\langle \beta, \beta \rangle$. Now suppose $(\alpha, 0) \in (\Lambda_S)_1^0$ maps to $(\alpha', \beta') \in \Lambda_S$ under some automorphism. Since $\langle \beta', \beta' \rangle = 0$ and the norm on $\mathbb{R}^m$ is positive definite, we conclude that $\beta' = 0$ and the automorphism preserves $(\Lambda_S)_1^0$. Similarly $(\Lambda_S)_2^0$ is also preserved under any automorphims of $\Lambda_S$. $\qquad\square$

Combining Theorem D.2, Theorem D.1 and Lemma D.1 proves Conjecture 1 for $m = n$ case. We remark that the methods of this Appendix clearly do not apply to $m \neq n$ case. One needs new tricks to prove the general result.

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
