# Peer review of "Non-Chiral Vertex Operator Algebra Associated To Lorentzian Lattices And Narain CFTs"

_SciPost Physics_

## Round 1 · Referee Report · Anonymous (Referee 1) · 2024-4-19

Strengths

  1. This paper gives mathematically rigorous and axiomatic definitions to various physical notions in 2-dimensional conformal field theory.

  2. This paper presents a new interesting construction of an example from a certain Lorentzian lattice.

  3. This paper formulats a precise mathematical conjecture, and presents a description of a certain moduli space based on this conjecture.

Weaknesses

  1. I wonder whether certain terminology is appropriate.

  2. Relations to preceding works should be clearly given.

  3. I have a doubt about validity of one remark.

Report

The authors formulate a non-chiral vertex operator algebra, its modules and intertwiners, and a non-chiral conformal field theory in terms of mathematical axioms, and construct an example for each even integral Lorentzian lattice. This construction extends a well-known construction of a vertex operator algebra for each even integral Euclidean lattice. They study the moduli space of non-chiral CFTs over certain Lorentzian lattices and identify this space based on a conjecture given in this paper. Then they show that Narain CFT's in physics literature fall within this framework. This is a nice contribution to mathematical physics of conformal field theory, and I recommend publication of this paper.

Requested changes

(1) The name "non-chiral vertex operator algebra" sounds unnatural to me and "full vertex operator algebra" sounds much more natural. If the authors insists on this name, one needs some justification.

(2) Relations of the formulation of the authors to that in Moriwaki [8] should be clarified.

(3) page 4:
Hence. N_{i,0}=\delta_{i,0}, N_{0,\bar i}=\delta_{0,\bar i}.

I don't understand this claim. There are type I modular invariants corresponding to D_{2n}, E_6, and E_8 as in the following paper. For them, N_{i,0} can be 1 for nonzero i, where the algebra A is given by the SU(2)_k WZW-model. There are many such examples.
Cappelli, A.; Itzykson, C.; Zuber, J.-B.
The A-D-E classification of minimal and A^{(1)}_1 conformal invariant theories
Comm. Math. Phys. 113 (1987), no. 1, 1-26.

(4) page 9:
Infact --> In fact

Recommendation

Ask for minor revision

  • validity: high
  • significance: high
  • originality: good
  • clarity: high
  • formatting: excellent
  • grammar: excellent

Author:  Ranveer Singh  on 2024-05-14  [id 4486]

(in reply to Report 1 on 2024-04-19)
Category:
answer to question
reply to objection
correction

We thank the reviewer for comments. Here are our replies to the questions raised by the reviewer:

(1) We use the term "non-chiral VOA" to distinguish from "full-field algebras" of Huang-Kong and "full VOAs" of Moriwaki since our definition differ in some key aspects. We have added a paragraph to emphasize this in the most recent update.

(2) We have added a paragraph in the Introduction explaining the relation to Moriwaki's work in the most recent update.

(3) By chiral algebra in the Introduction, we mean the maximally extended algebra of chiral operators. For example in D_4, if the chiral algebra is taken to be Virasoro then N_{3,0} is 1. However the maximally extended chiral algebra is W_3 algebra. In this case, the Verma module corresponding to the (3,0) Virasoro primary is included in the vacuum module of the W_3 algebra. The decomposition of the partition function into sum over characters changes and can be found for example in Di Francesco et.al. We have added a line to clarify that we always consider the maximally extended chiral algebra.

(4) We have corrected the typo.

We will resubmit the updated version of the paper once prompted by the editor.

---

## Round 1 · Referee Report · Anonymous (Referee 2) · 2024-5-13

Report

n this paper, the authors introduce the concept of non-chiral VOA, which gives a mathematical definition of conformal field theory, derive its basic properties, and mathematically construct a family of conformal field theories called Narain CFT. However, as pointed out in the attached file, a great number of definitions, propositions, and main theorems have
already been written in the previous work [1] (submitted to arXiv in 2020).
In [1], Moriwaki introduced the notion of full VOA and showed that a full VOA admits exactly marginal deformations if the full VOA has Heisenberg VOA as a subalgebra. In particular, he constructed the Narain CFT as an example of such a deformation family [1, Section 6.3].
However, this paper only mentions a single sentence in the introduction about prior work, and re-proves the same proposition and theorem in essentially the same way, without proper citation. This paper should be properly rewritten to show what is new and what is an existing result.
[1] Two-dimensional conformal field theory, full vertex algebra and current-current deformation, Adv. Math, 427, 2023

Attachment

Recommendation

Reject

  • validity: poor
  • significance: poor
  • originality: poor
  • clarity: good
  • formatting: good
  • grammar: good

Author:  Ranveer Singh  on 2024-05-15  [id 4489]

(in reply to Report 2 on 2024-05-13)
Category:
validation or rederivation

We thank the reviewer for comments. Here are our replies to the reviewer's comments:

However, this paper only mentions a single sentence in the introduction about prior work, and re-proves the same proposition and theorem in essentially the same way, without proper citation. This paper should be properly rewritten to show what is new and what is an existing result.

We have mentioned in the introduction of our paper that the definition of non-chiral VOA is inspired from the previous works of Huang-Kong and Moriwaki. We believe that some of our lemmas, propositions and theorems are inspired by earlier works of Frenkel-Lepowsky-Meurman (FLM), Frenkel-Huang-Leposwky (FHL) and Dolan-Goddard-Montague (DGM) from the 80s and 90s rather than that of Moriwaki. We have extended the statement and proof to the case of non-chiral VOA as the works of FLM, FHL and DGM are for chiral theories. So we do not believe that we need to rewrite our paper.

The definition of non-chiral VOA in this paper differs from that in [1]. In [1], full VOA is defined using bootstrap equation, but in this paper, it is defined using locality of multi-point correlation functions. In particular, it is not fully checked in [1] that Narain CFT constructed in [1] satisfies the axiom of non-chiral VOA in the sense of this paper.

As already pointed out by the reviewer, our notion of non-chiral VOA differs from that of Moriwaki's full VOA in that non-chiral VOA requires the locality of multi-point correlators as in full field algebra of Huang-Kong (which does not follow from locality of 2-point correlator as in VOA) while Moriwaki only requires locality of 2-point correlators and invokes bootstrap equations for locality of higher point correlators. In particular, for our example of Lorentzian lattice VOAs we discuss the locality of multi-point correlators. We also discuss intertwining operators for non-chiral VOA which is not discussed in Moriwaki.

Note that If it is a non-chiral VOA, then it is a full VOA in the sense of [1] (Proposition 2.2 in this paper). Although most of the pages of this paper overlap with [1], the referee believes that it is appropriate to take the form of a citation for these overlapping propositions, and that it is appropriate to recapitulate only the new parts as a new paper.

In this paper, we have proposed a definition and so we are obliged to provide proofs of results even though some form of those results appear somewhere else. We did this to make the paper self-contained. We have clearly indicated the reference for the lemmas and propositions which are inspired from previous works. So we do not think we need to delete parts of the paper and replace it with citation as it would reduce the clarity of the paper.

Lemma 2.1, Lemma 2.2, Lemma 2.3, Prop 2.1, Prop 2.2 can be found in [1, Prop 3.7 and Section 4.3]

Lemma 2.1, 2.2, 2.3 are extensions of these results from FHL and we have added the citation in the updated version. In Prop 2.1, 2.2 we have already cited the papers from which the propositions are inspired by. These papers are much older than Moriwaki's.

Definition of chiral vector (def 2.2) can be found [1, Section 3.2] and modules [1, Section 3.1].

Chiral vectors are clearly inspired from physics literature, we do not claim it to be our contribution. We believe it has existed in physics literature since 80s from the works of Moore-Seiberg and others.

Theorem 2.1 can be found in [1, Lemma 3.11]

This result is an extension of the proof of Borcherd's identity in Frenkel-Ben Zvi's book which was cited in the paper.

Construction of Narain CFT as full VOA can be found in [1, Section 4.4, Prop 4.10]

We will add a citation to this result. As already pointed out by the reviewer, Moriwaki's paper has not completely checked that Narain CFT is a non-chiral VOA.

Double coset description [1, Theorem 6.5] with example for Narain CFT [1, Section 6.3]

The double coset description of the moduli space of Narain CFT is a common knowledge in physics. We have provided a rigorous proof of the description based on non-chiral VOAs. This requires us to discuss a result about automorphisms of Lorentzian lattices. For general signature $(m,n)$ with $m\neq n$ of Lorentzian lattices, we have formulated a conjecture about these automorphisms of the lattice which is required to establish the moduli space of "Lorentzian lattice CFTs". These details are not contained in Moriwaki's work.

---

## Editorial Decision

resubmitted